# Clothianidin seed-treatment has no detectable negative impact on honeybee colonies and their pathogens

Julia Osterman[1,2,3], Dimitry Wintermantel [1,4,5], Barbara Locke [1], Ove Jonsson[6,7], Emilia Semberg [1], Piero Onorati[1], Eva Forsgren [1], Peter Rosenkranz[8], Thorsten Rahbek-Pedersen[9], Riccardo Bommarco [1], Henrik G. Smith [10,11], Maj Rundlöf [10,12] & Joachim R. de Miranda [1]

Interactions between multiple stressors have been implicated in elevated honeybee colony losses. Here, we extend our landscape-scale study on the effects of placement at clothianidin seed-treated oilseed rape fields on honeybees with an additional year and new data on honeybee colony development, swarming, mortality, pathogens and immune gene expression. Clothianidin residues in pollen, nectar and honeybees were consistently higher at clothianidin-treated fields, with large differences between fields and years. We found large variations in colony development and microbial composition and no observable negative impact of placement at clothianidin-treated fields. Clothianidin treatment was associated with an increase in brood, adult bees and *Gilliamella apicola* (beneficial gut symbiont) and a decrease in *Aphid lethal paralysis virus* and *Black queen cell virus* - particularly in the second year. The results suggest that at colony level, honeybees are relatively robust to the effects of clothianidin in real-world agricultural landscapes, with moderate, natural disease pressure.

[1] Department of Ecology, Swedish University of Agricultural Sciences, 750 07 Uppsala, Sweden. [2] General Zoology, Institute for Biology, Martin-Luther-University of Halle-Wittenberg, Hoher Weg 8, 06120 Halle (Saale), Germany. [3] Department of Computational Landscape Ecology, Helmholtz Centre for Environmental Research-UFZ Leipzig, ESCALATE, Permoserstrasse 15, 04318 Leipzig, Germany. [4] INRA, UE 1255 APIS, Le Magneraud, 17700 Surgères, France. [5] Centre d'Etudes Biologiques de Chizé, UMR 7372, CNRS & Université de La Rochelle, 79360 Villiers-en-Bois, France. [6] Department of Aquatic Sciences and Assessment, Swedish University of Agricultural Sciences, 750 07 Uppsala, Sweden. [7] Centre for Chemical Pesticides, Swedish University of Agricultural Sciences, 750 07 Uppsala, Sweden. [8] Apicultural State Institute, University of Hohenheim, August-von-Hartmannstrasse 13, 70599 Stuttgart, Germany. [9] Swedish Board of Agriculture, 511 82 Jönköping, Sweden. [10] Department of Biology, Lund University, 223 62 Lund, Sweden. [11] Centre for Environmental and Climate Research, Lund University, 223 62 Lund, Sweden. [12] Department of Entomology and Nematology, University of California, Davis, Davis, CA 95616, USA. These authors contributed equally to the study: Julia Osterman, Dimitry Wintermantel. Correspondence and requests for materials should be addressed to J. Osterman (email: jul.osterman@googlemail.com) or to J.R. de Miranda (email: joachim.de.miranda@slu.se)

Pollinating insects, mainly bees, provide an important eco-system service through maintaining wild plant biodiversity and contributing to global food security as well as beekeeper and farmer livelihoods[1,2]. Recent global declines in wild bee abundance[3,4] and diversity[5,6] are threatening plant biodiversity as well as the production of pollinator-dependent crops[2]. The European honeybee (Apis mellifera) is the most important managed pollinator and contributes to crop pollination worldwide[1,2]. However, high honeybee colony mortality rates have been reported[2] with disease being the main driver of honeybee colony losses[7,8]. Honeybees are plagued with a bouquet of pathogens and parasites, but the ectoparasitic mite, Varroa destructor, together with the viruses it vectors[9] are clearly the most devastating. Varroa mites require regular monitoring and mite control treatments by beekeepers to mitigate colony losses[10].

Another threat to honeybees is the chronic exposure to pesticides used in agriculture[11]. Neurotoxic neonicotinoids are a class of insecticides that are used globally[12] and are usually applied to arable agricultural crops as a seed dressing or a spray[13]. The active compounds are systemic and can be found in all parts of the plant including pollen and nectar[14], which is a route of exposure to foraging bees[15]. Sublethal doses of neonicotinoids have been found to affect honeybee foraging behaviour and success[16], impact memory and learning abilities of honeybees[17] and inhibit the development of brood, adults and queens[17,18].

Elevated winter colony mortality is probably best explained by the combination of several of these stressors[2,8]. There is laboratory evidence that neonicotinoids have additive or synergistic effects on honeybee longevity and immunocompetence when combined with pathogen or parasite pressure[19–25]. At sublethal doses, neonicotinoids can enhance the harmful effects of honeybee pathogens on larvae and adults bees, especially at high doses and infection levels[23], and the combination of neonicotinoid exposure and pathogen pressure is associated with higher individual honeybee mortality rates[20]. Furthermore, neonicotinoids can negatively affect individual immune competence, leading to greater susceptibility to opportunistic pathogens and parasites[11,19]. The alteration of the honeybee microbiota might contribute to the deterioration of honeybee health, but this is as yet understudied[26,27]. There is evidence that the most prominent gut symbionts Gilliamella apicola and Snodgrassella alvi play an important role in honeybee nutrition and pathogen defence[28,29]. The ability to combat infections might be reduced through a negative effect of neonicotinoids on the honeybee microbial community[30].

Since December 2013, a European Union (EU) moratorium has banned the use of three common neonicotinoids (clothianidin, imidacloprid and thiamethoxam), in bee-attractive crops[31]. This moratorium was widely debated, since it had mostly been based on laboratory rather than field studies[32]. As further evidence of adverse effects of neonicotinoids has accrued, the EU recently decided to ban all outdoor uses of these three neonicotinoids from December 2018 onwards[33], although they can still be used in permanent green houses and for outdoor use in countries outside of the EU. The negative impact of neonicotinoids on wild bees under field exposure has been demonstrated by several studies[34–37]. However, the impact of field-level neonicotinoid exposure on honeybee health and survival has been less decisive, with the spectrum of conclusions ranging from negative impact[36,38,39] to no impact[34,40–42], sometimes even in the same study[36]. Also, clear evidence of synergistic interactions between real-world field-level exposure to pesticides and pathogens on colony performance is lacking, especially within the context of the general adaptability of honeybee colonies to environmental challenges.

In 2015, we published a well-replicated field study on effects of the neonicotinoid clothianidin on solitary bees, bumblebees and honeybees[34]. Here, we present new data on honeybees from a second consecutive year of this experiment, designed to uncover the cumulative effects of placement at clothianidin-treated fields over 2 years, and analyses of honeybee samples from both years for symbiotic gut bacteria, several pathogens and immune gene expression levels. While most parameters remained unaffected by the clothianidin treatment, the impact that we do find is mostly positive. Placement at clothianidin-treated fields was associated with increased brood production in the second year and with more adult bees across both years. In 2014, colonies at clothianidin-treated fields showed a lower decline rate in Gilliamella apicola abundance over the oilseed rape bloom than control colonies and clothianidin treatment was negatively associated with the abundance of Black queen cell virus in 2014 and Aphid lethal paralysis virus (both years). However, these effects are minor compared to the extensive natural seasonal fluctuations in colony-level parameters and pathogen abundance, suggesting that relatively healthy honeybee colonies have sufficient colony-level social and demographic plasticity to compensate for possible individual bee-level impacts of neonicotinoid exposure.

## Results

**Study design.** Ninety-six honeybee colonies were placed at 16 spring-sown oilseed rape fields (six colonies per field) in southern Sweden in 2013. Eight fields were sown with seeds coated with clothianidin and a fungicide, and eight control fields were sown with seeds coated only with the fungicide. with Clothianidin-treated and control fields were matched in pairs based on geographical proximity and land use in the surrounding landscape[34]. After overwintering at a common location, the colonies were in 2014 randomly re-assigned to six clothianidin-treated and four control spring-sown oilseed rape fields from the same study design and farmers as 2013, with four colonies per field (40 colonies in total), except for the following two main conditions: first, that colonies from clothianidin-treated fields in 2013 were again placed at clothianidin-treated fields in 2014, and second, that the treatment allocation on each farm was reversed relative to 2013, with different local fields used, due to crop rotation (Fig. 1).

We examined honeybee colony development (number of adults, amount of brood), honey production, swarming/supersedure and colony mortality as well as the prevalence and abundance of 13 RNA viruses (Supplementary Table 1), two pathogenic microsporidian gut parasites, two non-pathogenic gut bacteria and the ectoparasitic mite Varroa destructor and, in the first year, the expression of eight genes related to the honeybee innate immune response. We analysed each of these parameters individually in relation to the seed treatment of the fields where the colonies were placed (clothianidin-treated, control), in relation to the oilseed rape bloom (before, after) and in relation to the interaction between seed treatment and bloom, represented by a differential response during the oilseed rape bloom between colonies at treated and untreated fields, for both years combined and (where necessary) also separately for each year. The colonies were managed according to recommended beekeeping practices, including treatment against Varroa after the post-exposure assessment in 2013 (Supplementary Fig. 1).

**Verification of clothianidin exposure.** To verify use of the focal crop and clothianidin exposure in both years, we estimated the proportion of oilseed rape pollen collected by honeybees and quantified the clothianidin concentrations in bee tissue and in bee-collected pollen and nectar (Table 1). In 2013, the pollen collected

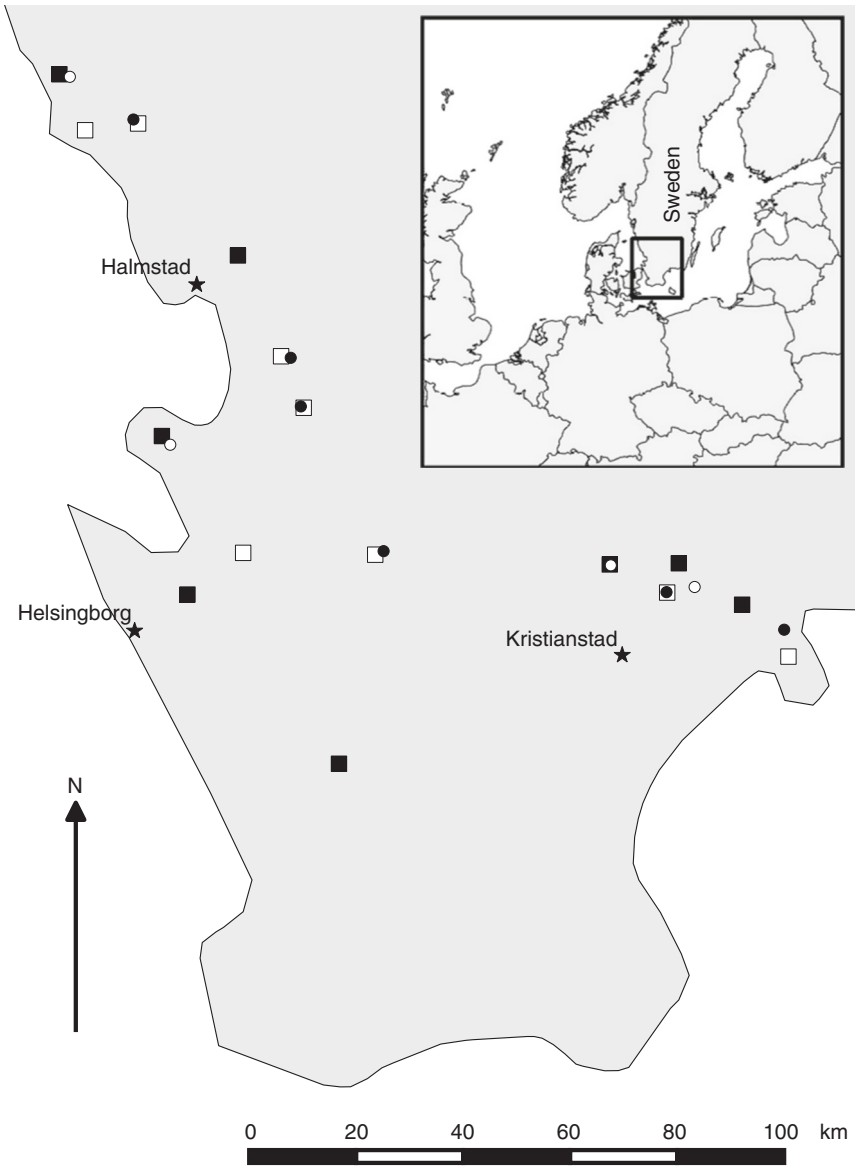

**Fig. 1** Study design with replicated landscapes in southern Sweden. In 2013 (squares), 16 oilseed rape fields and in 2014 (circles) 10 oilseed rape fields were either sown with seeds coated with clothianidin and a fungicide (black) or only a fungicide (control fields, white). Stars represent locations of the three weather stations in Supplementary Table 6. The map is based on the World Borders Database (downloadable here: http://thematicmapping.org/downloads/world_borders.php)

during the mid-oilseed rape bloom consisted on average of 53% and 63% oilseed rape pollen at control and clothianidin-treated fields respectively[34]. In 2014, the pollen collected contained substantially higher proportions of oilseed rape than in 2013, both at clothianidin-treated fields (93%, 95% confidence interval (CI): 88–97%) and at control fields (91%, CI: 83–95%), with no significant difference between control and clothianidin-treated fields (generalized linear model (GLM), $F_{1,7} = 0.98$, $P = 0.36$). In 2014, the clothianidin levels detected in pollen ($6.1 \pm 2.0$ ng g$^{-1}$; mean $\pm$ s.e. m.; $n = 6$) and nectar ($4.9 \pm 1.1$ ng ml$^{-1}$; $n = 6$) from honeybees foraging in the clothianidin-treated fields and honeybees ($1.1 \pm 0.20$ ng g$^{-1}$; $n = 6$) from colonies by the same farms were approximately half of the levels detected in 2013[34] (pollen $13.9 \pm 1.8$ ng g$^{-1}$; nectar $10.3 \pm 1.3$ ng ml$^{-1}$; bee tissue $2.4 \pm 0.5$ ng g$^{-1}$; $n = 8$ for all matrices). In both years, the clothianidin concentrations at the control fields were mostly below the limit of detection (LOD), although we still we detected clothianidin residues in some control samples in

2013 (honeybee tissue and nectar). However, exposure to clothianidin differed greatly between clothianidin-treated and control fields (Table 1). Four other neonicotinoids were also detected, both in clothianidin-treated fields and in control fields, in both years of the experiment (Supplementary Table 2). In 2013, when we examined individual honeybees and their nectar loads, there was large variation between three different clothianidin-treated fields in the residue levels in the honeybees (analysis of variance (ANOVA), $F_{2,33} = 15.84$, $P < 0.001$) and in the nectar collected from their honey stomachs (ANOVA, $F_{2,33} = 4.68$, $P = 0.016$; Supplementary Fig. 2), with a correlation between the clothianidin concentration in bee tissue and nectar from each honeybee (multiple linear regression, $R^2 = 0.866$, $F_{1,32} = 90.03$, $P < 0.001$).

**Colony development and honey production.** Placement at clothianidin-treated fields was associated with an increase in colony strength (amount of brood and number of adult bees)

**Table 1 Clothianidin residues in honeybees, pollen and nectar during 2013–2014**

| | Control fields[a] | | Treated fields[b] | | | | |
|---|---|---|---|---|---|---|---|
| | Range[c] | Mean ± s.e.m.[c] | Range[c] | Mean ± s.e.m.[c] | Z[d] | p[d] | n (T, C)[e] |
| **2013** | | | | | | | |
| Honeybees | <LOD-0.89 | 0.13 ± 0.11 | 0.35–4.90 | 2.4 ± 0.50 | −3.29 | **0.001** | 8/8 |
| Pollen | <LOD | <LOD | 6.60–23.00 | 13.9 ± 1.80 | −3.16 | **0.002** | 6*/8 |
| Nectar | <LOD-0.61 | 0.11 ± 0.08 | 6.70–16.00 | 10.30 ± 1.30 | −3.40 | **<0.001** | 8/8 |
| **2014** | | | | | | | |
| Honeybees | <LOD | <LOD | 0.15–1.50 | 1.10 ± 0.20 | −2.53 | **0.011** | 4/6 |
| Pollen | <LOD | <LOD | 2.40–16.00 | 6.10 ± 2.00 | −2.53 | **0.011** | 4/6 |
| Nectar | <LOD | <LOD | 2.60–9.80 | 4.90 ± 1.10 | −2.53 | **0.011** | 4/6 |

[a]Sample size: 2013, $n = 8$; 2014, $n = 4$. No honeybees with pollen could be found at two of the control fields
[b]Sample size: 2013, $n = 8$; 2014, $n = 6$
[c]Concentrations of clothianidin in honeybees (ng g$^{-1}$) and in pollen (ng g$^{-1}$) and nectar (ng ml$^{-1}$) sampled from foraging honeybees in clothianidin-treated oilseed rape fields (T) and control fields (C) during 2013–2014
[d]Wilcoxon test for differences between treatments. P values < 0.05 are highlighted in bold
[e]Sample size, T clothianidin-treated oilseed rape fields, C control fields, LOD limit of detection

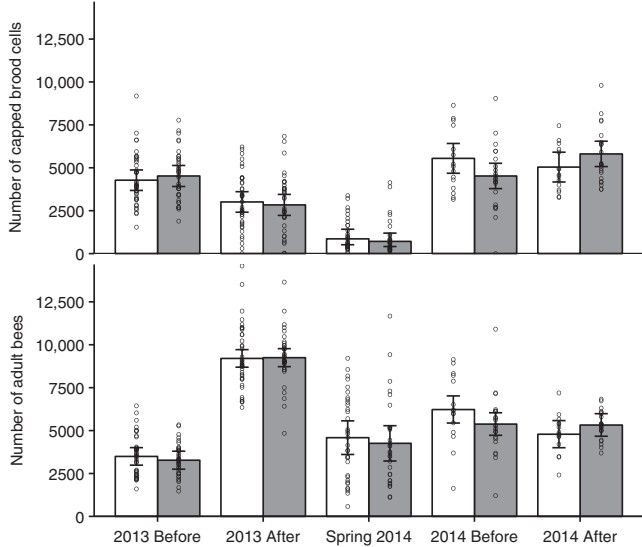

**Fig. 2** Honeybee colony development. The mean (±95% confidence limits) number of adult bees and the number of capped brood cells per colony in relation to seed treatment (control = white bars; clothianidin-treated = grey bars) in oilseed rape fields before and after the oilseed rape bloom in two years (2013 and 2014) and at the 2014 spring inspection; $n = 8$ fields per treatment in 2013 and $n = 4$ control fields and 6 clothianidin-treated fields in 2014. Circles indicate measured values per colony

particularly during the second year of the experiment, although this latter result should be interpreted with caution, since the colonies at clothianidin-treated fields were left slightly under-strength relative to the control colonies by the beekeeper interventions in June 2014, and may have overcompensated as a result. The colonies also gained strength much more quickly during the 2013 bloom than during the 2014 bloom, for all colonies irrespective of field treatment (Fig. 2, Table 2). Since the analysis of the number of capped brood cells (amount of brood) showed an interaction between seed treatment, bloom and year, this parameter was analysed separately for each year (Table 2). In 2013, colonies had less brood after the oilseed rape bloom than before, but this seasonal change in brood amount was not affected by the clothianidin treatment (Table 2), nor was there any delayed effect of the clothianidin treatment in 2013 on brood amounts after winter, during the first assessment in spring 2014 (Fig. 2, Table 2). During the 2014 oilseed rape bloom, the amount of brood in the

control colonies decreased slightly, while in the exposed colonies it increased, thus explaining the interaction between bloom and seed treatment and the positive effect of the clothianidin treatment on brood amount (Fig. 2). Clothianidin treatment was positively related to the number of adults, as clothianidin-exposed colonies had on average more adults than control colonies after the oilseed rape bloom despite starting with fewer adults (Fig. 2). This pattern was more pronounced in 2014, although the seed treatment × bloom × year interaction was not statistically significant (Table 2). In contrast to brood amount, the number of adult bees more than doubled during the oilseed rape bloom in 2013, but showed a weak decline in 2014 (Fig. 2). Similar to the brood amounts, there was no delayed effect of the 2013 clothianidin treatment on the number of adult bees during the first spring 2014 colony assessments (Fig. 2, Table 2). Honey production differed between years, with more honey produced in the first year than in the second year, but this was not affected by clothianidin treatment in either year (Fig. 3, Table 2).

**Swarming/supersedure and colony mortality.** During the 2013 oilseed rape bloom, 32 colonies (16 each at clothianidin-treated and control fields) prepared to replace their queen, with 27 eventually doing so: 15 at clothianidin-treated fields and 12 at control fields ($\chi^2_1 = 0.07$, $P = 0.795$), and about twice as often through swarming as through supersedure (Supplementary Table 3). The swarming/supersedure rates were higher for 2-year-old queens (50%) than for the 1-year-old queens (25%). Only 5 colonies prepared to swarm/supersede during the 2014 season (3 at treated and 2 at untreated fields), with 2 colonies eventually swarming (1 each at treated and untreated fields) all from 2-year-old queens, since only colonies with 1-year-old queens throughout 2013 were retained for the 2014 experiments. Swarming/supersedure in 2013 was a major factor for subsequent colony mortality (Supplementary Table 3), while placement at clothianidin-treated fields was not a factor ($\chi^2_1 = 0.10$, $P = 0.749$). Between May 2013 and April 2014, 28 experimental colonies died: 15 from clothianidin-treated fields and 13 from control fields. One colony was removed from the 2013 experiment after losing its queen during transport to its (clothianidin-treated) field. Of the remaining 27 fatalities, 22 colonies had swarmed or superseded during 2013, giving a mortality rate of 81% (22/27) for swarmed/superseded colonies, compared to 7% (5/68) for colonies that did not swarm/supersede. Sixteen colonies (all of which had swarmed/superseded) were removed already in September 2013 for being too small to overwinter, while the remaining 11 colonies (6 of which had swarmed/superseded) died during the winter 2013–2014.

**Table 2 Colony development and honey production in relation to fixed effects**

| Model | Model type[a] | Effect measure[a] | Estimates[b] | Degrees of freedom | F[b] | P[b] |
|---|---|---|---|---|---|---|
| **Colony development**[c] | | | | | | |
| Number of capped brood cells | Full model[d] | Intercept | 4445.953 | | | |
| | | Seed-treatment | −23.591 | 1, 16 | 0.04 | 0.837 |
| | | Bloom | −271.905 | 1, 110 | 6.06 | **0.015** |
| | | Year | 783.702 | 1, 19 | 42.12 | **<0.001** |
| | | Seed-treatment x Bloom | 171.907 | 1, 110 | 2.42 | 0.122 |
| | | Seed-treatment x Year | −41.545 | 1, 17 | 0.13 | 0.719 |
| | | Bloom x Year | 466.244 | 1, 110 | 17.83 | **<0.001** |
| | | Seed-treatment x Bloom x Year | 276.231 | 1, 110 | 6.26 | **0.014** |
| | 2013 | Intercept | 3656.020 | | | |
| | | Seed-treatment | 20.252 | 1, 7 | 0.22 | 0.872 |
| | | Bloom | −738.149 | 1, 75 | 46.74 | **<0.001** |
| | | Seed-treatment x Bloom | −104.324 | 1, 75 | 0.10 | 0.380 |
| | 2014 | Intercept | 5098.070 | | | |
| | | Seed-treatment | −60.730 | 1, 4 | 0.08 | 0.790 |
| | | Bloom | 194.339 | 1, 35 | 0.94 | 0.338 |
| | | Seed-treatment x Bloom | 448.139 | 1, 35 | 5.02 | **0.032** |
| Number of adult bees | Full model[d] | Intercept | 5866 | | | |
| | | Seed-treatment | −61 | 1, 17 | 0.24 | 0.629 |
| | | Bloom | 1274 | 1, 110 | 161.40 | **<0.001** |
| | | Year | −439 | 1, 20 | 11.26 | **0.003** |
| | | Seed-treatment x Bloom | 206 | 1, 110 | 4.22 | **0.042** |
| | | Seed-treatment x Year | −17 | 1, 17 | 0.02 | 0.893 |
| | | Bloom x Year | −1647 | 1, 110 | 269.79 | **<0.001** |
| | | Seed-treatment x Bloom x Year | −140 | 1, 110 | 1.94 | 0.166 |
| **Spring assessment** | | | | | | |
| Number of capped brood cells | 2014 | Intercept | 6.684 | | | |
| | | Seed-treatment | 0.029 | 1 | $X^2 = 0.82$ | 0.821 |
| Number of adult bees | 2014 | Intercept | 4421.691 | | | |
| | | Seed-treatment | −164.356 | 1, 7 | 0.19 | 0.674 |
| **Honey production** | Full model[d] | Intercept | 9.425 | | | |
| | | Seed-treatment | −0.095 | 1, 16 | 0.02 | 0.880 |
| | | Year | −1.378 | 1, 18 | 4.47 | 0.049 |
| | | Seed-treatment x Year | 0.009 | 1, 16 | <0.01 | 0.989 |

[a] The number of capped brood cells and the number of adult bees in relation to clothianidin seed-treatment, bloom (before or after the oilseed rape flowering period) and year (2013 and 2014) using linear mixed effect models
[b] Main effects/interactions were estimated using sum-to-zero contrasts and the deviation of the second level (after, clothianidin, 2014) of each factor (Bloom: before/after, Seed-treatment: control/clothianidin and Year: 2013/2014) from to the grand mean (intercept) is presented. P values < 0.05 are highlighted in bold
[c] Measurements were taken during pre- and post- exposure assessment
[d] Full model: $n = 14$, clothianidin-treated; $n = 12$, control fields; repeated measurements. 2013: $n = 8$ per treatment. 2014: $n = 6$, clothianidin-treated; $n = 4$ control fields

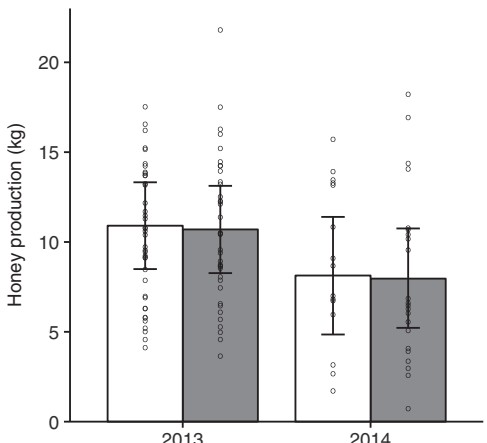

**Fig. 3** Honey production. Mean (±95% confidence limits) honey production measured in kilogram in relation to seed treatment (control = white bars; clothianidin-treated = grey bars) during the oilseed rape bloom in two years (2013 and 2014); $n = 8$ fields per treatment in 2013 and $n = 4$ control fields and 6 clothianidin-treated fields in 2014. Circles indicate measured values per colony

**Pathogen, parasite and microbe prevalence**. We detected both symbiotic gut bacteria (*Gilliamella apicola* and *Snodgrassella alvi*), both *Nosema* species (*Nosema apis* and *Nosema ceranae*), the *Varroa* mite and all 13 viruses screened for (Fig. 4, Supplementary Table 1). The overall prevalence of pathogens and parasites was higher in 2014 than in 2013, with often large differences between pre- and post-exposure assessments for individual microbes, which are furthermore not always consistent between years. However, placement at clothianidin-treated fields had generally no effect on the change in prevalence during the oilseed rape bloom for most pathogens and parasites (Supplementary Table 4, Fig. 4).

Several microbes (*G. apicola*, *S. alvi*, black queen cell virus (BQCV), Lake Sinai virus 1 (LSV-1), Sacbrood virus (SBV) and *N. apis* (in 2014)) were present in nearly all colonies, clothianidin-treated and control, throughout the oilseed rape bloom during both years, presenting insufficient variation for insightful analysis into the relative importance of various factors on their prevalence (Supplementary Table 1). Others (Big Sioux river virus (BSRV) and LSV-2) were effectively absent throughout the experiment, in all colonies, and were therefore similarly uninformative. Many viruses (acute bee paralysis virus (ABPV), chronic bee paralysis virus (CBPV), Israeli acute paralysis virus (IAPV), Kashmir bee virus (KBV) and slow bee paralysis virus (SBPV)) were largely

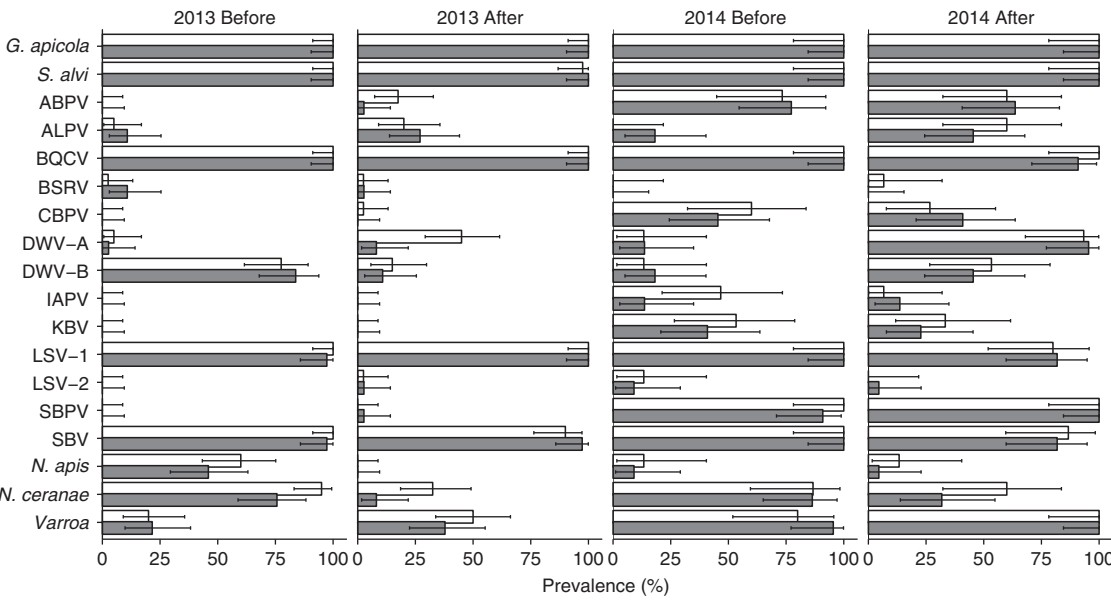

**Fig. 4** Microorganism/parasite prevalence. Percentage (±95% confidence limits) of honeybee colonies infected with the gut bacteria *Gilliamella apicola* and *Snodgrassella alvi*, the viruses acute bee paralysis virus (ABPV), aphid lethal paralysis virus (ALPV), black queen cell virus (BQCV), big Sioux River virus (BSRV), chronic bee paralysis virus (CBPV), deformed wing virus type-A (DWV-A), deformed wing virus type-B (DWV-B), Israeli acute paralysis virus (IAPV), Kashmir bee virus (KBV), Lake Sinai virus type-1 (LSV-1), Lake Sinai virus type-2 (LSV-2), slow bee paralysis virus (SBPV), Sacbrood virus (SBV), the microsporidians *Nosema apis, Nosema ceranae* or infested with the *Varroa* mite (*Varroa destructor*) in relation to treatment (control = white bars; clothianidin-treated = grey bars) in oilseed rape fields (*n* = 8 fields per treatment in 2013; *n* = 4 control and *n* = 6 treated fields in 2014) before and after the oilseed rape bloom in two years (2013 and 2014)

absent in 2013 but moderately/highly prevalent in 2014, showing major differences between years. The viruses showed different patterns of prevalence over the oilseed rape bloom season, but their seasonal prevalence could not be linked to clothianidin treatment (Fig. 4, Supplementary Table 4). Aphid lethal paralysis virus (ALPV) increased over the oilseed rape bloom seasons across both years, while changes in prevalence could not be statistically confirmed for ABPV, CBPV, KBV or IAPV (Fig. 4, Supplementary Table 4). Deformed wing virus-A (DWV-A) prevalence increased during the summers, which is a well-established seasonal pattern[9,43] (Fig. 4, Supplementary Table 4), probably attributable to its vector *Varroa*[9]. DWV-B prevalence decreased during the 2013 bloom, but increased during the 2014 bloom (Fig. 4, Supplementary Table 4).

Both *Nosema* species decreased in prevalence during summer, which corresponds to their known seasonal patterns[43], but this decrease was more drastic in 2013 than in 2014. The change in *Nosema* prevalence over the blooming season was not influenced by the placement at clothianidin-treated fields (Supplementary Table 4). In 2013, *N. ceranae* was less prevalent in colonies placed at clothianidin-treated fields than at control fields both before and after exposure, as is shown by the significant seed treatment effect (Supplementary Table 4).

**Parasite, pathogen and microbe abundance.** For most pathogens, microbes and parasites, we also tested the effect of placement at clothianidin-treated fields on their abundance in the colonies where they were detected (Fig. 4 (see above), Supplementary Table 1). Where possible, we included both years, in a full model, but we excluded years with a sample size ≤10 colonies (Supplementary Table 1). Placement at clothianidin-treated fields generally had relatively little detectable effect on pathogen, parasite or microbe abundance but was in 2014 positively related to *G. apicola* abundance, and negatively related to BQCV abundance as well as negatively to ALPV abundance across both years.

*Gilliamella apicola* abundance in adult bees was partly explained by an interaction between seed treatment, bloom and year, and therefore the dataset was analysed separately for each year (Supplementary Data 1). During the oilseed rape bloom in 2013, *G. apicola* abundance declined, irrespective of the placement at clothianidin-treated fields (Supplementary Data 1, Fig. 5). In 2014, there was an interaction between bloom and seed treatment, with the *G. apicola* abundance declining more quickly in control colonies than in clothianidin-exposed colonies (Supplementary Data 1, Fig. 5). The abundance of *S. alvi* remained stable in 2013 but increased during the bloom in 2014, irrespective of the placement at clothianidin-treated fields (Supplementary Data 1, Fig. 5). In contrast to *G. apicola*, ALPV abundance increased during the oilseed rape bloom and this increase was less pronounced in clothianidin-treated than in control colonies (Supplementary Data 1, Fig. 5). Similar to *G. apicola*, BQCV abundance was partly explained by an interaction between treatment, bloom and year, and therefore the dataset was analysed separately for each year (Supplementary Data 1). In 2013, BQCV abundance decreased over the oilseed rape bloom, irrespective of the placement at clothianidin-treated fields. The abundance of BQCV also decreased in 2014 clothianidin-exposed colonies but not in control colonies (Supplementary Data 1, Fig. 5). In 2014, IAPV abundance showed a similar non-significant seasonal pattern as its prevalence (Supplementary Data 1, Fig. 5). The abundances of ABPV, CBPV and KBV were not affected by the clothianidin treatment and also showed no seasonal patterns (Supplementary Data 1). In contrast, SBPV abundance increased during the oilseed rape bloom in 2014, but was otherwise not affected by the clothianidin treatment (Supplementary Data 1). Just as for prevalence, DWV-A abundance increased generally during the oilseed rape bloom but more strongly in 2014 than 2013, while the abundance of DWV-B showed contrasting seasonal patterns in the two years (Supplementary Data 1, Fig. 5). During the oilseed rape bloom in 2013, DWV-B abundance decreased, whereas in 2014 the

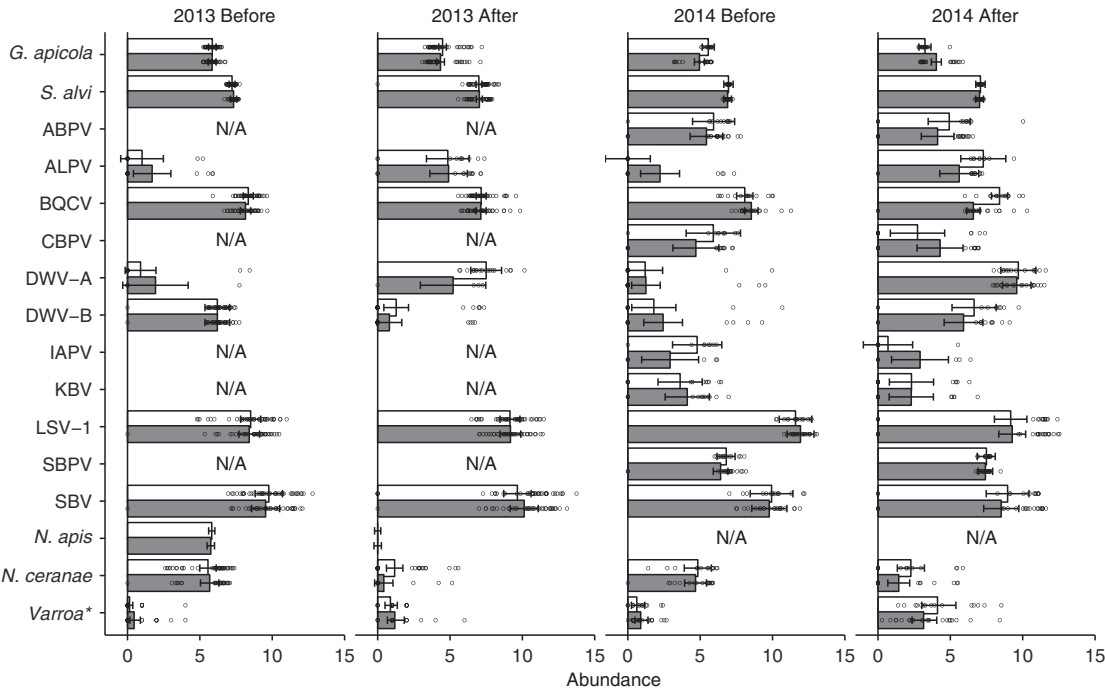

**Fig. 5** Microorganism/parasite abundance. The abundance (±95% confidence intervals) of frequently detected target organisms (*Gilliamella apicola*, *Snodgrassella alvi*, acute bee paralysis virus (ABPV), black queen cell virus (BQCV), deformed wing virus type-A (DWV-A), deformed wing virus type-B (DWV-B), Israeli acute paralysis virus (IAPV), Kashmir bee virus (KBV), Lake Sinai virus 1 (LSV-1), slow bee paralysis virus (SBPV), Sacbrood virus (SBV), *Nosema ceranae* and *Varroa destructor*) in honeybee colonies in relation to the treatment (control = white bars; clothianidin-treated = grey bars) in oilseed rape fields (n = 8 fields per treatment in 2013; n = 4 control and n = 6 treated fields in 2014) before and after the oilseed rape bloom in two years (2013 and 2014). *The abundance of V. destructor was represented by the number of mites per 100 bees, while the microorganisms were expressed as $\log_{10}$ [units] per bee[70]. Microorganism abundance was not analysed (N/A) if it was detected in less than 11 samples per year. Circles indicate measured values per colony

abundance increased (Supplementary Data 1, Fig. 5). However, neither DWV strain was affected by clothianidin treatment. The abundance of LSV-1 was not related to the clothianidin treatment but was explained by an interaction between bloom and year, with an increase in 2013 and a decrease in 2014 (Supplementary Data 1, Fig. 5). The abundance of SBV was relatively high and stable throughout the experiment (Supplementary Data 1, Fig. 5). The two *Nosema* species decreased during oilseed rape blooms, where we could statistically assess this (*N. ceranae* in both years and *N. apis* in 2013). In contrast, the adult bee *Varroa* infestation rate increased during oilseed rape blooms from, on average, around 0.3 to 1 mites per 100 bees in 2013 and from 0.8 to 3.6 mites per 100 bees in 2014. The abundances of *Varroa* or either species of *Nosema* were unaffected by placement at clothianidin-treated fields (Supplementary Data 1, Fig. 5).

**Immune gene expression.** The messenger RNA (mRNA) levels of several key honeybee immune genes involved in the honeybee molecular response to neonicotinoids[19,44,45] were tested in 2013 at the apiary level. The levels of the mRNAs of these genes relative to that of a standard internal reference gene were unaffected by the placement of colonies at clothianidin-treated sites, nor did they change over the oilseed rape bloom, except for *Apidaecin* and SPH51, which were more abundant after the oilseed rape bloom than before (Supplementary Table 5, Fig. 6).

## Discussion

In this extensive field experiment, we identified large fluctuations between and within years in honeybee colony development, (attempted) swarming/supersedure, colony mortality, microbial composition and *Varroa* infestation but no verified negative effects of placement at the clothianidin-treated fields on these parameters. Both these observations are keys to the interpretation of the results. It places the effects of neonicotinoid exposure on individual bees, as observed in laboratory[17] and (semi-)field studies[16,18] within the context of the high plasticity of the colony as a social unit in response to natural and anthropogenic environmental challenges. It therefore identifies sociality itself as a potent additional homoeostatic mechanism, available to social bees but not to solitary bees, for compensating the negative individual effects of neonicotinoid exposure, with or without additional pathogen/environmental pressures. Similarly, while neonicotinoid exposure can increase pathogen-induced mortality of individual bees, especially at high levels of infection[23], the forces that drive the natural volatility in pathogen and microbial prevalence and abundance at colony level appear stronger than the effects of clothianidin exposure, as shown by the large fluctuations during the bloom periods and between years, relative to the placement at clothianidin-treated fields.

One of the most potent colony-level responses to internal and external cues is a colony's decision to replace the queen, either through swarming or supersedure. This is a complex process, involving extensive sensory perception, communication and decision making among the worker bees[46]. Supersedure is often a consequence of (perceived) dissatisfaction with the queen's attributes[25,47] whereas swarming is also triggered by (perceived) lack of space and seasonal cues[46]. All these factors (queen attributes, sensory perception, communication, decision making) can be affected by neurological agents such as neonicotinoids[25,39,44,47–49]. We previously documented as part of this landscape study the severe effects of placement at clothianidin-treated fields on reproduction in *Bombus terrestris*

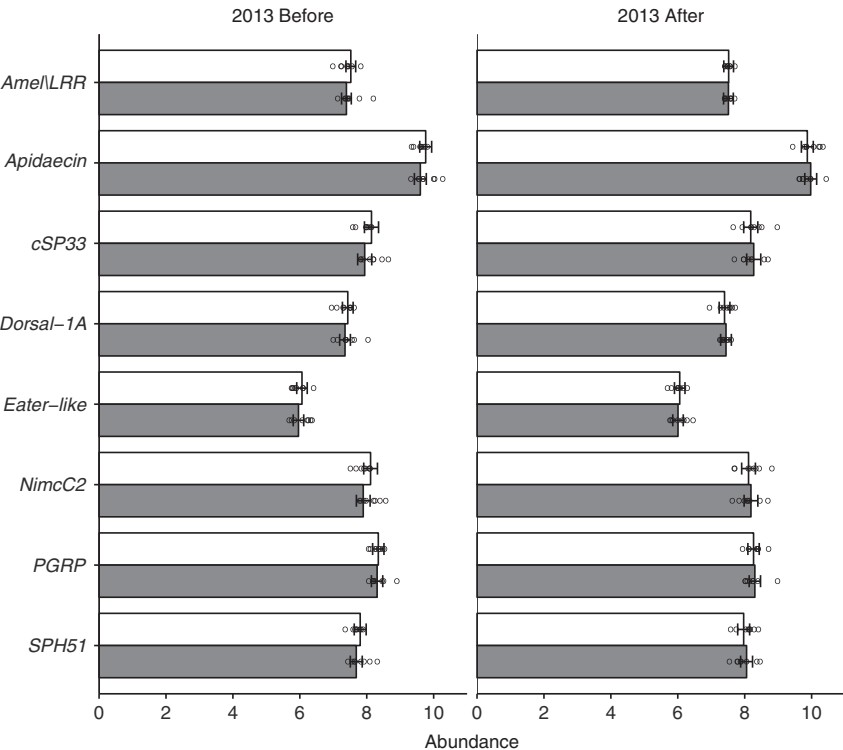

**Fig. 6** Immune gene expression at apiary level in 2013. The abundance (±95% confidence intervals) of expressed immune genes in honeybees at apiary level in 2013 in relation to the treatment (control = white bars; clothianidin-treated = grey bars) in oilseed rape fields ($n = 8$ fields per treatments) before and after their bloom. Immune gene expression data are presented as the $\log_{10}$ number of estimated mRNA copies per bee. Circles indicate measured values per colony

and *Osmia bicornis*[34], making its possible influence on queen acceptance in honeybee colonies particularly relevant. A large enough number of colonies swarmed/superseded during 2013 to establish that, in our experiment, the placement at clothianidin fields had minimal influence on the decision to swarm/supersede. This is supported by the low swarming/supersedure of the surviving colonies in 2014, under similar conditions. The high swarming/supersedure rate in 2013 is most likely due to the extensive splitting and forced re-queening when preparing the colonies, since it is not unusual for queens to be rejected under these circumstances. Also, the high colony mortality during 2013–2014 can be largely attributed to swarming/supersedure, rather than the placement at clothianidin-treated fields. Excluding the swarmed/superseded colonies, the 2013–2014 colony morality rate (7%) fits the average for the region and year (8%[50]), again with no difference between colonies at treated and untreated fields.

We also observed no delayed impact of placement at clothianidin-treated fields in 2013 on colony strength (brood amount or adult bees) after overwintering, during the spring 2014 assessments. We even observed a positive association between placement at clothianidin-treated fields and both adult bees (both years combined) and brood amount (2014). Placement at clothianidin-treated fields had no effect on the prevalence and no severe effect on the abundance of pathogens, parasites and beneficial gut symbionts and the few observed effects are typically positive for bee health. Clothianidin treatment was associated with a weaker increase in *Aphid lethal paralysis virus* abundance and in 2014, BQCV abundance declined only in colonies placed at clothianidin-treated fields during the oilseed rape bloom while not in control colonies. In addition, the abundance of the gut symbiont *G. apicola* declined less strongly in colonies at clothianidin-treated fields than in control colonies during the

oilseed rape bloom in 2014. The abundances of all other pathogens, the *Varroa* mite and the gut symbiont *Snodgrassella alvi* were not related to the clothianidin treatment. Jones et al.[30] demonstrated that although the gut microbial composition is influenced by the surrounding landscape, the relationship between environmental stressors, the gut symbiotic bacteria and its host is too complex and interactive for simple reductionist assessments. Two other field studies also found no effect of neonicotinoids on *Varroa* abundance[41,51], while a third found a positive association[52]. In addition, in-hive feeding with imidacloprid-spiked pollen increased *Varroa* abundance but only at a very high imidacloprid concentration[49].

Laboratory studies have shown that the expression of several immune genes can be influenced by pesticide exposure[19,53,54] including possible secondary effects on pathogen susceptibility[19,44,55–57]. However, a recent study by Collison et al.[58] found weak and inconsistent effects of neonicotinoids on transcriptional responses, and no harmful impact on bee health through suppressed immunocompetence could be detected. In our study the mRNA levels of several key honeybee immune genes were not affected by the placement at clothianidin-treated fields or, generally, by the oilseed rape bloom in 2013. mRNA is a short-lived intermediate in the communication between an organism's genetic resources and its physiological needs, whose induced response is measured in minutes or hours, as is generally the case in laboratory studies, rather than weeks or months, as was the case in our study. As Collison et al.[58] demonstrated, alterations in gene expressions changed over time often with a peak between 8 and 24 h after exposure, followed by a gradual decline within the next 6 days. At a longer time scale, a difference in mRNA levels would indicate a major biological shift in the constitutive expression rates of these genes, rather than the temporary induction most frequently investigated in laboratory

studies. Both types of changes are important and necessary features of an organism's molecular-adaptive response to its environment, but the longer term type of change is more impactful, since it represents a fundamental change in the molecular baselines and norms on which the organism functions. The more fleeting response may very well also have occurred here, in individual bees immediately after their exposure to clothianidin (i.e. forages encountering neonicotinoids in the fields, or house bees when handling contaminated pollen or nectar), but our samples lack the resolution, both in time and sampling unit (colony/apiary), to detect this. We sampled bees at the common overwintering site, after the oilseed rape bloom and possibly several days after being in direct contact with the neonicotinoid. Task-based division of labour may also have diffused the levels of exposure of different bees of the same colony, and gene expression can be socially regulated in honeybees[59].

In both years, honeybee-collected pollen and nectar contained substantially higher concentrations of clothianidin from colonies at treated fields than in control colonies, verifying the exposure scenarios of the treated and control field conditions in this study. Interestingly, the clothianidin concentrations at the treated fields in 2013 were twice as high as in 2014, despite almost identical seed coatings, study farms and analysis methods. Neonicotinoid degradation and leaching are related to temperature and moisture conditions in the soil, with longer half-life under cool and dry conditions[60]. The spring of 2013 had a lower average temperature, more days with frost and less precipitation than the spring of 2014 in our study region (Supplementary Table 6). We suspect that these differences in spring weather could have contributed to the variation in clothianidin residues in pollen and nectar between the two years. However, this does not imply a lower exposure of clothianidin to the honeybees in the second year compared to 2013, as they collected a higher proportion of oilseed rape pollen in 2014 than in 2013. The detection of clothianidin in honeybee-collected nectar and in honeybee tissue at control sites in 2013 demonstrates the difficulty of setting up control conditions in field experiments, since the wide flight range of honeybees means they can forage on other potentially treated fields. Still, this experiment adequately captured the exposure to clothianidin compared to control conditions, since we demonstrated large differences between the treated and control sites in measured clothianidin residues. Furthermore, the clothianidin concentrations in pollen from clothianidin-treated fields were higher in both years than what has been reported in other studies[36,40,41]. This is probably because the oilseed rape was sown in spring rather than autumn, as it was the case in the other studies. Moreover, we demonstrated considerable variation within and among treated fields in clothianidin concentrations of honeybee-collected nectar and bee tissue sampled from both individual bees and pooled bee samples from fields. These variations might be the result of uneven foraging alternatives in the various landscapes or differences in clothianidin concentrations in the plants. This indicates that the neonicotinoid exposure risk for bees may differ depending on the cultivar, the time of sowing, the geographical location and possibly also the weather conditions.

There have now been several more or less well-designed field studies that have failed to detect a major impact of field-level neonicotinoid exposure on honeybee colony development, performance and overwintering success[34,36,40,41,61]. At the same time several field-realistic studies have reported the impairment of individual and social bee life parameters due to chronic exposure to neonicotinoids[39,47–49]. The importance of the results presented here is that in fundamentally healthy colonies, like the ones studied here, the natural homoeostatic health mechanisms mediating the colony's response to its environment are robust enough to overcome these impairments, even in undersized colonies,

during two consecutive years of 1-month direct exposure, in the middle of the short Swedish bee foraging season. These conclusions are compatible with a recent French study, where individual free-flying honeybees placed near fields treated with thiamethoxam disappeared faster than bees at untreated fields, due to a higher mortality rate, but that the total number of adult bees and the honey production of the colonies remained unaffected[48] due to effective colony-level compensating mechanisms. We were able to detect relatively subtle differences, as our experiment had sufficient power to comply with the requirements set by the European Food Safety Authority (EFSA)[32] (effect size of <7% with a power of 80%) for the assessment of pesticide effects on colony size.

Our study provides insight into the interactions between two drivers of honeybee colony losses, pathogens and pesticides, by demonstrating that foraging on oilseed rape grown from clothianidin-coated seeds had no observable negative effects on honeybees at colony level on either constitutive immune gene expression, microbial composition (pathogens or symbiotic bacteria) or *Varroa* infestation under real-world field conditions. We acknowledge that detrimental effects could well have existed at individual bee level but were effectively compensated for by colony-level social regulatory mechanisms, supported by the robust general health of these colonies, and that less robust colonies may well have yielded different results. We also contrast the large natural plasticity of honeybee colony performance and microbial composition with the insignificant negative influence of the placement of colonies at clothianidin-treated fields on these parameters, to highlight the importance of sociality as an additional adaptive mechanism for managing environmental challenges even in undersized colonies, confirming the need for separate exposure study models for social and solitary bees. The contrasting results from different field studies on honeybee health show the importance of the context of the exposure and the study system, as well as the need for more extensive research in multiple biogeographical areas and crop systems. Additionally, the large within- and between-year differences in colony parameters demonstrate the importance of multiple sample points within a year and long-term studies of the cumulative effects of the placement of colonies at clothianidin-treated fields, including the general landscape context[3], which are currently lacking. Studying the effect of insecticides on honeybee colonies under field conditions is crucial to understand realistic effects of neonicotinoids for further policy decisions, in addition to laboratory experiments, which seem a more sensitive system. To make informative decisions on pesticide use in natural, agricultural and urban landscapes, there is a need for improved understanding of the context dependence of colony-level responses in social bees[36,62], as well as the as-yet largely understudied effects on wild and solitary bee species[35,36].

## Methods

**Study design**. In 2013, a total of 16 fields (8.9 ± 5.4 ha; mean ± s.d.) in southern Sweden, intended for spring-sown oilseed rape (*Brassica napus* L.) were paired according to geographical proximity (but separated by >4 km) and land use (Fig. 1, see above). The surrounding landscape was inspected for the absence of flowering crops. However in 2013, two fields remained in the study even though another oilseed rape field was present nearby, so as to retain as many farm-pairs as possible[34]. In each farm pair, one field was randomly assigned to be sown with clothianidin-treated oilseed rape seeds, while the other field was sown with seeds not treated with clothianidin (treated: 8; control: 8). The same paired farms were used in 2014 but with the treatments reversed, i.e., locations with treated fields in 2013 had control fields in 2014 and vice versa (treated: 6; control: 4). Due to crop rotation, different fields within each farm were used in 2013 and 2014 (Fig. 1, see above). In order to create Fig. 1, we downloaded a map from the World Borders Database (downloadable here: http://thematicmapping.org/downloads/world_borders.php).

Information on surrounding landscape variables for the different farms in 2014 is presented in Supplementary Table 7. In 2014, half of the focal fields had

additional spring-sown oilseed rape (1–13 ha) within a 2 km radius. The clothianidin-treated seeds for the focal fields were coated with Elado, a trademarked blend of two active ingredients: clothianidin (400 g l$^{-1}$) and β-cyfluthrin (+80 g l$^{-1}$), chosen for this study because it was the predominant seed insecticide treatment in oilseed rape in Sweden and in other parts of Europe[63]. Clothianidin is taken up by the plant and systemically distributed to all its parts for protection against insects[64]. β-Cyfluthrin is not considered to be systemic and no residues were detected in samples collected in this study[34]. Both clothianidin-treated and control seeds were coated with the fungicide thiram in 2013.

The participating farmers were instructed not to use other neonicotinoids in the fields during the study, although other insecticide foliar sprays; primarily Plenum (pymetrozine), Avaunt/Steward (indoxacarb) and Mavrik (tau-fluvalinate; also used as varroacide in beekeeping) were used for pest control (Supplementary Table 8). However, Biscaya, tradename for a spray formulation containing the neonicotinoid thiacloprid, was applied to one control field in 2013, followed by a Mavrik spray 1 week later, and to one treated field in 2014, on both occasions at 0.3 L ha$^{-1}$. Thiacloprid has a considerably lower acute toxicity for bees than clothianidin[65] and only trace amounts of thiacloprid were detected in the pollen, nectar and bee samples in 2013 and none in 2014. While Rundlöf et al.[34] did not observe any change in results when fields where Biscaya was applied were excluded from the analyses, we detected some qualitative changes (Supplementary Table 9). These changes could be due to the higher thiacloprid residues detected in 2014, but may just as well not relate to Biscaya but rather to the difference between the Biscaya/Mavrik and the alternative insecticide spray combinations used[34] or be due to reduced statistical power.

**Honeybee colonies.** One hundred and sixteen honeybee colonies were prepared at the end of May 2013 by a professional beekeeper in single full-size Langstroth hives containing two combs with mainly sealed brood (with bees), two full honeycombs (with bees), one drawn out empty comb, five combs with wax foundation, bees shaken from two combs and either a 1-year-old (84 experimental colonies) or 2-year-old (12 experimental colonies plus 20 reserve colonies) queen of known descent to produce relatively small, equally sized (3418 ± 123 adult bees; mean ± s.e. m.; n = 96 colonies) colonies with plenty of room for growth that could become strong enough to survive the coming winter, but not outgrow their space during the summer. Six experimental colonies were placed along the field edge in each of the 16 oilseed rape fields (96 colonies in total) between 14 and 28 June 2013 at the onset of oilseed rape flowering (Supplementary Fig. 1). Queen lineage and age was matched between farm-pairs, but colonies were otherwise distributed randomly. Colonies were kept at a 60 ha organically managed winter oilseed rape field in full bloom before placement at the 16 experimental fields (Supplementary Fig. 1) to ensure pre-experiment colony growth was based as much as possible on pesticide-free foraging.

When the oilseed rape bloom in the experimental field had ceased, the colonies were moved between 2 and 31 July (Supplementary Fig. 1) to a common apiary to overwinter. On 10 August, the colonies were given a formic acid vapour treatment against Varroa mites, consisting of 20 ml 60% formic acid soaked into a flat household sponge placed under the inner cover on top of the frames. The colonies were fed a total of 20 kg of sugar per colony in the form of a 55–60% v/v sucrose solution, provided in a feeding box across three occasions during August–September 2013. An additional light Varroa treatment was carried out on 4 December by sprinkling 30 ml 2.6% oxalic acid in 60% sucrose in between the frames, directly onto the bee cluster. In spring 2014, colonies were moved to an organically managed oilseed rape field before placement at the 10 spring-sown oilseed rape fields (Supplementary Fig. 1). Colonies were considered for inclusion in the 2014 part of the study if they had a 2-year-old, egg-laying queen in April 2014 (excluding colonies that died, re-queened or had 3-year-old queens in April 2014) and had not swarmed by the beginning of June 2014. These restrictions, in addition to the requirement that colonies would be exposed/unexposed to clothianidin for both years of the experiment, meant that in 2014 only four colonies could be allocated to each field. Colonies placed by treated fields in 2013 were again placed by treated fields in 2014, so as to assess the cumulative effects of multi-year clothianidin exposure, but were otherwise re-randomized prior to placement to minimize unintended biases. Even so, two control colonies from 2013 had to be placed by a clothianidin-treated field in 2014, due to insufficient qualifying exposed colonies for the six clothianidin-treated fields. Enough colonies were available for the four control fields. The strength of colonies was equalized as described for 2013, but only within each treatment group. The colonies were reduced and equalized a second time (8 June 2014), after some of them grew too large and attempted to swarm (Supplementary Fig. 1). Each reduced colony included 1 full honey comb (with bees), 3 combs with mainly sealed brood (with bees), and the original queen from 2013 and 6 combs with wax foundation. Colonies were moved to the spring oilseed rape fields between 16 and 25 June 2014 and brought back to the common overwintering site between 14 and 22 July 2014 (Supplementary Fig. 1).

**Residue analyses.** To confirm clothianidin exposure, 24 adult honeybees per field caught at the hive entrance, pollen pellets collected from 5 honeybees foraging in the oilseed rape fields and nectar removed from the stomachs of 5 nectar-foraging

honeybees in the oilseed rape fields were analysed from each site for clothianidin residues. Pollen (>25 ml) was collected using pollen traps, which were installed for 1 day on three colonies per site and analysed for plant species origin. Samples were handled and analysed as in Rundlöf et al.[34] and collected during the peak bloom assessments in both years (Supplementary Fig. 1), with the concentrations of clothianidin and four other neonicotinoids used in Sweden (Supplementary Table 2) quantified using liquid chromatography coupled with tandem mass spectrometry (LC-MS/MS) and pollen identified to oilseed rape type using light microscopy and a pollen reference library (see Supplementary Table 2 for limits of detection and quantification). For further analyses of the variation in neonicotinoid exposure of honeybees in different sites, we collected 12 honeybees per site from the hive entrance. This sampling was done at three clothianidin-treated sites in 2013. Nectar was extracted from the honey stomach of the collected bees. The concentrations of clothianidin was thereafter quantified in both nectar and bee tissue for each bee individual. More details on the sample treatment for different matrices, LC-MS/MS method and quality controls are given in Supplementary Methods.

**Colony development, re-queening and honey production.** Honeybee colony development was assessed by the same trained observer and one assistant. The presence of a laying queen was established, as well as the presence of queen cells. If a re-queening event was accompanied by a large loss of adult bees, it was deemed to have swarmed. If no loss of adult bees was observed, the colony was deemed to have re-queened through supersedure. Colony honey production and development was determined by weighing the colonies and by assessing colony strength using the Liebefeld method[66], as the total number of adult bees and the area of capped brood over all frames. The number of adult bees was estimated by counting honeybees on both sides of the 10 frames. The number of capped brood cells (amount of brood) was determined by multiplying the proportion of closed brood coverage by 2700, which is the number of cells on one side of the frames used. The colonies were weighed during pre-exposure and post-exposure assessments (using a Mettler Toledo bench scale able to weigh up to 32 kg with 1 g precision), to estimate honey production. Full honey frames were replaced by empty frames during the oilseed rape bloom, to allow the colony to grow and reduce swarming. Both the full and the empty frames were weighed, for inclusion in the calculating of honey production. During post-exposure assessment, as many honey frames as possible were removed (max 10% of the area covered with covered brood) to simulate the beekeepers honey harvest. Pre-exposure assessments were done at the organically managed winter-sown oilseed rape field on 6–17 June 2013 and 9–11 June 2014, and post-exposure assessments at the common overwintering apiary on 29 July to 9 August 2013 and 28–31 July 2014 (Supplementary Fig. 1). Furthermore, a spring colony strength assessment was performed in April 2014 by estimating the total number of adult bees and the number of capped brood (Supplementary Fig. 1). The colony assessor and assistants were blinded during data collection with respect to the treatment regimen of the fields.

**Pathogen and parasite sample collection and processing.** Samples of around 100 adult honeybees were taken from each colony during pre- and post-exposure assessments in the clothianidin-treated and control experimental oilseed rape fields in both 2013 and 2014 (Supplementary Fig. 1). Bees were taken from the outer comb of each colony and consisted therefore of a mixture of house bees and forager bees[67]. All bee samples were stored at −20 °C until the laboratory work was performed. The V. destructor infestation rates for each colony were determined by washing the adult bee samples with soapy water to dislodge and count the mites[68]. The abdomens of 60 adult honeybees per colony (for individual colony analyses) or per apiary (for the 2013 pooled colony analyses, 10 bees per colony) were removed and placed in a polyethylene bag with an inner mesh (BioReba). The abdomens were ground in the bag using a pestle and 30 ml of nuclease-free (Milli-Q) water (0.5 ml per bee) was mixed thoroughly with the sample to create a homogenous suspension. Several 1 ml aliquots of this suspension were removed and frozen immediately at −80 °C for DNA and RNA extraction and as future reference material.

**Parasites, pathogens, symbiotic microbes and immune genes.** The collected bee samples were assessed for a variety of pathogenic and non-pathogenic parasites and microbes in order to study the impact placement of colonies at clothianidin-treated fields on their prevalence and abundance. The organisms included the ubiquitous ectoparasite Varroa destructor, 13 viruses: acute bee paralysis virus (ABPV), aphid lethal paralysis virus (ALPV), Big Sioux River virus (BSRV), black queen cell virus (BQCV), chronic bee paralysis virus (CBPV), deformed wing virus type-A (DWV-A), deformed wing virus type-B (DWV-B), Israeli acute paralysis virus (IAPV), Kashmir bee virus (KBV), Lake Sinai virus strain 1 (LSV-1) and strain 2 (LSV-2), Sacbrood virus (SBV), slow bee paralysis virus (SBPV); two common honeybee microsporidian gut parasites (Nosema apis and Nosema ceranae) and two symbiotic gut bacteria (Gammaproteobacterium: Gilliamella apicola and Betaproteobacterium: Snodgrassella alvi). For the 2013 samples we also analysed at apiary level the mRNA levels of eight honeybee genes (Amel/LRR, Apidaecin, cSP33, Dorsal-1A, Eater-like, NimC2, PGRP-S2 and SPH51) whose

expression had previously been linked to pesticide, pathogen and/or parasite exposure[19,44] and (social) immunity in honeybees[45].

**Nucleic acid extraction**. DNA was extracted from the bee homogenates using the protocol for extracting DNA from *Nosema* spores[69], which is sufficiently robust to also extract DNA from bacteria and other microorganisms. A total of 500 µl primary bee homogenate was centrifuged for 5 min in a microfuge at 13,000 rpm. The pellet was repeatedly frozen-thawed with liquid nitrogen and ground with a sterile Teflon micropestle until pulverized. The pulverized pellet was resuspended in 400 µl Qiagen Plant tissues DNeasy AP1 lysis buffer containing 4 µl RNAse-A (10 mg ml⁻¹) and incubated and shaken for 10 min at 65 °C, after which 130 µl P3 neutralization buffer (3.0 M potassium acetate pH 5.5) was added, followed by 5 min of incubation on ice and centrifugation for 5 min at 14,000 rpm to remove the lysis debris. DNA was purified from 500 µl of the supernatant by the Qiagen automated Qiacube extraction robot, following the plant DNeasy protocol and eluting the DNA into 100 µl nuclease-free water. RNA was extracted by the Qiacube robot directly from 100 µl primary honeybee homogenate using the Qiagen Plant RNeasy protocol (including the Qia-shredder for additional homogenization[70]) and the RNA was eluted into 50 µl nuclease-free water. The approximate nucleic acid concentration was determined by NanoDrop, after which the samples were diluted with nuclease-free water to a uniform 10 ng µl⁻¹ (DNA and LSV-1(RNA)) or 20 ng µl⁻¹ (for all other RNA samples) and stored at −80 °C.

**RT-qPCR and qPCR**. The various microorganisms and host mRNA targets were detected and quantified by either One-Step reverse transcription-quantitative PCR (RT-qPCR) for pathogens with a RNA genome and the immune and internal reference gene mRNA targets, or by quantitative PCR (qPCR) for organisms with a DNA genome. Details of the assays are shown in Supplementary Table 10, Supplementary Table 11 and Supplementary Table 12. The reverse primer for *Amel/LRR* was slightly re-designed from Di Prisco et al.[19] because the extremely high complementarity between the original forward and reverse primers resulted in high levels of PCR artefacts dominating the quantitative signal. The reactions were conducted in duplicate, in 20 µl (DNA) or 10 µl (RNA) reaction volumes containing 2 µl (DNA) or 1.5 µl (RNA) template, 0.4 µM (DNA) or 0.2 µM (RNA) of forward and reverse primer and either the Bio-Rad Eva Green qPCR mix (DNA) or the Bio-Rad One-Step iTaq RT-qPCR mix with SYBR Green detection chemistry (RNA). The reactions were incubated in 96-well optical qPCR plates in the Bio-Rad CFX connect thermocycler, using the following amplification cycling profiles: 10 min at 50 °C for complementary DNA (cDNA) synthesis (RT-qPCR only): 5 min at 95 °C (to inactivate the reverse transcriptase and activate the Taq polymerase) followed by 40 cycles of 10 s at 95 °C for denaturation and 30 s at 58 °C for primer annealing, extension and data collection. For DNA assays the following amplification cycle profiles were used: 2 min at 98 °C for the initial denaturation followed by 40 cycles of 5 s at 98 °C for denaturation and 10 s at 60 °C for primer annealing, extension and data collection. The amplification cycles were followed by a melting curve analysis to determine the specificity of the amplification by reading the fluorescence at 0.5 °C increments from 65 °C to 95 °C. Included on each reaction plate were positive and negative (non-template) assay controls. For each type of assay (Supplementary Table 10, Supplementary Table 11 and Supplementary Table 12) a calibration curve was prepared through a 10-fold dilution series of a positive control of known concentration covering 6 orders of magnitude, for quantitative data conversion, establishing the reference melting curve profile of the amplicon and estimating the reaction performance statistics.

**Data conversion and normalization**. The melting curves of individual reactions were evaluated visually in order to separate out non-specific amplifications, which differ in melting temperature profiles from true target cDNA/DNA amplicons. Non-specific amplifications were deleted from the dataset. All assays were run in duplicate, with the mean value of these two duplicates used in further calculations. Both duplicates had to yield a positive quantitative value and pass the melting curve analysis for the data to be included in the dataset. The raw RT-qPCR data of all confirmed amplifications were subsequently converted to estimated copy numbers of each target RNA, using the corresponding calibration curve for the assay. These data were multiplied by the various dilution factors throughout the procedure to calculate the estimated copies of each target per bee[69]. Since RNA is easily degraded, there is a risk that differences between individual samples in RNA quality (i.e., degradation) can affect the results[70]. To correct for this, RT-qPCR assays for the mRNA of a common honeybee internal reference gene (RP49) were run on all samples. The data for the RNA targets of interest were then normalized to the average value for RP49 mRNA, thus correcting the data for sample-specific differences in RNA quality with respect to RT-qPCR performance[70].

**Statistical analyses**. The proportions of honeybee-collected pollen that originated from oilseed rape-type plants were compared between treatments (clothianidin seed treatment/untreated) using a generalized linear model assuming binominal distribution and correcting for overdispersion. The clothianidin concentrations in nectar and pollen collected by honeybees and in bee tissue were compared between treatments using Wilcoxon–Mann–Whitney tests. To compare the concentrations of clothianidin in the bee tissue and nectar in individual bees between fields, we used analyses of variance (ANOVA), with field identity as predictor. Furthermore, the concentrations of clothianidin in the tissue and nectar stomach content of individual bees were related using a multiple linear regression with field identity and concentration of clothianidin in nectar as explanatory variables and concentration of clothianidin in bee tissue as response variable.

The study followed in general a Before-After-Control-Impact (BACI) design, with a paired field structure, repeated for two consecutive years at colony level for data on colony development as well as the prevalence and abundance of parasites, pathogens and gut bacteria. The years, 2013 and 2014, were analysed together in one full model, with seed treatment, bloom, year and their interactions as fixed factors. The effect of the clothianidin treatment was assessed by the interaction between bloom and seed treatment, as this term reflects the difference in change between treatments over the oilseed rape bloom(s). If the three-way interaction (bloom × seed treatment × year) was significant (i.e., if the variable responded differently to the clothianidin treatment from one year to the next), the dataset was split by year and year was dropped as a fixed factor. Furthermore, the dataset was analysed for only 1 year if the data consisted of a sample size ≤10 in one year both for microbiota prevalence and abundance (Supplementary Table 1). Colonies that swarmed (8 at control fields and 10 at clothianidin-treated fields in 2013; one at a control field and two at clothianidin-treated fields in 2014) were excluded from the analysis, since swarming has a large effect on colony development. Also excluded is the single colony that lost its queen during transport before field placement in 2013 (treated field). Excluding colonies that swarmed from the analysis qualitatively altered some results (see Supplementary Table 13). Changes in significance level might be due to reduced statistical power, random chance or biological effects.

Linear mixed-effects models (LMM) were used to test the effect of the clothianidin treatment on colony development measured as the number of capped brood cells (amount of brood) and the number of adult bees. Seed treatment (clothianidin or control), bloom (before or after oilseed rape bloom), year (2013 or 2014) and their interactions were fixed factors. Farm pair identity, farm identity and colony identity were included as random factors. Honey production was compared between treatments using a LMM with farm pair identity and farm identity as random factors. Generalized linear mixed models (GLMMs) were used to test the influence of the clothianidin treatment on re-queening and mortality of the colonies with farm identity as a random factor.

The influence of the clothianidin treatment on spring colony development, measured as the number of adult bees and amount of brood, was tested using a LMM and a GLMM, respectively, with seed treatment as fixed factor and farm pair identity and farm identity as random factors. For the number of capped brood cells we used a negative binomial error distribution and logarithmic link function.

The microbiome and *Varroa* mite data were analysed both on their binomial (presence/absence) and quantitative (abundance) character, using GLMMs (with binomial error distributions and a logit link function) and LMMs (with normal error distributions), respectively, with seed treatment, bloom, year and their interactions as fixed factors. GLMMs on microorganism or *Varroa* mite prevalence included only colony identity as random factor, as the effective sample size (i.e., the less frequent outcome of the presence/absence data) did not allow for the inclusion of more random factors. Only organisms and years with an (effective) sample size >10 were analysed for both the prevalence and the abundance data. In addition, colonies that did not at least once test positive for a particular microorganism were excluded from the analysis of abundance. Bee pathogen and bacterial abundance were logarithmically (log₁₀) transformed, as they are generally exponentially distributed. LMMs on target organism abundance contained farm pair identity, farm identity and colony identity as random factors. *Varroa* mite numbers per 100 bees and colony weight were square-root transformed to avoid non-normally distributed residuals. Confidence intervals were calculated based on profile likelihood. For the square-root transformed data, estimates were back-transformed to the original scale for graphical illustrations.

The immune gene transcripts were only available for 2013 and at apiary level but also followed the BACI design. LMMs on gene expression contained seed treatment and bloom as fixed factors and farm identity as random factors.

Statistical data analyses were performed using R except for analyses addressing the verification of clothianidin exposure and land use, for which SAS 9.4 for Windows (SAS Institute Inc.) was used. LMMs were fit using the *lmer* function of the *lme4* package and GLMMs were fit using the *glmmTMB* function of the *glmmTMB* package in R. P values from GLMMs were calculated by likelihood ratio tests. P values from LMMs were calculated using the *Anova* function of the *car* package, whereby type-III F-tests were used for models containing interactions and type-II F-tests for models without interactions (i.e., spring assessment and honey production). Effects of fixed factors were estimated using sum-to-zero contrasts in all models except those on neonicotinoid residues. Sum-to-zero contrasts allow for the determination of main effects/interactions (i.e., estimation independently of other independent variables) and show the effects of factors as deviations from the grand mean (intercept). For factors with two levels the magnitude of the deviation of each level from the grand mean is the same but the direction differs. We represent effects of factors (seed treatment, bloom, year), as the deviance of the second level (clothianidin, after, 2014) from the grand mean. This was also the case for interactions, so that for example the seed treatment × bloom interaction indicates to what extent clothianidin-exposed colonies differed in change over the oilseed rape bloom from the mean change of both treatments.

**Power analysis**. We performed power analyses for number of adult bees and the number of capped brood cells and honey production to investigate the effect size we could potentially detect given our design, replication and model choice. Power was determined for a range of effect sizes at a nominal confidence level of $\alpha = 0.05$ by 1000 Monte Carlo simulations per effect size using the powerSim function of the simr package. Power was calculated for a range of effect sizes, expressed as the change in number of adult bees, number of capped brood cells or honey production. By dividing the effect size with the mean number of bees, the mean number of brood cells or honey production of all control colonies, we obtained effect size expressed as the percentage change of those matrices (Supplementary Fig. 3). This power analysis made it possible to compare our effect size with the effect size presented by Rundlöf et al.[34]. Using the full model, we could detect an effect size for the number of adult bees of below 5% with a power of 80% compared to the effect size below 20% presented by Rundlöf et al.[34]. This is even lower than the requirements of an effect size <7% set by EFSA[32]. As a result of a significant interaction of seed treatment, bloom and year, the dataset for the number of capped brood cells were analysed separately for each year. Therefore, we present here the power analysis for each year. The effect size at which 80% was reached increased from below 10% in 2013 to slightly below 11% in 2014 (Supplementary Fig. 3), likely due to the reduced replication in 2014. We also performed a power analysis of honey production (amount of honey per colony in kg) using the dataset of both years, showing that an effect size of below 20% could be detected with a power of 80% (Supplementary Fig. 3).

**Reporting summary**. Further information on experimental design is available in the Nature Research Reporting Summary linked to this article.

## Data availability

The authors declare that the data supporting the findings of this study are available within the paper, its supplementary information files and/or Rundlöf et al.[34]. The datasets generated during and/or analysed during the current study are available from the corresponding author on reasonable request.

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

## Acknowledgements

We thank the farmers for collaboration, the project group for feedback, A. Gunnarson for farmer contacts and seeds, B. Andréasson and J. Bergstrand for producing and managing honeybee colonies, T. Carling, A. Andersson and A. Main for assessing and sampling honeybee colonies, the Centre for Chemical Pesticides for financing the work on individual exposure and stability of pesticides in bees, M. Stjernman, J. Dänhardt and SAPES for land use information and farmer contacts and V. Hederström for pollen identification. We are deeply grateful to I. Fries for his guidance and support throughout this project. Funding for the field project was provided by the Swedish Civil Contingencies Agency (to T.R.-P., R.B. and H.G.S.), the Swedish Research Council VR (grant no 330–2014–6439) (to M.R.), and the Swedish Research Council FORMAS (to H.G.S and R.B.). Parts of the pathological study were financed internally at SLU. The study of individual exposure and stability of pesticides in bees was financed by the Swedish Environmental Protection Agency (to O.J.).

## Author contributions

Conceptualization: J.R.d.M., M.R., R.B., H.G.S., T.R.-P.; methodology: J.R.d.M., J.O., D.W., E.F., E.S., P.O., M.R., O.J.; validation: J.R.d.M., M.R., E.S., P.O., O.J.; investigation: J.O., D.W., E.S., P.O.; chemical analysis: O.J.; formal analysis: J.O., D.W., M.R., R.B.; data curation: J.R.d.M., M.R.; writing: B.L., J.O., D.W., M.R.; editing: all; visualization: B.L., J.O., D.W., M.R.; supervision: J.R.d.M., M.R., P.R.; project administration: E.F., M.R., T.R.-P; funding acquisition: E.F., J.R.d.M., M.R., O.J., R.B., T.R.-P., H.S.

## Additional information

**Competing interests:** The authors declare no competing interests.

