## [Peer Review File · Nature Communications]

Reviewers' comments:

Reviewer #1 (Remarks to the Author):

Key Results

The manuscript is a companion work to an earlier publication (Rundlöf et al, 2015, Nature 521, 77-80) that described a lack of effects of clothianidin from seed-treated oilseed rape on honey bee colony performance. The current manuscript expands the experiment to a successive year (2014) and examines additional effects on honey bee gut microbiota (*G. apicola* and *S. alvi*), pathogens (*N. apis/ceranae* and several major viruses), parasites (Varroa mite), and major immunity genes. Notably, the authors detect very limited effects of clothianidin exposure on honey bee colony performance, microbiota, and immune responses despite large differences in treatment clothianidin exposure (as detected in workers collected at the colony entrance). Clothianidin treatment affected only four parasites/pathogens but not in a consistent manner from year-to-year. Importantly, the authors note that deleterious effects of neonicotinoid exposure observed in highly structured laboratory or field experiments generally were not observed here under robust field trials. From a design perspective, the experiment thoroughly simulated clothianidin exposure from seed-treated oilseed rape (a major exposure source in Europe and other parts of the world) via robust replication of field treatment and apiary sites. Once subjected to increased variation from sites, seasons, and the surrounding landscape, differences attributable to clothianidin exposure were rarely observed. The authors also found considerable variation in the amount of clothianidin present in entrance workers (including foragers) and collected food materials, despite the uniformity of nearby treatment plots. As a manuscript, this study emphasizes that real world impacts of colony stressors observed under highly controlled conditions often do not occur at detectable levels in agricultural field settings.

Weak points of the experiment/manuscript include: 1) very small colony sizes, 2) sparse description of surrounding landscape (other sources of forage and pesticides), 3) lack of broad spectrum agrochemical testing (to determine full range of agrochemical exposure from the landscape outside the treatment fields), 4) lack of agrochemical testing of hive food stores (to assess what most colony bees are exposed to), 5) the underpowered mite sampling technique, and 6) uneven virus sample analysis replicate size. The experiment would have been stronger had the authors measured a long-term performance and survival metrics (perhaps colony survival and performance through the following winter) – the timeframe between start and finish of the rapeseed bloom was very short, especially given that the colonies did not expand much in size overall.

Validity

Three main concerns, all which might be readily addressed:

1) Conceptually, a major design question is whether the two years of experiments are sufficiently independent to be considered separately of one another or whether the two years represent different periods of one timeline (see Data/Methodology 1a). Colonies entering the 2014 treatments had a previous year treatment within the same treatment behind them. Although the authors clearly acknowledge that the second year is a continuation of the treatments, the manuscript should probably be reworked more towards this perspective rather than simply considering the second year to be a separate year.

2) The authors appear to have conducted a narrowly focused pesticide screening (for clothianidin only) on foragers and their loads at one timepoint only. Assessments of pesticide effects at the landscape level require pesticide assessments for the entire landscape. This screening may inadvertently miss other agrochemicals present from other landscape sources (i.e. nearby fields outside experimental control) and accumulation of agrochemical residues in food stores and wax (i.e. sources of colony bee exposures, see Data/Methodology 2). These latter agrochemical sources

might have chronic effects on bees other than foragers within the hive.

3) The number of samples used to determine virus prevalence varied considerably between virus type (see Data/Methodology 3b). This is a fundamental problem with low-incidence viruses, where the number of samples examined has a strong impact on detection rate.

Originality/Significance

This study is the first successful attempt to examine the sublethal effects of clothianidin seed-treated OSR exposure on honey bee colony performance, pathogens and gut microbes in realistic agricultural settings. Previous laboratory and semi-field experiments found sublethal effects of neonicotinoids on pollinators including honey bees that lead to a temporary ban on neonicotinoid applications. The European neonicotinoid moratorium resulted in a call for field-relevant sublethal studies in agricultural fields to accurately assess sublethal effects on pollinators. These authors chose to examine effects of clothianidin seed-treated oilseed rape (OSR) – one of the most common uses of neonicotinoids in the Northern European landscape – and hence a likely source of pesticide exposure for pollinators. While other studies have examined effects of OSR neonicotinoids on honey bees (Culter et al., 2014, Peer v2 e652); Rolke et al., 2016a and b, in *Ecotoxicology* 25), this manuscript differs by its focus on effects on parasites, pathogens and two common gut microbes in honey bees.

Aside from its companion paper (Rundlöf et al., 2015, *Nature* 521, 77-80), this study most closely resembles a study by Rolke et al., 2016, *Ecotoxicology* 25: 1648-1665 on the effects of OSR seed-treatment clothianidin on honey bee performance, survival, and pathogens in a Northern European landscape (Germany). Similar to this study, Rolke et al., 2016 found no significant effects of clothianidin seed-treated OSR on colony performance, colony weight (honey yield), and pathogen infections.

However, this study differs from Rolke et al., 2016 in several key ways. Critically, *Varroa* mite infestation rates were much more moderate in this study than in the Rolke study, which experienced a severe mite outbreak across all treatments. Since several major bee viruses are essentially vectored by *Varroa*, the impact of mites on virus abundance and colony stress in the Rolke study may have swamped any effects from clothianidin exposure. This study also had a broader diversity of study sites and field replicates than Rolke et al., 2016; however, Rolke et al., 2016 conducted many more assessments (7 vs. 2) in their single season than in each of this study's seasons and examined additional performance and site metrics. This study also differs from Rolke et al., 2016 most prominently by its inclusion of analysis of two dominant honey bee gut microbes, *G. apicola* and *S. alvi*.

These field-relevant results contradict recent findings describing negative interactions between neonicotinoid exposure and microbial health in honey bees, albeit under more controlled conditions. Given that these experiments are more realistic field tests, the lack of significant effects of clothianidin on colony microbes (including viruses) and performance is noteworthy. Findings such as these are of considerable interest to people both within and outside the immediate discipline. These results would improve understanding of impacts of neonicotinoids (most commonly used insecticides worldwide) on pollinators and likely inform policy debates worldwide surrounding the use of neonicotinoids.

Data/Methodology

Overall, the methods and design for the experiment represented a solid approach for assessment of the effects of seed-treated clothianidin exposure on honey bee colonies in realistic field conditions. With the exception of small colony size, landscape descriptions, and short distances between field plots, the field design is both appropriate and easy to replicate (conceptually, not logistically). Raw data was not provided with the manuscript, but could easily be included for transparency sake. The supplemental material was both appropriate and useful for the understanding of the results, although specific concerns/comments could improve the

presentation. Most concerns could be addressed by additional information, rewriting, or additional tests.

Specific concerns arose in three areas regarding data/methodology:

- 1) Does the experiment simulate realistic field exposure of honey bees to clothianidin from seed-treated oilseed rape flowers through a single crop forage exposure?
- 2) Does the experiment adequately capture the exposure of honey bees to clothianidin and other agrochemicals from both treatment fields and the surrounding landscape?
- 3) Does the experiment adequately measure targeted effects of clothianidin on honey bees?

1a. The first and second year experiments do not appear to be sufficiently independent because the second year colonies were exposed to treatments for two years. Colonies in the second year are specifically assigned to the same treatments as in the first year. Therefore, the second year appears to be a continuation of the first year treatments, and potentially contains carryover effects from the first year treatment. Because of this, direct head-to-head comparisons of yearly effects seems inappropriate since the 2014 colonies (unlike 2013 colonies) have had previous (and perhaps cumulative) exposure to clothianidin. In essence, the experimental design appears to be a single exposure timeline over two years. Interestingly, the absence of differences between treatment groups actually strengthens the results because exposed colonies endured two successive years of exposure without apparent effect.

Critical to how this might be resolved is whether the colonies were scrambled and "equalized" between the first and second years. At the beginning of the second year, the colonies are described as being of similar small size and strength. However, it is unclear whether this similarity in size occurred due to "equalization" of colonies within a treatment group (via frame, worker, and brood swaps) or simply as the state of colonies in June 2014.

1b. The honey bee colonies used here were exceptionally small compared to most managed colonies, even nucleus colonies. Wu-Smart and Spivak (2016) recently found that smaller colonies are more susceptible to neonicotinoid exposure than larger colonies. The authors should justify the reasoning behind the use of such small colonies. Concerns about the vulnerability of such small colonies is partially offset by the reasonably low overwintering mortality of colonies in both treatments; however, the effects observed on these small colonies may not be representative of most honey bee colonies.

1c. Not clear why 1-2 year old experimental queens were used to establish the colonies in the first year, given the widespread problems with queen failures among older queens. The older starting age and age range seems to introduce unnecessary variability into the experimental design, as older queens (older than 1 year) commonly fail for various reasons unrelated to pesticide exposure. The apparent exclusion of colonies headed by queens that were 2 years old (at the beginning of the first experimental year) before the second experimental year observations seems like an avoidable loss of colony numbers.

2a. Were nearby fields treated with agrochemicals before or during the experiment, and did experimental bees collect these agrochemicals? Bee colonies may be contaminated by agrochemical applications originating outside the treatment fields (i.e. the surrounding landscape). Researchers often have limited control over which agrochemicals are applied to fields within flying radius of their experimental colonies. The authors estimation of 4 km as a sufficient distance to separate colonies probably holds true during a strong bloom when bees forage near their colonies. However, foragers will venture much further than this distance when forage is less available. This is evident by the presence of clothianidin in 2013 control colony bee and nectar samples – pesticide contents that indicate that these fields are probably not completely separated.

2b. Were the agrochemical screenings broad spectrum or were these screenings specifically targeted only to the experimental plot pesticides? It is not clear from the text whether colonies

were screened for agrochemicals other than the chemicals applied within the experimental plots. Non-treatment agrochemical exposures would likely occur at some sites but not others – if so, deleterious effects from other non-treatment agrochemicals might increase variability in outcomes and metrics between sites. Neonicotinoids have recently been shown to act synergistically with some other agrochemicals. Broad spectrum screenings at each apiary would clear this question up.

2c. The pesticide screening methods used here provides a good snapshot of pesticide contamination of foragers (worker tissue, corbicular pollen pellet, and honey stomach nectar contents) during peak oilseed rape bloom, but do not provide a comprehensive screening assessment of materials most likely to lead to long-term contamination of the rest of the colony (stored pollen, honey stores, and wax). The authors' reports on pesticide screening of worker tissue, corbicular pellet, and honey stomach nectar appear to be limited to the pesticides present from seed and field management applications during peak rape oilseed bloom. Researchers commonly perform limited broad spectrum pesticide screenings of hive materials (stored pollen, honey, wax) to detect agrochemicals brought in from the landscape environment to the colony. Testing of food materials would also allow better comparisons to structured lab and field studies where the food pesticide exposure are well known. Broad spectrum pesticide screenings of these materials would allay these concerns.

2d. Were other oilseed rape fields located near the experimental fields, and if so, were they in bloom during the experimental period or treated with pesticides? Rundlöf et al, 2015 notes that in 2013 other oilseed rape fields were located as close as 2 km from the experimental fields. The presence of nearby non-experimental oilseed rape fields would add to the oilseed rape pollen detected in the pollen composition analysis and inflate estimates of treatment plot pollen exposures.

Most of the methods used to assess effects on colony development/performance and microbe/parasite prevalence/amounts were appropriate. Most of these metrics and their results are presented adequately in the Materials and Methods and Results. Suggestions to correct minor errors, misinterpretations, sections lacking clarity, and minor grammatical/idiomatic errors are listed below.

3a. The Varroa mite sampling method is underpowered and targets older workers that are not preferred as hosts by phoretic mites. Techniques for estimates of phoretic mites commonly sample 300 workers (not 100 workers), as the incidence of phoretic mites on workers in healthy colonies is often below 10% infestation (see Lee et al., 2010 J Econ Entom 103: 1039). Such sampling is also usually performed on nest worker bees as Varroa mites very much prefer younger adult bees engaged in nurse and housekeeping activities over older bees engaged in outside activities such as foraging. However, the Varroa infestation rates of entrance bees is probably correlated with infestation rates of nest bees.

3b. The limited testing (one year only with different sample numbers) of entrance bees for most viruses and immune gene expression is highly problematic. These parts of the experiment would be much more robust if two years of complete data had been obtained and comparisons were made with similar numbers of samples. Especially troublesome are comparisons of virus prevalence for low-incidence viruses using different sample sizes. One would imagine that the ability to detect a low-incidence virus would be highly dependent upon how many bees were examined. In this case, the sample numbers vary considerably between viruses. Likewise, the argument that the absence of the less common viruses in the first year justifies their exclusion from second year screening is not convincing given what is known about these viruses. The prevalence and tissue abundance of many honey bee viruses is not consistent but varies erratically seasonally, annually, and developmentally. I would be less than confident that one year's single timepoint screening fully captures virus expression given the seasonal, year-to-year, and colony-by-colony variation.

Appropriate Use of Statistics and Treatment of Uncertainties

The primary concern regarding statistics is whether the two years experiments were actually independent by year or represented one two year timeline (see Data/Methodology 1a). If the two years were not independent, then the authors need to rework their year-to-year comparisons. A second question is whether the experimental design was sufficiently robust to support conclusions of a lack of significant effects of clothianidin. The number of colonies was adequately robust for the statistical comparisons in 2013, but more marginal in 2014 (24 colonies (clothianidin), 16 colonies (control)). The 2014 colony numbers may not have been sufficient to detect low incidence virus infections (see Data/Methodology 3b). However, if the purpose was to identify moderate to high prevalence viruses for further comparisons (which it appears the authors intended), the colony numbers were sufficient for this purpose. The text could simply reflect this end more clearly.

All error bars in the figures and supplementary figures were defined clearly in the figure legend. For two figures displaying microbe/parasite prevalence (Figure 3 and Supplemental Figure 2), error bars are present without any apparent % prevalence. Shouldn't the average prevalence be above 0% in these cases?

Most of the statistical tests were sufficiently robust and presented clearly in the results and tables except for the sections on microbe prevalence and amounts. These sections are sometimes confusing because it is unclear which comparison is being made. Sample comparisons differ by year (2013/2014) and unit of comparison (colony/apiary (field)). Confusion from poor labeling is further exacerbated by the fact that similar sized sample sets were not analyzed for all microbes. Only the most prevalent microbes had full sample sets analyzed for the 2013 apiary, 2013 colony, and 2014 colony comparisons. The authors need to thoroughly label text, figure and table headings to ensure that readers know exactly which comparisons are being made.

Notably, most statistical tests used in the microbial analyses were sensitive to multiple occurrences of zero counts. Some microbe comparisons appear not to have been performed due to high numbers of apiaries or colonies with no detectable counts.

Conclusions

For the most part, the results obtained here support the conclusions laid forth in the manuscript. One main concern with experiments that find a lack of significant treatment effects is whether the design was sufficiently robust to adequately test this case. Unfortunately, the sections comparing microbial prevalence (particularly low-incidence viruses) are based on uneven sample sizes and are quite complicated. The fact that some microbes were assessed only in one year and some in two years becomes problematic if not well framed in the text. Although certainly unintended, the reader may be left with an impression that a given microbe was not present in both years when it was sampled only once. These microbe prevalence sections could be improved either by testing all microbes with equal sample numbers or making comparisons with more explicit note of how many samples the comparisons were based on. These sections need reworking for clarity.

One of the four significant impacts of clothianidin (2013 – greater seasonal decrease in *S. alvi* in clothianidin) is understated both in the abstract and the rest of the manuscript. In a related problem, the authors do not adequately discuss what the impacts of differences in the gut microbes *S. alvi* and *G. apicola* might mean. Would lower amounts lean toward microbial dysbiosis? Be indicative of different worker age structures? The role of these gut microbes in bees is a fairly important topic worth commenting on now.

One aspect of the conclusions that is somewhat problematic is the generalized statement that clothianidin does not impact pathogen levels. While most metrics were unaffected by the clothianidin treatment, some aspects of colony microbes were in fact affected by clothianidin exposure and should be presented as such. The manuscript could have just as easily been titled

“Clothianidin field exposure impacts honey bee pathogen, pathogen, and microbe levels, but not colony-level performance or immune gene expression” based on what presented about 2014 *G. apicola*, BQCV, and *Varroa* levels in the abstract and Table 3.

The results sections on microbial prevalence and amounts have a few errors and inconsistencies between the text and figure/tables (listed below), most of which are relatively minor and readily corrected. Some semantic problems occur in the results text due to confusion between time (period or before/after oilseed flowering) and time (between years).

line 163 Lacks supporting analysis. The 2013 ABPV comparisons are missing from Supplementary Table 2 (numbers too low for analysis?) and 2014 ABPV analysis does not show a significant effect by Period, Treatment, or the interaction. DWA-A increases significantly in prevalence through the Period in both 2013 and 2014 as shown both in the statistical analysis and graph.

line 165 Conflict between results text and Supplementary Table 2. Colony prevalence of *Varroa* mites significantly increases in 2013, but not 2014.

line 166 Conflict between results text and Supplementary Table 1. ABPV is not absent in 2013 at either the apiary or colony level. Perhaps “largely absent” would be better?

In183/Supplementary Table 2. Missing 2013 *N. apis*, BQCV, and LSV-1 prevalence data despite descriptions in results text.

line 199 Conflict between results text and Figure 4. For both 2014 clothianidin and control field colonies, the *Varroa* infestation rates appear to increase, but control field colonies levels appear to increase more.

line 216 Conflict between results text and Table 3. There was a treatment x period interaction for 2013 *S. alvi*, not a treatment effect.

Suggested improvements

The experiment would have been more robust if the post-bloom assessments had been extended beyond the end of the bloom. Several recent publications have noted chronic effects of neonicotinoids some time after the initial exposure.

1. A better understanding of the full agrochemical exposure of these colonies (aside from clothianidin) is essential for a landscape study, given that colonies could easily be contaminated by outside agrochemical exposures. Likewise, the authors could expand on descriptions of alternative landscape forage, particularly oilseed rape forage.

2. Pesticide analysis of food materials from inside select colonies would capture dietary pesticide exposures of the majority of colony bees, given that most food consumption other than forager nectar consumption for flight occurs with food stores. Neonicotinoids such as clothianidin persist for months once stored under dark conditions in honey stores. Such an analysis would also allow better comparisons with the existing literature as well.

3. A major problem with the virus prevalence analyses is that different numbers of apiary and colony samples were used to compare virus infection rates (see Data/Methodology 3b). Given that the ability to detect uncommon virus is largely dependent on the number of samples compared, this inequality is problematic. The authors could address this problem by analyzing sufficient additional virus samples to make comparisons with similar numbers of pooled apiary or pooled colony samples. Alternatively, the presentation could be reworked to simply present the first year virus screenings as just that – and avoid erroneous perceptions that viruses are being compared head-to-head in both years.

References

Overall, the authors cite and discuss the relevant literature quite well. I have only a few references to suggest as additions, most of which simply add to the discussion or citations.

75 A recent publication by Odemer et al. 2018 *Ecotox.* also found a lack of sublethal effects with clothianidin exposed bees.

80 Dussaubat et al., 2016 *Sci. Reports* 6: 31430 found synergistic effects between *Nosema* and queen survival.

253 "...to chronic exposure to neonicotinoids." Sandrock et al., 2014 *PLOS One* 9: e103592 and Dively et al., 2015 (citation 54 here).

322 See Meikle et al., 2016 *PLOS One*. Sublethal effects of imidacloprid on honey bee colony growth and activity at three sites in the United States for persistence of neonicotinoids in food stores.

327 "... reported in other studies." See Rolke et al, 2016 *Ecotox.* 25: 1691

One section that could use expansion concerns the two core gut microbes, *G. apicola* and *S. alvi*. These two microbes need to be considered separately from the parasites and pathogens because these are associated with a healthy microbiome. That being said, relatively little is known at this point as to their interaction with other colony stressors, although it is very much a hot topic of future interest. (see Collison et al., 2016, *Biol. Rev.* 91: 1006)

Clarity/Context

Since the manuscript is a continuation of a previously published experiment (Rundlöf et al., 2015), the authors face the dilemma of what to include from the previous manuscript and what to leave out in the current manuscript. The authors could readily incorporate more methods and results from Rundlöf et al., 2015 (recap?) to ensure that this manuscript stands alone in its presentation to readers. By comparison to its companion manuscript, the Materials and Methods section of this current manuscript is less descriptive, less specific, and lacking in some key details. Several parts of the M & M could be rewritten to provide a clearer narrative of what was performed, especially regarding the experimental timeline.

Some metrics are not clearly or consistently described. For example, estimates of adult bees are variously described as number of adult bees and references are made to colony weight. How did the authors measure both of these metrics? What exactly does colony weight measure and how was it adjusted for weight differences before the pesticide exposure/oilseed bloom? While the methods behind these metrics may be in the references, these metrics could be better detailed in the current manuscript. Areas that might need more details are listed in specific comments below.

The authors need to be more clear in their presentation of microbial and *Varroa* prevalence and amounts in the results text, Figure 2, Figure 3, and Supplementary Figure 2. As currently presented, the results can be very confusing to the reader given the variety of comparisons made. Data is presented for 2013/2014 samples and at the colony or apiary level, and some analyses were performed only in one year (many viruses and immune gene expression for 2013 samples). I would strongly suggest that the authors tightly label their results by year and by apiary/colony analysis levels. I would also strongly suggest that the authors indicate table or figure sections where samples that were not analyzed during a given year (i.e. many virus and immune gene expression analyses were only analyzed in 2013) with N/A (not attempted) rather than n.s. (not screened). The latter acronym could be easily confused with n.s (not significant). Also, the absence of analysis should be indicated directly on the table or figure by N/A due to the similarity

between zero values and skipped analyses. An asterisk on the left side next to the pathogen name is not as effective in indicating that the second year was not attempted.

Explicit distinctions need to be made between colony prevalence and apiary (field) prevalence in both the text and the figures/tables. Also, label parts of graphs or tables where no analysis was performed with a more prominent marking (perhaps N/A for not attempted, rather than n.s. which resembles not significant). Otherwise, it is hard to distinguish between zero values and analyses that were not performed.

On two figures displaying microbe/parasite prevalence (Figure 3 and Supplemental Figure 2), error bars are present without any apparent % prevalence. Shouldn't the average prevalence be above 0% in these cases?

I would hesitate to refer to two timepoints over one forage crop in two years as covering "seasonal" effects or a "season". Most longitudinal studies of this kind examine colonies at multiple timepoints to obtain a fuller perspective. A better set of terms would be "pre-forage/post-forage" or some variant centered around the bloom. But the authors should still consider effects from a seasonal perspective. Also, "abundance" would be a better term than "amount" to describe the number of pathogens or parasites detected.

Parts Outside My Purview

I've published projects on all aspects of this study.

Specific comments

Abstract and Introduction

Abstract Succinct and generally to the point. The authors need to comment on the finding that *S. alvi* abundance seasonally decreased more in clothianidin colonies than control colonies.

The introduction nicely sets up the the experiment and its rationale – first by focusing on bee stressors individually, then considering interactions, and finally by noting the lack of field-relevant experiments. The authors need to expand on the importance of the two gut microbes *S. alvi* and *G. apicola* – why include these in here, and is there any penalty to bees if these are deficient?

Results

119 Include the 2013 pollen species composition analysis here with the 2014 analysis, or at the very least, briefly recap the findings. The oilseed rape proportion of the pollen samples is very different between 2013 and 2014, although not between the treatments for each year. Interestingly, the 2013 corbicular pollen pellets contains less oilseed rape pollen than 2014 pollen, but results in higher clothianidin exposure as measured by bee tissue and nectar contents.

122 Please comment on the presence of clothianidin residues in 2013 control field bees (Table 1). In 2013, clothianidin residues were detected in workers collected from control field colony entrances. Such a finding is not unreasonable at all for a field study of this kind, but needs some explaining. What does this mean for isolation of colonies near treatment fields? Colonies likely forage predominantly from their assigned fields, but also probably forage a little from the other treatment field and other fields in the surrounding landscape.

123 Explicitly state that clothianidin residues are higher in bees from colonies in clothianidin-treated fields than bees from colonies in untreated control fields. This result is obvious from Table 1, but it should be presented in the results.

124 ... approximately half of ...

138 Field-to-field variation in clothianidin exposure is important and likely a reflection of landscape

forage differences.

139 Explicitly state period effects for the 2014 colony development in the results section. The strong effects of period on two of the three metrics is obvious in Table 2, but needs to be presented in the results.

138 For the 2014 Before/After results (Table 2), the differences in the capped brood number between treatments is only marginally non-significant ($p=0.066$). Notably too, the number of capped brood cells from before to after observations decreases in clothianidin treatment colonies but increases in control colonies. Do the results come out differently if pre-bloom (Before observation) metric differences are compensated for for?

147 overwintering rather than in-wintering?

163 Break this sentence into two parts.

161 ... highly prevalent in both colonies and apiaries, but were not affected by either treatment or sampling period.

171 ...the prevalence of *V. destructor* increased in colonies from ...

173 Difficult to comment on the prevalence of CBPV and SBPV when these viruses were analyzed in just one year. Be careful about giving the impression that the samples were examined for these viruses for more than one time point/collection set.

179 also Figure 3

187 Is there any way to compensate for this initial (before bloom) difference in 2013 *Nosema ceranae* prevalence? This would seem to disrupt further analysis if left uncorrected.

187 Is "prior to placement in the fields" the same as the Before oilseed rape bloom sampling time point? These would appear to be two different times.

193 *N. ceranae* was also included in this analysis.

202 ... were unaffected ...

211 Are these the increases in mites/100 bees or the actual values?

223 *S. alvi* titers increased during the oilseed rape bloom in both 2013 and 2014.

224 ... remained consistently high through the experiment and were not significantly altered during the bloom in either 2013 or 2014.

229 ... at the apiary level.

233 What kind of seasonal effects? Increases or decreases?

Discussion

Once again, comment on the gut microbes (half of the observed differences attributed to clothianidin treatment).

267 ("Healthy" colonies.) But the colonies were very small and the *Varroa* counts/virus abundance were relatively slight compared to highly stressful conditions. The fact that the colonies survived

so well despite being small under pesticide exposure is worth noting.

277 The Varroa infestation levels here are relatively slight and the amount of time from the beginning to the end of the oilseed rape bloom is relatively short.

290 What about the greater post-bloom decrease in *S. alvi* abundance in 2013? That result contradicts this statement.

309 Such as induced responses in a newly-emerged adult bee?

321 What about the dilution effects of landscape forage outside the treatment plots? I don't know that this was examined closely, but this could account for major differences.

327 I agree that these probably reflect constitutive gene expression for the reasons the authors elaborate on, but I don't think you can rule out induced responses entirely since this was not tested for.

Materials and Methods

364 Was land use the same over successive years (2013 and 2014)? If not, how did it change, especially as it relates to landscape pesticide exposure outside the experimental fields? In R15, land use similarities were based on 2011 use (not the experimental years). The experimental oilseed rape field sizes are quite sufficient to support and attract the colony density mentioned here.

388 Biscaya inclusion/exclusion results are located in Supplementary Table 6, not 4.

393 Rundlöf et al, 2015 mentions that colonies were equalized in 2013 – were the colonies equalized in 2014 between the spring and before bloom observations to obtain colonies of this description? The randomization of 2014 colony location between fields (within treatment group) was highly appropriate.

419 The switch in pesticide treatment of individual fields from 2013 to 2014 is described three times here. This repetition can be reduced.

423 Some confusion here as to which level (colony or field) these clothianidin concentrations represent. The wording here sounds like clothianidin contents are at the colony level, but Rundlöf et al, 2015 has similar results where clothianidin contents are at the field level.

431 Not clear how many fields were examined here for the comparison of clothianidin contents in individual bees and their honey stomach nectar contents. Was it only three fields from the clothianidin treatment fields, or were there also three additional fields from the control fields?

440 Given that adult worker number is an important metric, it would help to describe briefly how the number of bees were estimated.

440 The inclusion of both the 2013 and 2014 pesticide/fungicide applications is useful. Refer to Extended Data Table 3 of Rundlöf et al, 2015 for the 2013 field plant protection application information.

440 A timeline/phenology table of major events is helpful for this manuscript given the variation in experimental start dates. A Supplementary Table for 2014 events similar to Rundlöf et al, 2015 Extended Data Table 2 (2013 events) would round out the experimental timelines nicely.

446 Not clear what constitutes a “minimum of colonies” that leads to a virus not being screened. Is it excluded if it is not detected at all, or is it another threshold?

453/466 Suggest that authors alphabetize viruses by name or acronym during presentation in text, figures, or tables.

463 This mention of a modified primer would go better with the PCR methods on line 520.

466 Not clear what criteria is meant by a “minimum of colonies”. The threshold appears to be above 0 since 2013 SBPV and CBPV are not included in further analyses.

488 Teflon ... resuspended

496 ... and was eluted into ...

559 Reword this sentence, it's confusing.

595 List which tests used R84 software and which used SAS 9.4 software.

Figures and Tables, Supplementary Figures and Supplementary Tables

Figure 2. Indicate on the graph that colony weight was not measured for the Spring 2014 time point.

Figure 3. Microorganism prevalence at the colony level? Indicate on the graph that microbial prevalence was not analyzed for CBPV, IAPV, KBV, or SBPV. If there are error bars present, shouldn't there be at least a minimum % prevalence (minimum number of colonies with the microbe present)? Right here, several microbes have error bars with 0% prevalence.

Figure 4. Indicate which treatment-control pairs are significantly different on the graph.

Figure 5. SPH51 or SPH5?

Table 1. Use the same significant figures throughout the reported values. Some of the control field workers have detectable clothianidin in their tissues or nectar – isn't this worth commenting on in the results?

Table 2. The period x treatment interaction for the 2014 capped brood is quite close to being significantly different and is significantly different when Biscaya treated colonies are excluded. Is this marginality worth commenting on in the results?

Supplementary Figure 1. Include the year of sampling in the legend.

Supplementary Figure 2. Include the year for the samples. For apiary prevalence, shouldn't any detected organism have at least a 6.25% prevalence (1/16 of total number of apiaries (fields))? 2013 analyses are missing for ABPV, BQCV, and LSV-1, as well as *N. apis*, all of which are mentioned in the results (previously mentioned, 160 and 185). Same comment on error bars and minimum prevalence in the graph as Figure 4.

Supplementary Table 1 Honeybee viruses screened for prevalence at the apiary and colony levels.

Supplementary Table 2 Pathogen and parasite prevalence at the colony level. The 2013 analyses of the ABPV, BQCV, LSV-1, and *N. apis* are missing despite being discussed in the results

(previously mentioned). The changes in *N. apis* prevalence are quite large in Figure 3.

Supplementary Table 3. Immune gene expression.

Supplementary Table 6 388 The fact that the significance/non-significance of these three metrics is dependent on whether Biscaya treated colonies are included or not suggests that these are marginally significant/non-significant effects.

Supplementary Material 108 Reword this sentence

Supplementary Material 125 What are the sources of the chemical standards described here?

Reviewer #2 (Remarks to the Author):

In my opinion, this is an important field study on the effects of Clothianidin on the health of bee colonies.

As for the subject, the study is not particularly original, because others have investigated if and how neonicotinoids can influence bee health under field conditions. In fact, in the introduction, the authors acknowledge (and nicely review) the long list of previous studies on the subject, but state that the significance of the results they present here lies in the recognition and, moreover, the demonstration that the natural homeostatic mechanisms mediating the colony's response to its environment are robust enough to overcome the impairment of individual and social bee life parameters due to chronic exposure to neonicotinoids.

I very much appreciate both the concept that the authors present here (i.e. colony's homeostasis confers to honey bees a further "safety level" such that the unquestionable detrimental effects of certain substances at the individual level can be mitigated), as well as their balanced approach to the subject (i.e. eventually, we can read a study escaping from the two Manichaeistic extremes: no-effect implies no harm / negative effect implies universal danger).

However, in my opinion, the principal merit of this study lies, instead, in its methodological quality. In fact, to my knowledge, this is probably the best ever conducted study in terms of experimental design, depth of the investigation and quality of data analysis, apart from a little detail which I try to explain below. Unfortunately, this little detail could be rather relevant because it affects the principal authors' conclusions; however, I can see a simple solution to this problem that I suggest below.

As reported in the introduction, neonicotinoids can have several sublethal effects. In particular, it has been shown that Clothianidin can impair immunity and indirectly influence the proliferation of Deformed wing virus. According to the proposed mechanism, Clothianidin should affect the abundance of the virus rather than its prevalence. In fact, the authors did not notice any effect of the treatment on DWV prevalence, whereas they did not check any possible effect on abundance, because prevalence was generally low. Accordingly, the lack of any effect of Clothianidin on DWV may well depend on the initial conditions (virus absent and thus impossible proliferation) rather than the natural homeostatic mechanisms mediating the colony's response. Under this perspective, one of the most important conclusions of the study would be unsupported and the present study would just become the last of a long series providing contrasting results on the possible effects of Clothianidin on bee health, because of the variability of the experimental conditions.

However, according to Supplementary table 1, in 2014, there were 44/87 DWV type A positive samples. Why do not the authors re-analyze those samples to assess DWV type A abundance and compare virus titers in treated and control colonies that tested positive for the virus? Clearly, numbers will not be large, but I don't see why these data should not be used for the purpose of the comparison. Then, if a significant effect of the treatment on DWV abundance was noted, in

view of the absence of effects at colony level, the authors could confidently say that the natural homeostatic mechanisms mediating the colony's response allowed the colony to overcome the impairment of individual and social bee life parameters due to chronic exposure to neonicotinoids, and their important conclusion would be fully supported. On the other hand, if a significant effect of the treatment on DWV abundance was not observed, authors could discuss why the effect observed under lab conditions was not confirmed under field conditions.

Following are some more minor comments.

Title 1: in the actual formulation the title suggests an universal lack of effects; this looks a little bit pretentious.

Line 38: the wording "despite seasonal fluctuations" suggests a link between seasonality and the effect under study.

Line 51: ref. 9 does not seem the most appropriate in this case.

Line 52: ref. 10 does not seem the most appropriate in this case.

Line 52: I'd add a reference about the situation in Europe, since Europe is mentioned in the sentence.

Line 54: ref. 12 does not seem the most appropriate in this case.

Line 57: ref. 17 does not seem the most appropriate in this case.

Line 59: a reference may be added at the end of this sentence.

Line 61: ref. 19 does not seem the most appropriate in this case since the cited article deals with neonicotinoids and not all pesticides.

Line 64: ref. 23 does not seem the most appropriate in this case.

Lines 78-89: in my opinion this paragraph should be moved after line 67, at the end of the discussion about the sublethal effects because the ones described here are indeed another sublethal effect of neonicotinoids.

Line 82: there are studies stating that effects can be noted also at low doses.

Line 86-89: the fact that field level evidence is still controversial does not mean that the possible impact is "overemphasized". I don't think that the actual situation of bees (-20% of colonies every year in most countries of the Northern Hemisphere) allows us to underestimate any possible threat to bee health.

Line 92 (and elsewhere): I believe that the corresponding number in the reference list should be added whenever Rundlof et al., is cited, but please check the journals prescriptions.

Line 122: I'd first report the levels found in 2013 and then say that in 2014 they were halved.

Line 151: I understand the attempt to simplify the description, but including Varroa among microorganisms seems a little bit exaggerated to me.

Lines 151-190: nearly 40 rows are dedicated to comment about the possible effect of the treatment on the prevalence of microorganisms (Varroa is not that "micro" actually!). This seems a lot of space for something that, very likely, should not be affected by the treatment. In fact, to my knowledge, chemical treatments should not make pathogens appear in previously pathogen free bees. Therefore I'd suggest to shorten this part of the manuscript which can be distracting.

Line 206: "increased to"?

Line 295: "In 2013"?

Line 362: why the standard error of the mean was used here instead of the standard deviation?

Line 398: "descent"?

Line 426: how long were the pollen traps used for?

Line 491: I'm not sure "Amel" is sufficient as the gene name.

Line 490 (and elsewhere): check if a space has to be put after the number in 65°C.

Line 791: "influence"

Line 795 (and maybe elsewhere): Varroa destructor should go in italic

Supplementary table 1, line 4: remind to the reader why the number of samples does not correspond to the number of colonies.

Supplementary table 5: use "Product" instead of "Compound treatment".

Reviewer #3 (Remarks to the Author):

This is an interesting, thoughtful and well written manuscript that makes a major contribution to the field. These authors previously reported differential impacts of field exposure to clothianidin treated oilseed rape for honeybees, bumblebees and managed cavity nesting solitary bees based on their 2013 data (Rundlöf et al. 2015). Here the authors build on the basis of their 2013 honeybee data, adding a second season of colony performance data, in addition they report findings on pathogen prevalence, loads and associated patterns of immune gene expression. The potential interactive nature of exposure to combined environmental stress factors is a current hot topic, to which this paper provides valuable and novel insights. Assuming the authors can adequately address my comments below, then I think this manuscript would be a good fit for Nature Communications.

A key conclusion here is the apparent resilience of honeybee colonies to showing measurable impacts of clothianidin exposure under these field conditions, even when pathogen infection status is also considered. In their earlier paper, Rundlöf et al. (2015) presented a useful statistical power analysis to show the magnitude of effect size that their experimental design would have been able to detect. Logistical constraints have restricted the degree of replication in this experiment for the 2014 season both at site level, and the number of colonies placed per site. As such, the ability of this experiment to detect effects of a given size will also have been reduced concomitantly. I would like to see the authors provide indications in the manuscript of their sensitivity to detect measurable impacts for the 2014 data. This will help readers to put the apparent absence of detectable impacts of pesticide exposure (with or without pathogens) into appropriate context.

L61: "Another threat to honeybees is the chronic exposure to pesticides used in agriculture¹⁹." Given the wealth of publications in this field, I was surprised to see only the older Blacquiere et al (2011) review cited in support of this statement. I would encourage the authors to cite additional work alongside this review [19] here, e.g. Godfray et al. 2014, 2015.

L65-67: All references here come from studies of honeybees, so I assume the intent of the authors was that this sentence refers specifically to these impacts in honeybees alone. If so, then the authors should specifically refer to honey bees (rather than generic "bees") here. If the points were intended to cover impacts on a wider range of bee species, then they should cite work on other taxa relevant here. For example, impacts of neonicotinoid exposure on foraging success in bumblebees (e.g. Gill et al. 2012; Feltham et al. 2014; Stanley et al. 2016), impacts on learning and memory in bumblebees (e.g. Stanley et al. 2015; Piironen & Goulson 2016), and impacts on worker production and queen production in bumblebees (e.g. Whitehorn et al. 2012; Gill et al. 2012).

L69-70: The authors should update this sentence to reflect the recent EU decision to further restrict use of these three neonicotinoids outside of greenhouses.

L147: The term "in-wintering" is unusual. Could this be replaced with something in more common usage. Perhaps change to "...and were therefore slightly undersized going into winter, early September 2013."

L655: "*Apis mellifera*" should be in italics.

L873: Change from "The data is shown on a logarithmic.." to "The data are shown on a logarithmic..."

References

Feltham, H., K. Park and D. Goulson. Field realistic doses of pesticide imidacloprid reduce bumblebee pollen foraging efficiency. *Ecotoxicology* 23: 317-323 (2014).

Gill, R. J., O. Ramos-Rodriguez and N. E. Raine. Combined pesticide exposure severely affects individual- and colony-level traits in bees. *Nature* 491: 105-108 (2012).

Godfray, H. C. J., et al. A restatement of the natural science evidence base concerning neonicotinoid insecticides and insect pollinators. *Proceedings of the Royal Society B-Biological Sciences* 281: 20140558 (2014).

Godfray, H. C. J., et al. A restatement of recent advances in the natural science evidence base concerning neonicotinoid insecticides and insect pollinators. *Proceedings of the Royal Society B-Biological Sciences* 282: 20151821 (2015).

Piironen, S. and D. Goulson. Chronic neonicotinoid pesticide exposure and parasite stress differentially affects learning in honeybees and bumblebees. *Proceedings of the Royal Society B-Biological Sciences* 283: 20160246 (2016).

Rundlöf, M., G. K. S. Andersson, R. Bommarco, I. Fries, V. Hederström, L. Herbertsson, O. Jonsson, B. K. Klatt, T. R. Pedersen, J. Yourstone and H. G. Smith. Seed coating with a neonicotinoid insecticide negatively affects wild bees. *Nature* 521: 77-80 (2015).

Stanley, D. A., K. E. Smith and N. E. Raine. Bumblebee learning and memory is impaired by chronic exposure to a neonicotinoid pesticide. *Scientific Reports* 5: 16508 (2015).

Stanley, D. A., A. L. Russell, S. J. Morrison, C. Rogers and N. E. Raine. Investigating the impacts of field-realistic exposure to a neonicotinoid pesticide on bumblebee foraging, homing ability and colony growth. *Journal of Applied Ecology* 53: 1440-1449 (2016).

Whitehorn, P. R., S. O'Connor, F. L. Wackers and D. Goulson. Neonicotinoid pesticide reduces bumble bee colony growth and queen production. *Science* 336: 351-352 (2012).

Point-by-point response to remarks by the editor and the reviewers:

Referee #	Number	Remark from the Editor/Reviewers	Response from the Authors
		Referee #1	Response:
1	1	Conceptually, a major design question is whether the two years of experiments are sufficiently independent to be considered separately of one another or whether the two years represent different periods of one timeline (see Data/Methodology 1a). Colonies entering the 2014 treatments had a previous year treatment within the same treatment behind them. Although the authors clearly acknowledge that the second year is a continuation of the treatments, the manuscript should probably be reworked more towards this perspective rather than simply considering the second year to be a separate year.	We agree with the reviewer that the two years of data are not independent and we handle that by analysing data from the two years in a single model with random effects for farm pair identity, farm identity and colony identity. The models contained a three-way interaction between year, treatment and period (before or after oilseed rape blooming) to identify whether the effect of clothianidin varied between the two years. If this was the case, the years were analysed separately, otherwise the effect of clothianidin was analysed in a full model containing data of both years with the random factors accounting for non-independence of colonies (and farms) used in both years. As our experiment was designed as a repeated Before-After-Control-Impact (BACI) study, the effect of clothianidin was measured as the difference between treatments in change over the oilseed rape period, as measured by the interaction between treatment and period. Time could not be treated as a continuous variable because (i) colonies were homogenised before the oilseed rape bloom in the second year (ii) (differential) clothianidin exposure was limited to oilseed rape blooming and therefore interrupted by several months (see further details in response number 7).
1	2	The authors appear to have conducted a narrowly focused pesticide screening (for clothianidin only) on foragers and their loads at one time point only. Assessments of pesticide effects at the landscape level require	The aim in this study was to specifically test the influence of clothianidin exposure and we therefore matched pairs of landscapes based on geographical proximity and land use and then randomly assigned treatments within pair to reduce influence of other factors, including

		pesticide assessments for the entire landscape. This screening may inadvertently miss other agrochemicals present from other landscape sources (i.e. nearby fields outside experimental control) and accumulation of agrochemical residues in food stores and wax (i.e. sources of colony bee exposures, see Data/Methodology 2). These latter agrochemical sources might have chronic effects on bees other than foragers within the hive.	additional agrochemical exposure. Because of this study design, additional agrochemical exposure is expected to be similar between treatments. The main aim of collecting samples of bees and bee-related material was to verify the clothianidin treatment and the bees' use of the oilseed crop. In addition to clothianidin, we include information on the residue in bees, pollen and nectar for four additional neonicotinoids for 2013 and 2014 (Supplementary Table 2). We also provide data on land use in the surrounding landscapes of the study sites, indicating that land use was similar between treatments (Supplementary Table 7). It is true that the bee colonies are most likely exposed to other agrochemicals, but we believe that exposure to clothianidin should represent the greatest difference between colonies at treated and untreated fields. The honeybees foraged extensively on oilseed rape and we generally selected fields without other flowering crops in the vicinity in 2013. See also the methods section (lines 447 - 449), Rundlöf et al. (2015) as well as responses 11, 12 and 13 for further details.
1	3	The number of samples used to determine virus prevalence varied considerably between virus type (see Data/Methodology 3b). This is a fundamental problem with low-incidence viruses, where the number of samples examined has a strong impact on detection rate.	We have performed additional assays so that now all colony samples have been screened for all viruses. Since we now have a complete pathogen dataset at colony level, we have removed the apiary-level virus data, which is now largely redundant.
1	4	Does the experiment simulate realistic field exposure of honey bees to clothianidin from seed-treated oilseed rape flowers through a single crop forage exposure?	We believe that the study setup simulates a realistic exposure of honey bees to clothianidin from seed treatment in spring-sown oilseed rape, as the colonies were placed in the vicinity of a focal crop field and the bees were allowed to forage freely. Our choice to let the bees forage freely allowed the bees to collect pollen and nectar from a mixture of crop and non-crop plants, with potential for reduced exposure by collection of untreated, non-crop pollen and nectar. However, we verify that, despite the free foraging choice, our colonies on average collected

			the majority of their pollen from oilseed rape during crop bloom (53-63% oilseed rape pollen in 2013 and 91-93% in 2014; see lines 140 to 144 in the manuscript). Exposure could also be limited by having only one (treated) field in each landscape and this choice was made for study design reasons, to increase control of the exposure and placement of colonies at the start of the crop bloom. We partly address this by including a second year of exposure.
1	5	Does the experiment adequately capture the exposure of honey bees to clothianidin and other agrochemicals from both treatment fields and the surrounding landscape?	We believe that our experiment reflects the exposure of honey bees to clothianidin and other agrochemicals in Sweden and other similar types of agricultural landscapes, even if our main focus was to simulate a realistic exposure of honey bees to clothianidin from seed treatment in spring-sown oilseed rape, which is the purpose of the study design. Clothianidin was (also prior to the EU moratorium) rarely used in other crops in Swedish agricultural landscapes. The farmers of the focal oilseed rape fields were allowed to use the pest management strategy of their choice in the focal oilseed rape field, apart from other neonicotinoids. This would in reality mean non-use of the neonicotinoid thiacloprid (in Biscaya). The focal farmers and all other farmers in the surrounding landscape were free to use any pest management strategy on the other land.
1	6	Does the experiment adequately measure targeted effects of clothianidin on honey bees?	We believe that we have thoroughly measured aspects relevant for estimating the effects of clothianidin on honey bees under field conditions. We have included two of the three primary assessment endpoints suggested for field studies on honey bees by EFSA (2013): colony strength and over-winter success. We have in addition estimated honey collection and a range of relevant symbiotic microbiota, parasites and viruses critically important for honey bee health. In line with the EFSA guidance document, we have also verified that the bees use the focal crop and the level of exposure to the target pesticide. In addition, the use of rather small colonies which are more vulnerable

			towards environmental stressors (Wu-Smart & Spivak, 2016) makes our experimental setup more sensitive to environmental influences than if we had used full size colonies. EFSA (2013) EFSA guidance document on the risk assessment of plant protection products on bees (Apis mellifera, Bombus spp. and solitary bees). EFSA J. 11, 3295. Wu-Smart & Spivak (2016). Sub-lethal effects of dietary neonicotinoid insecticide exposure on honey bee queen fecundity and colony development. Scientific Reports. 6, 32108.
1	7	The first and second year experiments do not appear to be sufficiently independent because the second year colonies were exposed to treatments for two years. Colonies in the second year are specifically assigned to the same treatments as in the first year. Therefore, the second year appears to be a continuation of the first year treatments, and potentially contains carryover effects from the first year treatment. Because of this, direct head-to-head comparisons of yearly effects seems inappropriate since the 2014 colonies (unlike 2013 colonies) have had previous (and perhaps cumulative) exposure to clothianidin. In essence, the experimental design appears to be a single exposure timeline over two years. Interestingly, the absence of differences between treatment groups actually strengthens the results because exposed colonies endured two successive years of exposure without apparent effect.	We agree with the reviewer that the data from the two years are not completely independent as the same colonies and farmers were involved. Therefore, we analysed the data now in a model containing data of both years and random effects to control for non-independence of farms and colonies. The study was designed as a repeated Before-After-Control-Impact experiment to measure the effect of exposure to clothianidin seed-coated oilseed rape in two flowering periods with colonies that were homogenised before exposure in both years. The effect of clothianidin exposure in the two years was analysed simultaneously with an interaction between seed-treatment and bloom, which indicates whether the change over the oilseed rape bloom differed between treatments (clothianidin-exposed or untreated control). In addition, a three-way interaction between seed-treatment, bloom and year was included in the model to see whether this effect differed between years. If this was the case (i.e. $P < 0.05$ for the three-way interaction), we analysed the data of the two years separately to identify how clothianidin exposure affected honeybee colonies in each of the two years. As the reviewer pointed out, we exposed

			colonies in both years to the same treatment (except for two control colonies assigned to treated fields). This means the control colonies in 2014 were not exposed to clothianidin in 2013. Therefore, effects of clothianidin in 2014 could not be ameliorated by potential effects in 2013. The results of the second year are therefore also valid if the two years were analysed separately. As we used a BACI design in our study, with equalization of colonies before placement in the field each year, analysing the dataset as one timeline is not appropriate as it would not account for seasonality and equalization. In addition to the reviewers' suggestions, we changed the terminology to prevent confusions. We changed "Period" to "Bloom" and "Treatment" to "Seed-treatment".
1	8	Critical to how this might be resolved is whether the colonies were scrambled and "equalized" between the first and second years. At the beginning of the second year, the colonies are described as being of similar small size and strength. However, it is unclear whether this similarity in size occurred due to "equalization" of colonies within a treatment group (via frame, worker, and brood swaps) or simply as the state of colonies in June 2014.	The colonies were equalized again in 2014 and we included more detailed information now in the manuscript (see lines 516 - 518) concerning the establishment and composition of the colonies at the start of each year.
1	9	The honey bee colonies used here were exceptionally small compared to most managed colonies, even nucleus colonies. Wu-Smart and Spivak (2016) recently found that smaller colonies are more susceptible to neonicotinoid exposure than larger colonies. The authors should justify the reasoning behind the use of such small colonies. Concerns about the vulnerability of such small colonies is partially offset by the reasonably low overwintering mortality of	The choice of initial colony size was determined by honey bee expert Ingemar Fries and an experienced local beekeeper so that the colony 1) would not outgrow the colony box during the summer, to reduce the need to add material during the experiment, and 2) be large enough to have a chance to survive the winter. We have now clarified this reasoning in the manuscript under Methods together with the statement that the experiments were initiated with splits prepared at the end of May, which consisted of two frames of

		colonies in both treatments; however, the effects observed on these small colonies may not be representative of most honey bee colonies.	brood, two frames of honey, about a pound of bees and a laying queen (see lines 486 to 488). Moreover, as the reviewer states, smaller colonies might be a more sensitive way to measure effects of environmental change than full size colonies. The fact that we could not detect negative impacts of clothianidin on small honey bee colonies would support that we would most likely also not find negative effects on larger colonies, with even greater compensating capacity.
1	10	Not clear why 1-2 year old experimental queens were used to establish the colonies in the first year, given the widespread problems with queen failures among older queens. The older starting age and age range seems to introduce unnecessary variability into the experimental design, as older queens (older than 1 year) commonly fail for various reasons unrelated to pesticide exposure. The apparent exclusion of colonies headed by queens that were 2 years old (at the beginning of the first experimental year) before the second experimental year observations seems like an avoidable loss of colony numbers.	Indeed, queen failure is one of the most mentioned causes of honeybee losses (vanEngelsdorp et al., 2011; Pettis et al., 2016) and in commercial beekeepers' re-queen colonies usually annually or every 2 years (Neumann & Blacquière, 2017). In Sweden queens are normally reared during June to ensure enough drones are present for optimal mating. This coincides with the experimental time-frame, wherefore no newly reared queens could be used. This meant that most queens were 1 year old at the start of the experiment, with 12/96 two-year old queens. This allowed us to use the bulk of the (1-year old) queens for both consecutive years, while also investigating the effects of exposure on a smaller subset of 2-year old queens during 2013. We have added an extensive section analysing the differences between the 1-year and 2-year old queens in swarming, supersedure and winter mortality, confirming the greater failure rate among older queens but also the minimal effect of clothianidin exposure on these rates, or the age-related differential. The exclusion of the 2-year old queens at the start of the second year had no effect at all, since very few of these were left by spring 2014. Moreover, the entire 2014 experiment was now populated by 2-year old queens, with similarly little effect of clothianidin exposure on colony and pathogen parameters as with the same queens in 2013. Neumann, P., Blacquière, T. (2017). The

			Darwin cure for apiculture? Natural selection and managed honeybee health. Evolutionary Applications, 10, 226-230. Pettis, J. S., Rice, N., Joselow, K., vanEngelsdorp, D., & Chaimanee, V. (2016). Colony failure linked to low sperm viability in honey bee (Apis mellifera) queens and an exploration of potential causative factors. PLoS One, 11(2), e0147220. doi:10.1371/journal.pone.0147220 van Engelsdorp, D., Hayes, J. Jr., Underwood, R.M., Caron, D., & Pettis J. (2011). A survey of managed honey bee colony losses in the USA, fall 2009 to winter 2010. Journal of Apicultural Research, 50, 1–10.
1	11	Were nearby fields treated with agrochemicals before or during the experiment, and did experimental bees collect these agrochemicals? Bee colonies may be contaminated by agrochemical applications originating outside the treatment fields (i.e. the surrounding landscape). Researchers often have limited control over which agrochemicals are applied to fields within flying radius of their experimental colonies. The authors estimation of 4 km as a sufficient distance to separate colonies probably holds true during a strong bloom when bees forage near their colonies. However, foragers will venture much further than this distance when forage is less available. This is evident by the presence of clothianidin in 2013 control colony bee and nectar samples – pesticide contents that indicate that these fields are probably not completely separated.	Unfortunately, we do not have information on treatments of agrochemicals within the surrounding area of our focal fields. However, the residues of other neonicotinoids in the pollen, nectar and bee samples were very limited (Supplementary Table 2). In addition, the surrounding areas of the focal fields were inspected for presence of flowering crops (see lines 447 – 458, Supplementary Table 7), including other oilseed rape fields. As the study was set up before the EU moratorium was in effect, there could have been a possibility that the bees were exposed by neonicotinoids from surrounding fields. Potential field sites had been excluded on account of the nearby presence of other oilseed rape fields or fields with red clover, which is known to be very attractive to bumblebees in 2013. In two cases, we accepted the presence of a single other oilseed rape field in the surrounding area, in order to complete the desired landscape-level replication. Furthermore, residue analysis of the 2014 pollen, nectar and bee samples from untreated fields were all below the detection limits for clothianidin, confirming the appropriateness of the design of our study system. In 2014, additional spring-sown oilseed rape (1 -

			13 ha) within a radius of 2 km were found in half of the landscapes (see lines 456 and 458). As the colonies were only placed next to the study fields during oilseed rape blooms, we believe that the honeybees did predominantly forage within a 2 km radius around their hives. This is underlined by the observation that most pollen originates from oilseed rape. Nevertheless, we agree that information of the surrounding landscape in 2014 could be of interest and this is now included in the Supplementary Information (Supplementary Table 7).
1	12	Were the agrochemical screenings broad spectrum or were these screenings specifically targeted only to the experimental plot pesticides? It is not clear from the text whether colonies were screened for agrochemicals other than the chemicals applied within the experimental plots. Non-treatment agrochemical exposures would likely occur at some sites but not others – if so, deleterious effects from other non-treatment agrochemicals might increase variability in outcomes and metrics between sites. Neonicotinoids have recently been shown to act synergistically with some other agrochemicals. Broad spectrum screenings at each apiary would clear this question up.	We specified that the screenings were performed for clothianidin as well as four other neonicotinoids that were commonly applied to plants in Sweden before the moratorium (see line 532). We did not screen for further agrochemicals. Indeed, other agrochemicals like fungicides could have impacted honeybee health and development directly or synergistically with neonicotinoids, but the experiment was not set up to test such effects. However, as fields with other mass flowering crops in the surrounding were excluded in 2013 and in 2014 collected pollen mainly consisted of oilseed rape pollen (>90%), we think that exposures from other fields had only minor influence in this field study.
1	13	The pesticide screening methods used here provides a good snapshot of pesticide contamination of foragers (worker tissue, corbicular pollen pellet, and honey stomach nectar contents) during peak oilseed rape bloom, but do not provide a comprehensive screening assessment of materials most likely to lead to long-term contamination of the rest of the colony (stored pollen, honey stores, and wax). The authors' reports on pesticide screening of worker tissue, corbicular pellet, and honey stomach nectar appear to be limited	The pesticide residue quantification was, alongside pollen analysis of plant origin, mainly intended to verify the difference in exposure to clothianidin and use of the oilseed rape crop between treatments and secondly to confirm that there were no systematic differences in exposure to other neonicotinoids. Indeed, it would have been very interesting to analyse material from the hive to test for potential long-term contamination of the hive. As Wu et al. (2011) demonstrated, pesticide residues in combs can lead to delayed honeybee larval development and adult emergence as well as reduced longevity of

		to the pesticides present from seed and field management applications during peak rape oilseed bloom. Researchers commonly perform limited broad spectrum pesticide screenings of hive materials (stored pollen, honey, wax) to detect agrochemicals brought in from the landscape environment to the colony. Testing of food materials would also allow better comparisons to structured lab and field studies where the food pesticide exposure are well known. Broad spectrum pesticide screenings of these materials would allay these concerns.	adult honeybees (Wu et al., 2011). Some other studies (e.g. Sandrock et al., 2014) have tested hive material from exposed colonies but could not detect any residues. Even if these additional analyses would have given insight into the fate of residues in the hive, and for studying the effect of pesticide accumulation in the hive, we mainly focused on the direct exposure to clothianidin from blooming oilseed rape fields. Unfortunately, we do not have any hive material left to perform additional pesticide screening even though we highly appreciate the suggestion. Sandrock, C. et al. Impact of chronic neonicotinoid exposure on honeybee colony performance and queen supersedure. PLoS One 9, 1–13 (2014). Wu JY, Anelli CM, Sheppard WS (2011). Sub-Lethal Effects of Pesticide Residues in Brood Comb on Worker Honey Bee (Apis mellifera) Development and Longevity. PLoS ONE 6(2): e14720. doi:10.1371/journal.pone.0014720
1	14	Were other oilseed rape fields located near the experimental fields, and if so, were they in bloom during the experimental period or treated with pesticides? Rundlöf et al., 2015 notes that in 2013 other oilseed rape fields were located as close as 2 km from the experimental fields. The presence of nearby non-experimental oilseed rape fields would add to the oilseed rape pollen detected in the pollen composition analysis and inflate estimates of treatment plot pollen exposures.	We generally selected fields for the experiment that did not have oilseed rape or red clover fields in the surrounding landscape (i.e. within 2 km) in 2013. In two cases, we accepted the presence of a single other oilseed rape field in 2013. Residue analysis of honeybee-collected pollen and nectar as well as bee samples showed that colonies at treated sites were clearly exposed to clothianidin, while colonies at untreated sites were practically not exposed to clothianidin. In 2014, the experiment consisted only of fields from farmers that took already part in the study in 2013 and planned to grow oilseed rape again. Therefore, we had no control over the presence of additional spring-sown oilseed rape in the surroundings of the focal fields in 2014. There was additional spring-sown oilseed rape (1 – 13 ha) within a 2 km radius of half of the focal fields in 2014 (see lines 455 - 457).

			See the methods section, Rundlöf et al. (2015) and response 11 for further details.
1	15	The Varroa mite sampling method is underpowered and targets older workers that are not preferred as hosts by phoretic mites. Techniques for estimates of phoretic mites commonly sample 300 workers (not 100 workers), as the incidence of phoretic mites on workers in healthy colonies is often below 10% infestation (see Lee et al., 2010 J Econ Entom 103: 1039). Such sampling is also usually performed on nest worker bees as Varroa mites very much prefer younger adult bees engaged in nurse and housekeeping activities over older bees engaged in outside activities such as foraging. However, the Varroa infestation rates of entrance bees is probably correlated with infestation rates of nest bees.	To investigate the prevalence and abundance of pathogens and parasites we collected 100 bees from each colony. These samples were taken from the outer comb covered by honey bees, which consists of a representative mixture of forager and house bees when sampling on a normal foraging day (van der Steen et al. 2012). This sampling technique had not been sufficiently enough described in the previous version of the manuscript. We therefore added more detailed information in the Methods section (see lines 570 – 573). This sample is adequate for the pathogen analyses (Pirk et al. 2013) but not ideal for estimating Varroa infestation rates, as pointed out by the reviewer (see also Dietemann et al. 2013). The reason for the small sample size was partly not to burden the (small) colonies too much and partly because the sample was needed for multiple analyses. During 2014, with higher infestation rates, the problem of the small sample size of bees becomes less acute (since in Poison-distributed parameters, the accuracy of an estimate is largely determined by the numerator, rather than the denominator). The extent to which these estimation inaccuracies affect the overall conclusions is mitigated in part by the large number of colonies involved, increasing the power of the analyses. Dietemann, V. et al., 2013. Standard methods for varroa research. Journal of Apicultural Research, 52(1), pp.1–54 Pirk et al. 2013. Statistical guidelines for Apis mellifera research, Journal of Apicultural Research, 52:4, 1-24 van der Steen et al. 2012. How honey bees of successive age classes are distributed

			over a one storey, ten frames hive. Journal of Apicultural Research 51(2): 174-178
1	16	The limited testing (one year only with different sample numbers) of entrance bees for most viruses and immune gene expression is highly problematic. These parts of the experiment would be much more robust if two years of complete data had been obtained and comparisons were made with similar numbers of samples. Especially troublesome are comparisons of virus prevalence for low-incidence viruses using different sample sizes. One would imagine that the ability to detect a low-incidence virus would be highly dependent upon how many bees were examined. In this case, the sample numbers vary considerably between viruses. Likewise, the argument that the absence of the less common viruses in the first year justifies their exclusion from second year screening is not convincing given what is known about these viruses. The prevalence and tissue abundance of many honey bee viruses is not consistent but varies erratically seasonally, annually, and developmentally. I would be less than confident that one year's single timepoint screening fully captures virus expression given the seasonal, year-to-year, and colony-by-colony variation.	We performed additional assays to obtain a complete dataset at colony level for both years and for all regarded viruses. All assays were conducted on samples of 60 bees taken from within the hives.
1	17	The primary concern regarding statistics is whether the two years experiments were actually independent by year or represented one two year timeline (see Data/Methodology 1a). If the two years were not independent, then the authors need to rework their year-to-year comparisons.	We agree with the reviewer that two years are not completely independent. Therefore, we analysed data of both years in a single model that reflected the BACI study design and included random effects to control for non-independence of fields. For details see Response 7.
1	18	A second question is whether the experimental design was sufficiently robust to support conclusions of a lack of significant effects of clothianidin. The number of colonies	We have now included a power analysis for the BACI effect (seed-treatment x bloom x year) on the number of adult bees and the amount of brood (seed-treatment x bloom) for each year separately as well as

		was adequately robust for the statistical comparisons in 2013, but more marginal in 2014 (24 colonies (clothianidin), 16 colonies (control)). The 2014 colony numbers may not have been sufficient to detect low incidence virus infections (see Data/Methodology 3b). However, if the purpose was to identify moderate to high prevalence viruses for further comparisons (which it appears the authors intended), the colony numbers were sufficient for this purpose. The text could simply reflect this end more clearly.	for the treatment effect on the amount of honey produced during the oilseed rape blooms. We analysed statistical power for the number of adults and the amount of honey produced in full models containing data of both years. For the amount of brood, we analysed power separately for the two years as the three-way interaction (year x seed-treatment x bloom) was statistically significant. We showed that power reduced from 2013 to 2014 for the BACI effect on the amount of brood, but in both years effect sizes below 11% could be detected with 80% power for the number of capped brood. Using the full model containing both years (number of adult bees), we could detect an effect size for the number of adult bees of below 5% with a power of 80%, which is even lower than the requirements of an effect size <7% set by EFSA (2013) (see lines 746 – 766). Since we screened now for all viruses in both years and detected some of them in considerably higher prevalence in 2014 than in 2013, we analysed the abundance of more viruses than in the previous version of the manuscript. All pathogens, parasites and bacteria that we analysed for their abundance were more than 10 times detected per year (see Supplementary Table 1). The fact that we detected mostly an even higher prevalence of viruses in 2014, we believe that we have a robust study system, with sufficient colony number even in the second year to identify impacts of clothianidin on honeybee health. EFSA (2013) EFSA guidance document on the risk assessment of plant protection products on bees (Apis mellifera, Bombus spp. and solitary bees). EFSA J. 11, 3295.
1	19	All error bars in the figures and supplementary figures were defined clearly in the figure legend. For two	The error bars represent confidence intervals for two-sided binomial tests. Confidence intervals should in 95% of the

		figures displaying microbe/parasite prevalence (Figure 3 and Supplemental Figure 2), error bars are present without any apparent % prevalence. Shouldn't the average prevalence be above 0% in these cases?	cases cover the true population value. Since we conducted a two-sided test, the error bars should show the upper 2.5% of (population) values of a binomial distribution (and the lower 2.5% of values) that would yield the observed result. For the cases where we observed no positive samples, the upper limit of the confidence interval shows the maximal proportion of infected colonies of the population that would yield in no more than 2.5% of cases zero detection in our sample. So, if the true proportion of infected colonies is p and our sample size is n then $(1-p)*n = 0.025$ should be true. We can solve this for p: $p = 1 - 0.025/n$. For example, for ABPV in 2013 before the oilseed rape bloom this yields in each treatment group ($n=48$) 0.074 or 7.4% as shown in Figure 4.
1	20	Most of the statistical tests were sufficiently robust and presented clearly in the results and tables except for the sections on microbe prevalence and amounts. These sections are sometimes confusing because it is unclear which comparison is being made. Sample comparisons differ by year (2013/2014) and unit of comparison (colony/apiary (field)). Confusion from poor labeling is further exacerbated by the fact that similar sized sample sets were not analyzed for all microbes. Only the most prevalent microbes had full sample sets analyzed for the 2013 apiary, 2013 colony, and 2014 colony comparisons. The authors need to thoroughly label text, figure and table headings to ensure that readers know exactly which comparisons are being made	To avoid some of this confusion we have now screened for all target organisms at colony level and removed the then redundant apiary-level analyses. We have rewritten large parts of the results section and changed the figures and table headings for gene expression to clarify that this analysis was only done in 2013 and on apiary level.
1	21	For the most part, the results obtained here support the conclusions laid forth in the manuscript. One main concern with experiments that find a lack of significant treatment effects is whether the design was sufficiently	We agree that the uneven sample sizes make it harder for the reader to understand what was done and found when. We have therefore conducted additional laboratory analyses and screened for all viruses for all time points.

		robust to adequately test this case. Unfortunately, the sections comparing microbial prevalence (particularly low-incidence viruses) are based on uneven sample sizes and are quite complicated. The fact that some microbes were assessed only in one year and some in two years becomes problematic if not well framed in the text. Although certainly unintended, the reader may be left with an impression that a given microbe was not present in both years when it was sampled only once. These microbe prevalence sections could be improved either by testing all microbes with equal sample numbers or making comparisons with more explicit note of how many samples the comparisons were based on. These sections need reworking for clarity.	
1	22	One of the four significant impacts of clothianidin (2013 – greater seasonal decrease in S. alvi in clothianidin) is understated both in the abstract and the rest of the manuscript. In a related problem, the authors do not adequately discuss what the impacts of differences in the gut microbes S. alvi and G. apicola might mean. Would lower amounts lean toward microbial dysbiosis? Be indicative of different worker age structures? The role of these gut microbes in bees is a fairly important topic worth commenting on now.	Yes, indeed this was not thoroughly discussed in the previous version of the manuscript. However, the modified analysis method, with including both years in the same model, testing for a difference in BACI effect between years, S. alvi is no longer influenced by the interaction of seed-treatment and bloom.
1	23	One aspect of the conclusions that is somewhat problematic is the generalized statement that clothianidin does not impact pathogen levels. While most metrics were unaffected by the clothianidin treatment, some aspects of colony microbes were in fact affected by clothianidin exposure and should be presented as such. The manuscript could have just as easily been titled “Clothianidin field exposure impacts honey bee pathogen, pathogen, and	We agree that the title could be misleading, as it states that clothianidin had no impact at all on colony health parameters. With new results due to a the use of a full model, excluding swarmed colonies and more screenings, we found rather positive effects on honeybee colonies, which might be due to different reasons like overcompensation. We therefore reworded the title to “Clothianidin seed-treatment has no detectable negative impact on honeybee colonies and their pathogens”.

		microbe levels, but not colony-level performance or immune gene expression” based on what presented about 2014 G. apicola , BQCV, and Varroa levels in the abstract and Table 3.	Indeed, we detected only few effects (mostly positive), suggesting that at colony level, honeybees are relatively robust to the effects of clothianidin exposure in real-world agricultural landscapes. We rephrased the abstract, results, discussion and conclusion, to describe our findings more precisely.
1	24	Some semantic problems occur in the results text due to confusion between time (period or before/after oilseed flowering) and time (between years).	We have edited the manuscript to clarify whether differences between periods of the same year (i.e. pre- and post-exposure) or different years were regarded. Furthermore, we changed terminology in the tables. We use “Bloom” instead of “Period” now.
1	25	line 163 Lacks supporting analysis. The 2013 ABPV comparisons are missing from Supplementary Table 2 (numbers too low for analysis?) and 2014 ABPV analysis does not show a significant effect by Period, Treatment, or the interaction. DWA-A increases significantly in prevalence through the Period in both 2013 and 2014 as shown both in the statistical analysis and graph.	We rephrased most parts of the result section. We have decided to statistically analyse only target organisms and years with an effective sample size (i.e. the number of samples with the less frequent outcome of the presence/absence variable) > 10 colonies per year. This is stated in the Method section (see lines 683 and 684). We state that no effect of clothianidin exposure could be found for ABPV prevalence (see lines 219 - 222). Result for DWV-A prevalence is rephrased in lines 225 to 227.
1	26	line 166 Conflict between results text and Supplementary Table 1. ABPV is not absent in 2013 at either the apiary or colony level. Perhaps “largely absent” would be better?	Yes indeed. Revised as suggested (line 219).
1	27	In183/Supplementary Table 2. Missing 2013 N. apis , BQCV, and LSV-1 prevalence data despite descriptions in results text.	For statistical reasons we could not analyse prevalence for pathogens with a low effective sample size (less frequent outcome of the presence/absence variable). We have decided to statistically analyse only target organisms and years with an effective sample size > 10 colonies. The very high prevalence of BQCV and LSV-1 meant that the effective sample size was low for BQCV and LSV-1, therefore we could not test the impact of clothianidin on these viruses (Supplementary Table 1). We have now described this in the text (see lines 214 – 217, 683 & 684), as well as for N. apis in 2013

1	28	line 199 Conflict between results text and Figure 4. For both 2014 clothianidin and control field colonies, the Varroa infestation rates appear to increase, but control field colonies levels appear to increase more.	Yes, indeed. We have now deleted the statement about Varroa infestation rate, as it was not significant in the full model.
1	29	line 216 Conflict between results text and Table 3. There was a treatment x period interaction for 2013 S. alvi , not a treatment effect.	Indeed. However, we rewrote this part of the result section as results changed with using a full model.
1	30	The experiment would have been more robust if the post-bloom assessments had been extended beyond the end of the bloom. Several recent publications have noted chronic effects of neonicotinoids some time after the initial exposure.	Indeed, the oilseed rape bloom is very short and long-term effects might not have been detected in the first year. However, we believe that our study design examined also potential long-term effects as colonies were exposed in two consecutive years to the same treatment (i.e. either clothianidin-treated or untreated) oilseed rape and we present data from two years both before and after the oilseed rape bloom plus a spring colony assessment for colony strength in 2014 and winter survival during winter 2013/2014, we think that long-term effect could have indeed been detected.
1	31	A better understanding of the full agrochemical exposure of these colonies (aside from clothianidin) is essential for a landscape study, given that colonies could easily be contaminated by outside agrochemical exposures. Likewise, the authors could expand on descriptions of alternative landscape forage, particularly oilseed rape forage.	The aim in this study was to specifically test the influence of clothianidin exposure and we therefore matched pairs of landscapes based on geographical proximity and land use and then randomly assigned treatments within pair to reduce influence of other factors, including additional agrochemical exposure. Because of this design, additional agrochemical exposure is expected to be similar between treatments. We include information on the residue analyses of bees, pollen and nectar for four additional neonicotinoids for 2014 (Supplementary Table 2). We also provide data on land use in the surrounding landscapes of the study sites (Supplementary Table 7).
1	32	Pesticide analysis of food materials from inside select colonies would capture dietary pesticide exposures of the majority of colony bees, given that most food consumption other than forager nectar consumption for flight occurs with food stores.	Unfortunately, we do not have any hive material left to perform additional pesticide screening even though we highly appreciate the suggestion. For a detailed response see Response 13.

		Neonicotinoids such as clothianidin persist for months once stored under dark conditions in honey stores. Such an analysis would also allow better comparisons with the existing literature as well.	
1	33	A major problem with the virus prevalence analyses is that different numbers of apiary and colony samples were used to compare virus infection rates (see Data/Methodology 3b). Given that the ability to detect uncommon virus is largely dependent on the number of samples compared, this inequality is problematic. The authors could address this problem by analyzing sufficient additional virus samples to make comparisons with similar numbers of pooled apiary or pooled colony samples. Alternatively, the presentation could be reworked to simply present the first year virus screenings as just that – and avoid erroneous perceptions that viruses are being compared head-to-head in both years.	We agree and have now analysed all viruses for all colony samples to have a complete and robust dataset.
1	34	75 A recent publication by Odemer et al. 2018 Ecotox. also found a lack of sublethal effects with clothianidin exposed bees.	We added the suggested reference.
1	35	80 Dussaubat et al., 2016 Sci. Reports 6: 31430 found synergistic effects between Nosema and queen survival.	We added the suggested reference.
1	36	322 See Meikle et al., 2016 PLOS One. Sublethal effects of imidacloprid on honey bee colony growth and activity at three sites in the United States for persistence of neonicotinoids in food stores.	We have sampled nectar and pollen directly from stomachs of bees trapped at hive entrances. Therefore, the plant residues should be freshly collected by honeybees and neonicotinoid concentrations should be unrelated to the persistence of these in food stores. In contrast, year-specific differences in humidity and temperature that affect the persistence of neonicotinoids in soil may explain differences in clothianidin concentrations between years.
1	37	327 "... reported in other studies." See Rolke et al, 2016 Ecotox. 25: 1691	We added the suggested reference (see line 391).

1	38	One section that could use expansion concerns the two core gut microbes, G. apicola and S. alvi. These two microbes need to be considered separately from the parasites and pathogens because these are associated with a healthy microbiome. That being said, relatively little is known at this point as to their interaction with other colony stressors, although it is very much a hot topic of future interest. (see Collison et al., 2016, Biol. Rev. 91: 1006)	We expanded the introduction including the symbiotic gut microbes G. apicola and S. alvi, which is indeed a very interesting and hot topic. We also tried to address this topic more in the discussion.
1	39	Since the manuscript is a continuation of a previously published experiment (Rundlöf et al., 2015), the authors face the dilemma of what to include from the previous manuscript and what to leave out in the current manuscript. The authors could readily incorporate more methods and results from Rundlöf et al., 2015 (recap?) to ensure that this manuscript is stand alone in its presentation to readers. By comparison to its companion manuscript, the Materials and Methods section of this current manuscript is less descriptive, less specific, and lacking in some key details. Several parts of the M & M could be rewritten to provide a clearer narrative of what was performed, especially regarding the experimental timeline.	We agree that the manuscript should be able to stand alone. However, as the reviewer states we face the dilemma of what to include and what to leave out. We have now decided to present some data that was already included in Rundlöf et al. (2015), for example in Supplementary 2, 7 and 8. Furthermore, we have tried to be more specific in the methods section. We have now also included a timeline to make it easier to understand, what was done when (see Supplementary Fig. 1).
1	40	Some metrics are not clearly or consistently described. For example, estimates of adult bees are variously described as number of adult bees and references are made to colony weight. How did the authors measure both of these metrics? What exactly does colony weight measure and how was it adjusted for weight differences before the pesticide exposure/oilseed bloom? While the methods behind these metrics may be in the references, these metrics could be better detailed in the current	We agree that some metrics were not clearly described in the methods section as in Rundlöf et al. (2015). As suggested we added some key parts that were missing before (e.g. equalization of colonies in 2014 (see lines 516 – 520), reasoning for usage of small colonies (see lines 487 – 489), residues analyses (see lines 525 – 541) or description of number of adult bee assessment (see lines 550 - 552)). Instead of reporting the change in absolute colony weight, we have now estimated the amount of honey produced over the

		manuscript. Areas that might need more details are listed in specific comments below.	oilseed rape period. We believe this is a biologically more easily interpretable measure. For this, we subtracted the pre-exposure weight from the post-exposure weight of the colonies; then we subtracted the weight of added empty frames and added the weight of removed frames full of honey. We detail this in the methods section (see lines 553 – 560).
1	41	The authors need to be more clear in their presentation of microbial and Varroa prevalence and amounts in the results text, Figure 2, Figure 3, and Supplementary Figure 2. As currently presented, the results can be very confusing to the reader given the variety of comparisons made. Data is presented for 2013/2014 samples and at the colony or apiary level, and some analyses were performed only in one year (many viruses and immune gene expression for 2013 samples). I would strongly suggest that the authors tightly label their results by year and by apiary/colony analysis levels.	We agree that the results were not very clearly presented in the previous version of the manuscript. As the reviewer suggested, we performed additional analysis of the missing viruses on colony level. We excluded the now largely redundant analyses at the apiary level, as this was originally done to identify prevalent viruses, which could then be screened at colony level. Now, we present the full dataset of 13 viruses in both years and we hope that the result section is less confusing.
1	42	I would also strongly suggest that the authors indicate table or figure sections where samples that were not analyzed during a given year (i.e. many virus and immune gene expression analyses were only analyzed in 2013) with N/A (not attempted) rather than n.s. (not screened). The latter acronym could be easily confused with n.s (not significant). Also, the absence of analysis should be indicated directly on the table or figure by N/A due to the similarity between zero values and skipped analyses. An asterisk on the left side next to the pathogen name is not as effective in indicating that the second year was not attempted.	We have now screened for all viruses consistently in both years. Immune gene expression was only determined in 2013, but presented separately, so that we do not need any more an acronym for missing laboratory analyses. We have now indicated directly on the figure, years in which target organisms were not statistically analysed by the acronym (N/A = not analysed).
1	43	Explicit distinctions need to be made between colony prevalence and apiary (field) prevalence in both the text and the figures/tables. Also, label parts of graphs or tables where no	See Response 42.

		analysis was performed with a more prominent marking (perhaps N/A for not attempted, rather than n.s. which resembles not significant). Otherwise, it is hard to distinguish between zero values and analyses that were not performed.	
1	44	On two figures displaying microbe/parasite prevalence (Figure 3 and Supplemental Figure 2), error bars are present without any apparent % prevalence. Shouldn't the average prevalence be above 0% in these cases?	See Response 19.
1	45	I would hesitate to refer to two timepoints over one forage crop in two years as covering "seasonal" effects or a "season". Most longitudinal studies of this kind examine colonies at multiple timepoints to obtain a fuller perspective. A better set of terms would be "pre-forage/post-forage" or some variant centered around the bloom. But the authors should still consider effects from a seasonal perspective.	We tried to be more clear and used more often year/years and pre- and post-exposure.
1	46	Also, "abundance" would be a better term than "amount" to describe the number of pathogens or parasites detected.	We replaced the term "amount" with "abundance" as suggested.
1	47	Abstract Succinct and generally to the point. The authors need to comment on the finding that S. alvi abundance seasonally decreased more in clothianidin colonies than control colonies.	On the basis of the new statistical analysis S. alvi abundance is not significantly affected by clothianidin exposure. However, we included the additional findings in the abstract.
1	48	The introduction nicely sets up the the experiment and its rationale – first by focusing on bee stressors individually, then considering interactions, and finally by noting the lack of field-relevant experiments. The authors need to expand on the importance of the two gut microbes S. alvi and G. apicola – why include these in here, and is there any penalty to bees if these are deficient?	We have added information on the importance of S. alvi and G. apicola in the introduction (see lines 77 - 81).
1	49	119 Include the 2013 pollen species composition analysis here with the	We have included additional information of the pollen species composition in 2013

		2014 analysis, or at the very least, briefly recap the findings. The oilseed rape proportion of the pollen samples is very different between 2013 and 2014, although not between the treatments for each year. Interestingly, the 2013 corbicular pollen pellets contains less oilseed rape pollen than 2014 pollen, but results in higher clothianidin exposure as measured by bee tissue and nectar contents.	in the result part (see lines 140 - 145). We also discuss this finding briefly in the discussion (see lines 382 - 390).
1	50	122 Please comment on the presence of clothianidin residues in 2013 control field bees (Table 1). In 2013, clothianidin residues were detected in workers collected from control field colony entrances. Such a finding is not unreasonable at all for a field study of this kind, but needs some explaining. What does this mean for isolation of colonies near treatment fields? Colonies likely forage predominantly from their assigned fields, but also probably forage a little from the other treatment field and other fields in the surrounding landscape.	We included these findings in the result part (see lines 149 - 153).
1	51	123 Explicitly state that clothianidin residues are higher in bees from colonies in clothianidin-treated fields than bees from colonies in untreated control fields. This result is obvious from Table 1, but it should be presented in the results.	Revised as suggested.
1	52	124 ... approximately half of ...	Revised as suggested.
1	53	138 Field-to-field variation in clothianidin exposure is important and likely a reflection of landscape forage differences.	We mention this now in the discussion as well as the possibility of differences in the uptake of clothianidin within the plants (see lines 395 – 397).
1	54	139 Explicitly state period effects for the 2014 colony development in the results section. The strong effects of period on two of the three metrics is obvious in Table 2, but needs to be presented in the results.	We have now stated period effects revealed by the new statistical analysis (lines 162 – 184).
1	55	138 For the 2014 Before/After results (Table 2), the differences in the capped brood number between treatments is only marginally non-	In the new analysis we excluded colonies that swarmed during the experiment, to be consistent with Rundlöf et al. (2015) (see method section; lines 684 – 690) and

		significant ($p=0.066$). Notably too, the number of capped brood cells from before to after observations decreases in clothianidin treatment colonies but increases in control colonies. Do the results come out differently if pre-bloom (Before observation) metric differences are compensated for for?	because swarming has a large effect on colony development. Excluding colonies that swarmed from the analysis qualitatively altered some results including the number of capped brood (see Supplementary Table 12). The analysis method of before-after control-impact (BACI) account for differences during pre-exposure assessment as the difference between treatments in the change (in the amount of brood) during the oilseed rape bloom is regarded rather than the difference at a specific time point.
1	56	147 overwintering rather than in-wintering?	Revised as suggested.
1	57	163 Break this sentence into two parts.	We rewrote the result section.
1	58	161 ... highly prevalent in both colonies and apiaries, but were not affected by either treatment or sampling period.	We rewrote the result section.
1	59	171 ...the prevalence of V. destructor increased in colonies from ...	We rewrote the result section.
1	60	173 Difficult to comment on the prevalence of CBPV and SBPV when these viruses were analyzed in just one year. Be careful about giving the impression that the samples were examined for these viruses for more than one time point/collection set.	We present now a full dataset with viruses screened at all time points.
1	61	179 also Figure 3	We have now deleted this sentence, as we detected all viruses by screening all samples.
1	62	187 Is there any way to compensate for this initial (before bloom) difference in 2013 Nosema ceranae prevalence? This would seem to disrupt further analysis if left uncorrected.	Our statistical analyses compensates for differences that existed before exposure to oilseed rape. The study was designed as a Before-After-Control-Impact (BACI) experiment. In BACI analyses importance is not so much put on differences between treatments at any time point, but on the interaction between bloom (before/after exposure) and seed-treatment. This interaction indicates whether the change over time differed between treatments or in this specific case whether Nosema prevalence developed differently in clothianidin-exposed than in unexposed colonies.

1	63	187 Is “prior to placement in the fields” the same as the Before oilseed rape bloom sampling time point? These would appear to be two different times.	Prior to placement in the fields is the same time point as before oilseed rape bloom. We agree that the wording might be confusing for the reader. We do not use this term in the result section, which is reworded.
1	64	193 N. ceranae was also included in this analysis.	The result section was reworded. We analysed the effect of clothianidin on the abundance of ALPV, DWV-A and DWV-B (full model) and of ABPV, CBPV, IAPV, KBV and SBPV in 2014, which we included in the text.
1	65	202 ... were unaffected ...	We rewrote the result section.
1	66	211 Are these the increases in mites/100 bees or the actual values?	These values represent the increase. We specified this in the text as it might be confusing (lines 271 - 273).
1	67	223 S. alvi titers increased during the oilseed rape bloom in both 2013 and 2014.	In the full model we could not detect a significant effect by bloom or year and did not mention it therefore.
1	68	224 ... remained consistently high through the experiment and were not significantly altered during the bloom in either 2013 or 2014.	We reworded this part of the result section.
1	69	229 ... at the apiary level.	Revised as suggested.
1	70	233 What kind of seasonal effects? Increases or decreases?	Both immune gene expressions increase in titres after the oilseed rape bloom compared to before. We added this information in the text (see lines 283 & 284).
1	71	Once again, comment on the gut microbes (half of the observed differences attributed to clothianidin treatment).	We have now commented on the gut microbes in the discussion (see lines 337 – 345).
1	72	267 (“Healthy” colonies.) But the colonies were very small and the Varroa counts/virus abundance were relatively slight compared to highly stressful conditions. The fact that the colonies survived so well despite being small under pesticide exposure is worth noting.	We have now included that the experimental colonies were undersized in the previous paragraph on colony development (lines 408 and 428).
1	73	277 The Varroa infestation levels here are relatively slight and the amount of time from the beginning to the end of the oilseed rape bloom is relatively short.	The reviewer is correct. We rewrote the discussion. However, please note that we did not detect a significant impact of clothianidin on Varroa infestation excluding swarmed colonies and using a full model including data from both years.
1	74	290 What about the greater post-bloom decrease in S. alvi abundance	There was no detectable three-way interaction between bloom, year and seed-

		in 2013? That result contradicts this statement.	treatment on S. alvi . However, there was a two-way interaction between bloom and year, indicating that the change between the pre- and post-exposure times differed between years. This difference was, however, not influenced by the clothianidin treatment.
1	75	309 Such as induced responses in a newly-emerged adult bee?	Here, we mean “induced responses immediately upon first exposure”, but this applies to all bees, not just newly emerged adults: foragers (when encountering neonicotinoids in the field), house bees (including newly emerged) when handling contaminated nectar/pollen or communicating with foragers. We reformulated the sentences to make it clearer (see lines 365 – 368).
1	76	321 What about the dilution effects of landscape forage outside the treatment plots? I don’t know that this was examined closely, but this could account for major differences.	Here, we discuss the differences between clothianidin concentration found in bee collected nectar and pollen. In 2013 the concentrations were twice as high as in 2014. This could well be a result of dilution effect of different foraging patterns. However, in 2014 we found that collected pollen originated on average 93% (treated) and 91 % (control) from oilseed rape, while only 53% (control) and 63% (treated). So one would rather expect a higher concentration of clothianidin. There was additional spring-sown oilseed rape apart from the focal fields in half of the landscapes in 2014 (1 to 13 ha), which is now reported in the manuscript (see lines 453 - 457).
1	78	327 I agree that these probably reflect constitutive gene expression for the reasons the authors elaborate on, but I don’t think you can rule out induced responses entirely since this was not tested for.	Line 337? We reworded this part of the discussion slightly, with the emphasis on no observable negative effect. Furthermore, with the sentence following that we acknowledge that detrimental effects could well have existed, we address the issue that we do not test at individual level (see lines 420 – 425).
1	79	364 Was land use the same over successive years (2013 and 2014)? If not, how did it change, especially as it relates to landscape pesticide exposure outside the experimental fields? In R15, land use similarities were based on 2011 use (not the	In Supplementary Table 7 we report on the land use in the surrounding of the focal oilseed rape fields as well as the field sizes in 2014. We also added the fact that there was other spring-sown oilseed rape within the 2 km in half of the landscape (1 to 13 ha) (see lines 453 -

		experimental years). The experimental oilseed rape field sizes are quite sufficient to support and attract the colony density mentioned here.	457).
1	80	388 Biscaya inclusion/exclusion results are located in Supplementary Table 6, not 4.	These results are presented in Supplementary Table 8 now and we changed it accordingly.
1	81	393 Rundlöf et al, 2015 mentions that colonies were equalized in 2013 – were the colonies equalized in 2014 between the spring and before bloom observations to obtain colonies of this description? The randomization of 2014 colony location between fields (within treatment group) was highly appropriate.	Yes, colonies were equalized in 2014. This process was done twice (see lines 516 - 521). We hope that this is clarified by the additional figure presenting the experimental timeline (see Supplementary Fig. 1).
1	82	419 The switch in pesticide treatment of individual fields from 2013 to 2014 is described three times here. This repetition can be reduced.	We reduced the unintended repetition as suggested. Two things are described here, that the pesticide treatment was switch between the farms but also that the colonies were exposed to the same treatment twice.
1	83	423 Some confusion here as to which level (colony or field) these clothianidin concentrations represent. The wording here sounds like clothianidin contents are at the colony level, but Rundlöf et al, 2015 has similar results where clothianidin contents are at the field level.	As the reviewer correctly points out, the wording is confusing. The sampling was done per field and not per hive. We corrected the unintended error.
1	84	431 Not clear how many fields were examined here for the comparison of clothianidin contents in individual bees and their honey stomach nectar contents. Was it only three fields from the clothianidin treatment fields, or were there also three additional fields from the control fields?	Samples were only taken from three clothianidin-treated sites and no control sites. We reworded this sentence to make it clearer.
1	85	440 Given that adult worker number is an important metric , it would help to describe briefly how the number of bees were estimated.	We added information about how the number of adult bees was estimated (see lines 550 & 551).
1	86	440 The inclusion of both the 2013 and 2014 pesticide/fungicide applications is useful. Refer to Extended Data Table 3 of Rundlöf et al, 2015 for the 2013 field plant protection application information.	We included the plant protection applications used at the experimental sites from both years, to make it easier for an audience without having to read Rundlöf et al. (2015). See Supplementary Table 8.
1	87	440 A timeline/phenology table of	We included a timeline of the experiment.

		major events is helpful for this manuscript given the variation in experimental start dates. A Supplementary Table for 2014 events similar to Rundlöf et al, 2015 Extended Data Table 2 (2013 events) would round out the experimental timelines nicely.	See Supplementary Fig. 1.
1	88	446 Not clear what constitutes a “minimum of colonies” that leads to a virus not being screened. Is it excluded if it is not detected at all, or is it another threshold?	We screened the colonies for all viruses, to have a complete dataset. The sentence in line 466 was deleted.
1	89	453/466 Suggest that authors alphabetize viruses by name or acronym during presentation in text, figures, or tables.	Revised as suggested.
1	90	463 This mention of a modified primer would go better with the PCR methods on line 520.	The sentence “The reverse primer for Ame \LRR was...” can be now found under the paragraph RT-qPCR and qPCR (see line 622).
1	91	466 Not clear what criteria is meant by a “minimum of colonies”. The threshold appears to be above 0 since 2013 SBPV and CBPV are not included in further analyses.	We screened the colonies for missing viruses. The sentence was deleted from the manuscript
1	92	488 Teflon ... resuspended 496 ... and was eluted into ... 559 Reword this sentence, it’s confusing.	Revised as suggested.
1	93	595 List which tests used R84 software and which used SAS 9.4 software.	The software SAS 9.4 was only used for statistical test addressing the verification of clothianidin exposure. We added this information in the text now (see lines 728 & 729).
1	94	Figure 2. Indicate on the graph that colony weight was not measured for the Spring 2014 time point.	We now use colony weight to calculate colony honey production. This value is available only per year. Therefore, we decided to split the figure into two (Figures 2 and 3).
1	95	Figure 3. Microorganism prevalence at the colony level? Indicate on the graph that microbial prevalence was not analyzed for CBPV, IAPV, KBV, or SBPV. If there are error bars present, shouldn’t there be at least a minimum % prevalence (minimum number of colonies with the microbe	Prevalence was analysed for all viruses at all timepoints. See Response 19 for a detailed answer regarding the comment on error bars.

		present)? Right here, several microbes have error bars with 0% prevalence.	
1	96	Figure 4. Indicate which treatment-control pairs are significantly different on the graph.	We like the idea of marking the treatment-control pairs that are significant on the graph, however we believe as many effects and their interactions are tested for it might get too confusing for the reader to understand, what this significance stands for.
1	97	Figure 5. SPH51 or SPH5?	The correct term is SPH51 as used in the graph. We revised it in the result part (line 282).
1	98	Table 1. Use the same significant figures throughout the reported values. Some of the control field workers have detectable clothianidin in their tissues or nectar – isn't this worth commenting on in the results?	Revised as suggested. We also mention it in the discussion (lines 382 - 390)
1	99	Table 2. The period x treatment interaction for the 2014 capped brood is quite close to being significantly different and is significantly different when Biscaya treated colonies are excluded. Is this marginality worth commenting on in the results?	In the new analysis which is excluding swarmed colonies, this result is significant, which is reported in the result section (see lines 172 – 175) and discussed (see lines 326-330).
1	100	Supplementary Figure 1. Include the year of sampling in the legend.	Revised as suggested (Supplementary Fig. 2).
1	101	Supplementary Figure 2. Include the year for the samples. For apiary prevalence, shouldn't any detected organism have at least a 6.25% prevalence (1/16 of total number of apiaries (fields))? 2013 analyses are missing for ABPV, BQCV, and LSV-1, as well as N. apis, all of which are mentioned in the results (previously mentioned, 160 and 185). Same comment on error bars and minimum prevalence in the graph as Figure 4.	In the previous Supplementary Table 1 we report the number of samples that a virus has been detected in. In that table we do not differ between treatments. Therefore, all 16 fields are reported together. In Supplementary Figure 2, treatments are compared with each other and therefore if a virus was prevalent in only one colony, this results in a 12.5% prevalence (1/8). However, we decided to exclude the data on apiary level for viruses to make it clearer for the audience.
1	102	Supplementary Table 1 Honeybee viruses screened for prevalence at the apiary and colony levels	We excluded the prevalence data at apiary level and rearranged the Supplementary Table 1.
1	103	Supplementary Table 2 Pathogen and parasite prevalence at the colony level. The 2013 analyses of the ABPV, BQCV, LSV-1, and N. apis are missing despite being discussed in the results (previously mentioned). The changes in N. apis prevalence are	We reworded the result section and clarified if statistical analyses could not be obtained (see also Supplementary Table 1).

		quite large in Figure 3.	
1	104	Supplementary Table 3. Immune gene expression.	We changed the headings of the Supplementary Tables throughout the manuscript accordingly to the requirements of Nature Communications.
1	105	Supplementary Table 6 388 The fact that the significance/non-significance of these three metrics is dependent on whether Biscaya treated colonies are included or not suggests that these are marginally significant/non-significant effects.	We believe that the most reliable results are those from the whole dataset (including the control field and the clothianidin-treated field that have been sprayed with Biscaya). Excluding these fields reduces the sample size and to a minor extent the paired study design. Thiacloprid contained in Biscaya has a considerably lower acute toxicity for honeybees than clothianidin and only trace amounts of thiacloprid were detected in the pollen, nectar and bee samples. The latter was also true for other neonicotinoids. Therefore, we focus in the manuscript on these results, but include in the Supplementary Information for transparency the few results that differed qualitatively between analyses with and without the Biscaya sprayed fields.
1	106	Supplementary Material 108 Reword this sentence	Revised as suggested
1	107	Supplementary Material 125 What are the sources of the chemical standards described here?	We added additional information (see lines 168 & 169 in the Supplementary Material).
		Referee #2	Response:
2	108	As reported in the introduction, neonicotinoids can have several sublethal effects. In particular, it has been shown that Clothianidin can impair immunity and indirectly influence the proliferation of Deformed wing virus. According to the proposed mechanism, Clothianidin should affect the abundance of the virus rather than its prevalence. In fact, the authors did not notice any effect of the treatment on DWV prevalence, whereas they did not check any possible effect on abundance, because prevalence was generally low. Accordingly, the lack	We agree with the reviewer and included a full model analysing the effect of clothianidin on the abundance of both DWV-A and DWV-B for two years. We could not detect an effect of treatment over the oilseed rape bloom.

	of any effect of Clothianidin on DWV may well depend on the initial conditions (virus absent and thus impossible proliferation) rather than the natural homeostatic mechanisms mediating the colony's response. Under this perspective, one of the most important conclusions of the study would be unsupported and the present study would just become the last of a long series providing contrasting results on the possible effects of Clothianidin on bee health, because of the variability of the experimental conditions. However, according to Supplementary table 1, in 2014, there were 44/87 DWV type A positive samples. Why do not the authors re-analyze those samples to assess DWV type A abundance and compare virus titers in treated and control colonies that tested positive for the virus? Clearly, numbers will not be large, but I don't see why these data should not be used for the purpose of the comparison. Then, if a significant effect of the treatment on DWV abundance was noted, in view of the absence of effects at colony level, the authors could confidently say that the natural homeostatic mechanisms mediating the colony's response allowed the colony to overcome the impairment of individual and social bee life parameters due to chronic exposure to neonicotinoids, and their important conclusion would be fully supported. On the other hand, if a significant effect of the treatment on DWV abundance was not observed, authors could discuss why the effect observed under lab conditions was not confirmed under field conditions.	
2	109 Title 1: in the actual formulation the title suggests an universal lack of effects; this looks a little bit pretentious.	We changed the title to: "Clothianidin seed-treatment has no detectable negative impact on honeybee colonies and their pathogens". See also Response 23 (reviewer 1).

2	110	Line 38: the wording “despite seasonal fluctuations” suggests a link between seasonality and the effect under study.	We reformulated the abstract and deleted this part.
2	111	Line 51: ref. 9 does not seem the most appropriate in this case.	We use Potts et al. (2016) as well as Klein et al. (2007) here.
2	112	Line 52: ref. 10 does not seem the most appropriate in this case.	We formulated the sentence more generally and included Potts et al. (2016) as a reference. Potts, S. G. et al. Safeguarding pollinators and their values to human well-being. Nature 540, 220–229 (2016).
2	113	Line 54: ref. 12 does not seem the most appropriate in this case.	We deleted this part of the introduction as we had to limit our citations to 70.
2	114	Line 57: ref. 17 does not seem the most appropriate in this case.	We changed to: Mondet, F., de Miranda, J. R., Kretzschmar, A., Le Conte, Y. & Mercer, A. R. On the front line: quantitative virus dynamics in honeybee (Apis mellifera L.) colonies along a new expansion front of the parasite Varroa destructor . PLoS Pathog. 10, e1004323 (2014).
2	115	Line 59: a reference may be added at the end of this sentence.	We decided not to add another reference, as we are above the expected number of references by Nature Communications.
2	116	Line 61: ref. 19 does not seem the most appropriate in this case since the cited article deals with neonicotinoids and not all pesticides.	We changed to: Sanchez-Bayo, F. & Goka, K. Pesticide residues and bees - A risk assessment. PLoS One 9, (2014).
2	117	Line 64: ref. 23 does not seem the most appropriate in this case.	We removed Rundlöf et al. (2015).
2	118	Lines 78-89: in my opinion this paragraph should be moved after line 67, at the end of the discussion about the sublethal effects because the ones described here are indeed another sublethal effect of neonicotinoids.	We rearranged the introduction according to suggested changes.
2	119	Line 82: there are studies stating that effects can be noted also at low doses.	Yes, indeed. However we have to be selective to conform to the requirements for the journal on the number of cited references.
2	120	Line 86-89: the fact that field level evidence is still controversial does not mean that the possible impact is “overemphasized”. I don’t think that the actual situation of bees (-20% of colonies every year in most countries of the Northern Hemisphere) allows us to underestimate any possible threat to bee health.	We removed the second part of the sentence.

2	121	Line 92 (and elsewhere): I believe that the corresponding number in the reference list should be added whenever Rundlof et al., is cited, but please check the journals prescriptions.	Revised as suggested.
2	122	Line 122: I'd first report the levels found in 2013 and then say that in 2014 they were halved.	We reformulated the sentence but still have the same order. We still hope that it is clearer (see lines 146 – 150)
2	123	Line 151: I understand the attempt to simplify the description, but including Varroa among microorganisms seems a little bit exaggerated to me.	Revised as suggested and changed the title of the paragraph to “pathogen, parasite and microbe prevalence”.
2	124	Lines 151-190: nearly 40 rows are dedicated to comment about the possible effect of the treatment on the prevalence of microorganisms (Varroa is not that “micro” actually!). This seems a lot of space for something that, very likely, should not be affected by the treatment. In fact, to my knowledge, chemical treatments should not make pathogens appear in previously pathogen free bees. Therefore I'd suggest to shorten this part of the manuscript which can be distracting.	The problem with prevalence measurements is that they are very much affected by the ability to detect a pathogen, which has more to do with detection threshold limits than the actual presence or absence of a pathogen. Our default stance on prevalence is therefore that pathogens are ‘detected/not detected’, rather than ‘present/absent’. This is why many pathogens that were ‘absent’ before foraging became ‘present’ after foraging: not because they were suddenly acquired from the environment, but because they were always there, just not detectable. In this context, subtle influences of pesticide exposure on pathogen susceptibility become interesting and can be monitored through this shift at low abundance levels across the detection limit. That said, the reviewer is correct to criticise the excessive length of discussion non-prevalence, since it is in fact pathogen abundance that is the much more interesting and relevant parameter for overall bee colony health. We have reduced this section considerably, and removed instances of redundancy with the Results section.
2	125	Line 206: “increased to”?	We reformulated the result section. We write now: “In contrast, the adult bee Varroa infestation rate increased during oilseed rape blooms from, on average, around 0.3 to 1 mites per 100 bees in 2013 and from 0.8 to 3.6 mites per 100 bees in

			2014.” We hope it gets clearer.
2	126	Line 295: “In 2013”?	We write now: “In our study the mRNA levels of several key honeybee immune genes were not affected by the placement at clothianidin treated fields or, generally, by the oilseed rape bloom in 2013.”
2	127	Line 362: why the standard error of the mean was used here instead of the standard deviation?	As suggested, we report the standard deviation now, instead of the standard error (line 445).
2	128	Line 398: “descent”?	Revised as suggested (line 485)
2	129	Line 426: how long were the pollen traps used for?	The pollen traps were used for a day until 25 ml of pollen of collected from three colonies together. We included this information now (line 529).
2	130	Line 491: I’m not sure “Amel” is sufficient as the gene name.	The reviewer is correct: the gene name is in fact LLR, prefixed by Amel to distinguish it from the drosophila LLR gene, i.e. Ame ^LLR. This has been changed throughout the MS.
2	131	Line 490 (and elsewhere): check if a space has to be put after the number in 65°C	Revised as suggested.
2	132	Line 791: “influence”	Revised as suggested.
2	133	Line 795 (and maybe elsewhere): Varroa destructor should go in italic	Revised as suggested.
2	134	Supplementary table 1, line 4: remind to the reader why the number of samples does not correspond to the number of colonies.	Unfortunately, the information previously presented in Supplementary Table 1 had a mistake. For 2014 as the number of samples was 80 and not 87. We apologize for the mistake. At apiary level: n = 16 if screened only before the oilseed rape bloom, n = 32 if screened before and after; on colony level: for 2013 n = 192 (16 fields x 6 colonies x 2 sampling points), for 2014 n = 80 (10 fields x 4 colonies x 2 sampling points). As we exclude colonies now, that either swarmed or lost their queens, we have a reduced samples size which is explained in the Supplementary Table 1: “Number of samples in 2013 (38 samples were excluded from the 192 (96 before and 96 after oilseed rape bloom) samples due to swarming (2 x 18) or loss of queen (2 x 1)) and in 2014 (6 samples were excluded from the 80 samples (2 x 40) due to swarming (2 x 3)).”
2	135	Supplementary table 5: use “Product” instead of “Compount treatment”.	Revised as suggested (see Supplementary Table 8)

		Referee #3	Response:
3	136	A key conclusion here is the apparent resilience of honeybee colonies to showing measurable impacts of clothianidin exposure under these field conditions, even when pathogen infection status is also considered. In their earlier paper, Rundlöf et al. (2015) presented a useful statistical power analysis to show the magnitude of effect size that their experimental design would have been able to detect. Logistical constraints have restricted the degree of replication in this experiment for the 2014 season both at site level, and the number of colonies placed per site. As such, the ability of this experiment to detect effects of a given size will also have been reduced concomitantly. I would like to see the authors provide indications in the manuscript of their sensitivity to detect measurable impacts for the 2014 data. This will help readers to put the apparent absence of detectable impacts of pesticide exposure (with or without pathogens) into appropriate context.	As suggested, we added a power analysis for honeybee colony strength (number of adult bees and number of capped brood) and honey production (see Supplementary Figure 3 and Method section). For the number of adult bees we detected an effect size of below 5% with a power of 80%. This is even lower than the requirements of an effect size < 7% set by EFSA. EFSA (2013) EFSA guidance document on the risk assessment of plant protection products on bees (Apis mellifera, Bombus spp. and solitary bees). EFSA J. 11, 3295.
3	137	L61: "Another threat to honeybees is the chronic exposure to pesticides used in agriculture¹⁹." Given the wealth of publications in this field, I was surprised to see only the older Blacquiere et al (2011) review cited in support of this statement. I would encourage the authors to cite additional work alongside this review [19] here, e.g. Godfray et al. 2014, 2015.	We changed this citation to Sanchez-Bayo, F. & Goka, K. Pesticide residues and bees - A risk assessment. PLoS One 9, (2014), as it presents more in general threats from pesticides and not only neonicotinoids. We could not include additional references and still conform to the maximum allowed number of references.
3	138	L65-67: All references here come from studies of honeybees, so I assume the intent of the authors was that this sentence refers specifically to these impacts in honeybees alone. If so, then the authors should specifically refer to honey bees (rather than generic "bees") here. If the points were intended to cover impacts on a wider range of bee	We changed bee to honeybee to be more specific.

		species, then they should cite work on other taxa relevant here. For example, impacts of neonicotinoid exposure on foraging success in bumblebees (e.g. Gill et al. 2012; Feltham et al. 2014; Stanley et al. 2016), impacts on learning and memory in bumblebees (e.g. Stanley et al. 2015; Piroinen & Goulson 2016), and impacts on worker production and queen production in bumblebees (e.g. Whitehorn et al. 2012; Gill et al. 2012).	
3	139	L69-70: The authors should update this sentence to reflect the recent EU decision to further restrict use of these three neonicotinoids outside of greenhouses.	Revised as suggested (see lines 85 – 89).
3	140	L147: The term "in-wintering" is unusual. Could this be replaced with something in more common usage. Perhaps change to "...and were therefore slightly undersized going into winter, early September 2013."	We changed the term "in-wintering" to over-wintering
3	141	L655: " Apis mellifera " should be in italics.	Revised as suggested.
3	142	L873: Change from "The data is shown on a logarithmic.." to "The data are shown on a logarithmic..."	Reworded to "Immune gene expression data are presented as the log10 number of estimated mRNA copies per bee"

REVIEWERS' COMMENTS:

Reviewer #1 (Remarks to the Author):

The revised manuscript very much addresses the concerns that I had in my original review. The revised manuscript is both compact and focused for the large amount of results that it covers. The authors imposed structure, clarity and intent to an experiment that, like other landscape experiments, easily could otherwise become confusing and unmanagable. The authors rewrote or corrected ambiguous and erroneous sections throughout the manuscript. The few remaining remarks that could not be addressed (i.e. small colony size, absence of broad spectrum or stored food material pesticide analyses) did not really alter the findings of the study, and understandably may not have been within the scale of this experiment. Notably, the removal of microbe analyses at the apiary level and filling out of all microbe analyses to a two year period critically improved the results. Critically, too, the inclusion of queen/colony mortality data (particularly the "autumn decision" stats) provides much needed information on colony survivorship through the experiment. The extended picture discussed here is much more in line with small colonies attempting to overwintering in northern temperate climates. The authors nicely provide big picture summaries of the varied results with details on specific outcomes.

I particularly appreciate the authors' discussion of the significance of their results in the broader context of neonicotinoid sensitivity and colony stressors. Landscape studies on honey bees attempt to resolve the findings of smaller scale studies with field colonies in a highly complex and expansive setting. I found the revised manuscript to be very well considered in its assessment of the impacts of neonicotinoids relative to overall stressors facing field colonies in landscape studies.

Suggestions/corrections

In184 Is the honey table in Figure 3?

Otherwise, the figure/table citations matched the actual figures/tables. The authors may want to double check the figure/table numbering to make sure that the numbering follows the sequence of appearance.

In 250 ... declining more in control ...

In 535 (see quantification). (put this in parentheses).

Reviewer #3 (Remarks to the Author):

The authors have undertaken a substantial revision of this manuscript which has addressed the issues I raised in response to their original submission. However, there are now a number of sections in this version of the manuscript that need to be written more clearly (to avoid ambiguity of meaning) - I have tried to detail these areas of concern below:

L62-63: "Neurotoxic neonicotinoids are a class of insecticides that are used globally¹² and usually applied as a seed dressing or sprayed¹³." Whilst both seed treatments and spray applications are ways in which neonicotinoids are applied to crops, there are also other ways in which these insecticides can be applied: e.g. soil drenches for horticultural crops, and even direct injection into tree trunks. The mode of application is rather crop specific, so I would suggest the authors are less prescriptive here about how "usual" (or not) the modes of application are, rather they are more inclusive with the possible routes of exposure for agricultural crops.

L85-89: "As evidence of adverse effects of neonicotinoids amounted, the European Union has decided to ban all outdoor uses of these three neonicotinoids from December 2018 ... of the

European Union." Suggest this is changed to: "As further evidence of adverse effects of neonicotinoid exposure has accrued, the European Union recently decided to ban all outdoor uses of these three neonicotinoids from December 2018³³, although they can still be used in permanent green houses and for outdoor use in countries outside of the European Union."

L104-107: "In 2014, colonies at clothianidin treated fields decline less strongly in *Gilliamella apicola* abundance than control colonies and clothianidin treatment is negatively associated with the abundance of Black queen cell virus (2014) and Aphid lethal paralysis virus (both years)." Does this mean that colonies in clothianidin-treated fields showed a lower rate of decline in *Gilliamella apicola* abundance over the season, compared with colonies placed on control fields? If so, this could be more clearly expressed in the manuscript text.

L110: what do the authors mean by the term "adaptive capacity" used here? This requires further explanation and clarification.

L116: the authors need to be careful here with the use of the term "untreated seeds". My understanding of the information they provide in Rundlof et al. (2015) is that oilseed rape seeds in either treatment were either treated with the neonicotinoid clothianidin and the fungicide thiram, or just the fungicide thiram. As such, all seeds were "treated seeds", but only half of them were "neonicotinoid treated seeds".

L122-123: I am still a little confused about what these two statements mean for allocation of hives across the two seasons. L123 suggests that the application of neonicotinoid treated seed was reversed between seasons - such that if site A and B received neonicotinoid (and fungicide) treated seed and control (only fungicide treated seed) respectively in season 1, then site A would get control seeds and site B would get neonicotinoid treated seed in season 2. However, looking at Figure 1 it is clear that there were substantial changes in site/ field locations between seasons. As such, it looks like only one site to which neonicotinoid seed was applied in season 1, had control seed applied in season 2. But it looks as though five fields with control seed in season 1, had neonicotinoid treated seed in season 2 (although the alignment of symbols in the figure might indicate some movement of field site within the local vicinity between seasons for some of those sites). Given these site changes between seasons, I think the statement in L123 is somewhat misleading, and should be qualified with regards to these points. Additionally, does L122 suggest that colonies that were on neonicotinoid treated sites in season 1, were also deployed to a neonicotinoid treated site (though not the same site location) in season 2?

L130: I suggest replacing "(treated, untreated)" with "(neonicotinoid-treated or not)" here.

L130: "...and in relation to placement at clothianidin treated fields, represented by a differential response during the oilseed rape bloom between colonies at treated and untreated fields (i.e. the seed-treatment x bloom interaction) for both years combined," I think this part of the sentence requires more clarification/ rewording. I think this is related to confusions arising from L122-123 (see comment above).

L143: replace "treated fields" with "neonicotinoid treated fields" here. See also L145.

L152: revise "treated and untreated fields" here.

L163-164: suggest revising from "(brood amount and adult bees)" to "(amount of brood and numbers of adult bees)".

L164: "The change in colony strength was also much stronger during the 2013 bloom than during the 2014 bloom" Do you mean that colonies gained strength more quickly during bloom in 2013, than during bloom in 2014 (irrespective of whether colonies were in neonicotinoid-treated or control fields)? If so, this could be more clearly explained.

L186: revise "treated and untreated fields" here.

L187: revise "treated fields" to "neonicotinoid treated fields".

L188: revise "untreated fields" to "control fields".

L197: revise "treated fields and 13 from untreated fields".

L198: revise "treated" to "neonicotinoid treated".

L215: "...mostly ubiquitous in all colonies, ..." This is loosely and inaccurately phrased. Ubiquitous has a specific meaning. If these were not ubiquitous, then don't use the word.

L215: revise "treated and untreated".

L234: revise to "neonicotinoid treated fields".

L250: revise "declining stronger" to "declined more strongly."

L254: revise from "seed-treated" to "neonicotinoid treated" (both fungicide and neonicotinoid are "seed treatments in this system).

L261 & 263: replace "seed-treatment" with "neonicotinoid treatment".

L292: what do the authors mean by the term "high adaptive capacity" used here? This requires further explanation and clarification. Perhaps a different term would be more appropriate here - e.g. "strong resilience".

L318: "Also the excessive colony mortality..." Is it reasonable to describe the mortality as "excessive" here? I suggest "higher colony mortality" would be a more appropriate descriptor here.

L328-330: "...were left slightly understrength relative to the control colonies by the beekeeper interventions in June 2014, and may have overcompensated as a result." This seems like important information that needs further explanation in the paper. Why is the reader only hearing about such important impacts of apiary management on the outcome of the study at this stage in the manuscript?

L390: revise to "neonicotinoid treated fields."

L395: "These variations might be the result of different foraging alternatives in the different landscapes or differences in clothianidin concentrations in the plants." Different/ce used 3 times in this sentence. Consider alternative words for 1 or 2 instances in this sentence.

L481: Insert "and" so reads "One hundred and sixteen honeybee..."

L509: revise to "exposed/unexposed to neonicotinoids".

L516: Suggest you revise from "The colonies were equalized as described for 2013,..." to "The strength of colonies were equalized as described for 2013,..."

L570: revise "treated and untreated".

L681: revise from "...to the treatment from..." to "...to the neonicotinoid treatment from..."

L686: revise "treated fields".

L854: "spp." should not be in italics.

L866: Please give correct details of volume and page numbers here. This should be: "Nat. Ecol. Evol. 1, 1308-1316 (2017)."

L884: the article number is missing here. Should be: "PLoS One 6, e20656 (2011)."

L892: the article number is missing here. Should be: "PLoS One 9, e103592 (2014)."

L901-902: volume/ page number info is wrong here. Should be: "Ecotoxicology 27, 772-783 (2018)".

L990: "untreated (control fields, white) seeds" see comment relating to L116 above.

Reviewer #4 (Remarks to the Author):

The manuscript from Osterman et al describes a comprehensive, large-scale, multi-year analysis of the impacts of proximity to oilseed rape (with or without neonicotinoid seed treatments) on honey bee colony health. The authors confirm there are consistent differences in the amount of clothianidin returned to the hive by foraging bees, and examine a large number of parameters, including colony population metrics, parasite, pathogen and gut microbe loads, and gene expression. Overall, there is no indication of negative impacts on the honey bee colonies that have been placed adjacent to the treated fields. The authors also conducted power analyses to evaluate their ability to detect weak effects.

Overall, these results are consistent with previous studies demonstrating that honey bee colonies can be relatively robust to the impacts of stressors, including low-level pesticide residues found in nectar and pollen resources that arise from seed treatments. However, as the authors note, there are several studies which demonstrate negative impacts of neonicotinoid seed treatment in some sites and conditions. Again, this is consistent with the larger consensus of the scientific community that while honey bee colonies, due to their social structure, show remarkable resilience to individual stressors, this resilience is eroded when colonies are challenged by multiple stressors. The colonies in the current study are provided access to good forage and supplementary feed throughout the year, Varroa mite levels are controlled (Varroa represent the major parasite of managed honey bees in Europe and North America), and, aside from the treatment period, the colonies are housed at sites which appear to have low pesticide pressure. Thus, the differences between the colonies are really only due to exposure during a ~4 week period.

While the authors note in the Discussion, "The importance of the results presented here is that in fundamentally healthy colonies, like the ones studied here, the natural homeostatic health mechanisms mediating the colony's response to its environment are robust enough to overcome these impairments, even in undersized colonies" this should be elaborated on so that the reader fully understands what the treatment actually consisted of (1 month of exposure and 11 months of relatively optimized management conditions). This information should also be included in the abstract.

Point-by-point response to remarks by the reviewers:

R e f e r e e #	N u m b e r	Remark from the Reviewers	Response from the Authors
		Referee #1	Response:
1	1	ln184 Is the honey table in Figure 3?	Indeed, it is Figure 3 instead of Figure 2. We corrected the error.
1	2	The authors may want to double check the figure/table numbering to make sure that the numbering follows the sequence of appearance.	The numbering of tables and figures follows the sequence of appearance.
1	3	ln 250 ... declining more in control ...	Changed to: declining more quickly in control
1	4	ln 535 (see quantification). (put this in parentheses).	Revised as suggested.
		Referee #3	Response:

3	5	L62-63: "Neurotoxic neonicotinoids are a class of insecticides that are used globally¹² and usually applied as a seed dressing or sprayed¹³." Whilst both seed treatments and spray applications are ways in which neonicotinoids are applied to crops, there are also other ways in which these insecticides can be applied: e.g. soil drenches for horticultural crops, and even direct injection into tree trunks. The mode of application is rather crop specific, so I would suggest the authors are less prescriptive here about how "usual" (or not) the modes of application are, rather they are more inclusive with the possible routes of exposure for agricultural crops.	We have rephrased this according to the reviewers suggestions as follows: “Neurotoxic neonicotinoids are a class of insecticides that are used globally¹² and are usually applied in arable agricultural crops as a seed dressing or spray¹³ (see lines 63 and 64).
3	6	L85-89: "As evidence of adverse effects of neonicotinoids amounted, the European Union has decided to ban all outdoor uses of these three neonicotinoids from December 2018 ... of the European Union." Suggest this is changed to: "As further evidence of adverse effects of neonicotinoid exposure has accrued, the European Union recently decided to ban all outdoor uses of these three neonicotinoids from December 2018³³, although they can still be used in permanent green houses and for outdoor use in countries outside of the European Union."	Revised as suggested.

3	7	L104-107: "In 2014, colonies at clothianidin treated fields decline less strongly in Gilliamella apicola abundance than control colonies and clothianidin treatment is negatively associated with the abundance of Black queen cell virus (2014) and Aphid lethal paralysis virus (both years)." Does this mean that colonies in clothianidin-treated fields showed a lower rate of decline in Gilliamella apicola abundance over the season, compared with colonies placed on control fields? If so, this could be more clearly expressed in the manuscript text.	We have rephrased the sentence in question in a manner similar to that suggested by the reviewer: “Placement at clothianidin treated fields was associated with increased brood production in the second year and with more adult bees across both years. During the oilseed rape bloom in 2014, Gilliamella apicola abundance declined more slowly in colonies at clothianidin-treated fields than in control colonies, and clothianidin treatment was negatively associated with the abundance of Black queen cell virus in 2014 and Aphid lethal paralysis virus (both years)”
3	8	L110: what do the authors mean by the term "adaptive capacity" used here? This requires further explanation and clarification.	Here and elsewhere the reviewer conveys unease with our use of ‘adaptive’ which is also used in an evolutionary context, and thus, though technically correct, slightly ambiguous. We have re-phrased these terms to avoid the use of ‘adaptive’ in order to minimize such ambiguity.
3	9	L116: the authors need to be careful here with the use of the term "untreated seeds". My understanding of the information they provide in Rundlof et al. (2015) is that oilseed rape seeds in either treatment were either treated with the neonicotinoid clothianidin and the fungicide thiram, or just the fungicide thiram. As such, all seeds were "treated seeds", but only half of them were "neonicotinoid treated seeds".	The reviewer is correct. We have reworded the sentence to be more precise.

3	10	L122-123: I am still a little confused about what these two statements mean for allocation of hives across the two seasons. L123 suggests that the application of neonicotinoid treated seed was reversed between seasons - such that if site A and B received neonicotinoid (and fungicide) treated seed and control (only fungicide treated seed) respectively in season 1, then site A would get control seeds and site B would get neonicotinoid treated seed in season 2. However, looking at Figure 1 it is clear that there were substantial changes in site/ field locations between seasons. As such, it looks like only one site to which neonicotinoid seed was applied in season 1, had control seed applied in season 2. But it looks as though five fields with control seed in season 1, had neonicotinoid treated seed in season 2 (although the alignment of symbols in the figure might indicate some movement of field site within the local vicinity between seasons for some of those sites). Given these site changes between seasons, I think the statement in L123 is somewhat misleading, and should be qualified with regards to these points. Additionally, does L122 suggest that colonies that were on neonicotinoid treated sites in season 1, were also deployed to a neonicotinoid treated site (though not the same site location) in season 2?	The sites in 2014 are not exactly the same as in 2013, as the same farms but not the same fields were used. This is because farmers do not cultivate oilseed rape twice in a row to ensure crop rotation. In addition, since fewer sites were used in 2014 than in 2013, some squares do not have a circle nearby. The field-design for the two years is described in detail in the materials and methods. To clarify that the same farms but not the same fields were used we replaced “localities” and “field-pairs” by “farms” and “farm-pairs”.
3	11	L130: I suggest replacing "(treated, untreated)" with "(neonicotinoid-treated or not)" here.	We changed these terms to “(clothianidin-treated, control)”.

3	12	L130: "...and in relation to placement at clothianidin treated fields, represented by a differential response during the oilseed rape bloom between colonies at treated and untreated fields (i.e. the seed-treatment x bloom interaction) for both years combined," I think this part of the sentence requires more clarification/ rewording. I think this is related to confusions arising from L122-123 (see comment above).	We have re-phrased this sentence to remove the ambiguities in the description of the results.
3	13	L143: replace "treated fields" with "neonicotinoid treated fields" here. See also L145.	We replaced "treated fields" with "clothianidin-treated fields" in both lines.
3	14	L152: revise "treated and untreated fields" here.	We changed these terms to "clothianidin-treated and control fields".
3	15	L163-164: suggest revising from "(brood amount and adult bees)" to "(amount of brood and numbers of adult bees)".	Revised as suggested.
3	16	L164: "The change in colony strength was also much stronger during the 2013 bloom than during the 2014 bloom" Do you mean that colonies gained strength more quickly during bloom in 2013, than during bloom in 2014 (irrespective of whether colonies were in neonicotinoid-treated or control fields)? If so, this could be more clearly explained.	We reworded the sentence to remove the ambiguities in the description of the results.
3	17	L186: revise "treated and untreated fields" here.	Revised as suggested.
3	18	L187: revise "treated fields" to "neonicotinoid treated fields".	Changed to "clothianidin-treated fields".
3	19	L188: revise "untreated fields" to "control fields".	Revised as suggested.
3	20	L197: revise "treated fields and 13 from untreated fields".	Revised as suggested.

3	21	L198: revise "treated" to "neonicotinoid treated".	Changed to "clothianidin-treated".
3	22	L215: "...mostly ubiquitous in all colonies, ..." This is loosely and inaccurately phrased. Ubiquitous has a specific meaning. If these were not ubiquitous, then don't use the word.	The reviewer is correct. We re-worded the sentence.
3	23	L215: revise "treated and untreated".	Revised as suggested.
3	24	L234: revise to "neonicotinoid treated fields".	Changed to "clothianidin-treated fields"
3	25	L250: revise "declining stronger" to "declined more strongly."	Changed to "declining more quickly".
3	26	L254: revise from "seed-treated" to "neonicotinoid treated" (both fungicide and neonicotinoid are "seed treatments in this system).	Changed to "clothianidin-treated".
3	27	L261 & 263: replace "seed-treatment" with "neonicotinoid treatment".	Changed to "clothianidin-treated".
3	28	L292: what do the authors mean by the term "high adaptive capacity" used here? This requires further explanation and clarification. Perhaps a different term would be more appropriate here - e.g. "strong resilience".	This request is similar to point #8 (L110). 'Resilience' is too static a term for what we wish to convey. We have re-phrased the sentence to avoid 'adaptive', which may be ambiguous.
3	29	L318: "Also the excessive colony mortality..." Is it reasonable to describe the mortality as "excessive" here? I suggest "higher colony mortality" would be a more appropriate descriptor here.	Changed to "high colony mortality".

3	30	L328-330:"..were left slightly understrength relative to the control colonies by the beekeeper interventions in June 2014, and may have overcompensated as a result." This seems like important information that needs further explanation in the paper. Why is the reader only hearing about such important impacts of apiary management on the outcome of the study at this stage in the manuscript?	We moved this information earlier in the manuscript, immediately with the first description of the results concerned, which is indeed a more appropriate point for mentioning this information (see lines 173 – 179).
3	31	L390: revise to "neonicotinoid treated fields."	Changed to “clothianidin-treated fields”.
3	32	L395: "These variations might be the result of different foraging alternatives in the different landscapes or differences in clothianidin concentrations in the plants.' Different/ce used 3 times in this sentence. Consider alternative words for 1 or 2 instances in this sentence.	Revised as suggested.
3	33	L481: Insert "and" so reads "One hundred and sixteen honeybee..."	Revised as suggested.
3	34	L509: revise to "exposed/unexposed to neonicotinoids".	Changed to “exposed/unexposed to clothianidin”.
3	35	L516: Suggest you revise from "The colonies were equalized as described for 2013,..." to "The strength of colonies were equalized as described for 2013,..."	Revised as suggested.
3	36	L570: revise "treated and untreated".	Changed to “clothianidin-treated and control”.
3	37	L681: revise from "...to the treatment from..." to "...to the neonicotinoid treatment from..."	Changed to “to the clothianidin treatment from..”
3	38	L686: revise "treated fields".	Revised as suggested.
3	39	L854: "spp." should not be in italics.	Revised as suggested.

3	40	L866: Please give correct details of volume and page numbers here. This should be: "Nat. Ecol. Evol. 1, 1308-1316 (2017)."	Revised as suggested.
3	41	L884: the article number is missing here. Should be: "PLoS One 6, e20656 (2011)."	Revised as suggested.
3	42	L892: the article number is missing here. Should be: "PLoS One 9, e103592 (2014)."	Revised as suggested.
3	43	L901-902: volume/ page number info is wrong here. Should be: "Ecotoxicology 27, 772-783 (2018)".	Revised as suggested.
3	44	L990: "untreated (control fields, white) seeds" see comment relating to L116 above.	Changed to: "In 2013 (squares) 16 oilseed rape fields and in 2014 (circles) 10 oilseed rape fields were either sown with insecticide-coated (black) or untreated (control fields, white) seeds coated with clothianidin and a fungicide (black) or only a fungicide (control fields, white)."
		Referee #4	Response:
4	45	While the authors note in the Discussion, "The importance of the results presented here is that in fundamentally healthy colonies, like the ones studied here, the natural homeostatic health mechanisms mediating the colony's response to its environment are robust enough to overcome these impairments, even in undersized colonies" this should be elaborated on so that the reader fully understands what the treatment actually consisted of (1 month of exposure and 11 months of relatively optimized management conditions). This information should also be included in the abstract.	Unfortunately, to stay within the limit of 150 words for the abstract we could not add anything more. In the discussion we changed the sentence to: "The importance of the results presented here is that in fundamentally healthy colonies, like the ones studied here, the natural homeostatic health mechanisms mediating the colony's response to its environment are robust enough to overcome these impairments, even in undersized colonies, during two consecutive years of one month direct exposure in the middle of the short Swedish bee foraging season."